# ASK ME ANYTHING: A SIMPLE STRATEGY FOR PROMPTING LANGUAGE MODELS

**Simran Arora\*, Avanika Narayan\* , Mayee Chen, Laurel Orr, Neel Guha, Kush Bhatia, Ines Chami, Christopher Ré**

`{simarora,avanika,mfchen,lorr1,nguha,kushb,chrismre}@cs.stanford.edu`
`{ines.chami}@numbersstation.ai`

## ABSTRACT

Large language models (LLMs) transfer well to new tasks out-of-the-box simply given a natural language prompt that demonstrates how to perform the task and no additional training. Prompting is a brittle process wherein small modifications to the prompt can cause large variations in the model predictions, and therefore significant effort is dedicated towards designing a painstakingly crafted *perfect prompt* for a task. To mitigate the high degree of effort, we instead ask whether collecting multiple decent, yet imperfect, prompts and aggregating them can lead to a high quality prompting strategy. Our observations motivate our proposed method, ASK ME ANYTHING PROMPTING (AMA). We first develop an understanding of the effective prompt formats, finding question-answering (QA) prompts, which encourage open-ended generation ("Who went to the park?") tend to outperform those that restrict the model outputs ("John went to the park. True or False?"). AMA recursively uses the LLM to transform task inputs to the effective QA format. AMA generates multiple questions per input and applies these prompts to collect several noisy *votes* for the input's true label. We find the prompts have varying accuracies and dependencies and thus propose to use weak supervision, a procedure for combining the noisy predictions, to produce the final predictions. We evaluate AMA across open-source model families (EleutherAI, BLOOM, OPT, and T0) and sizes (125M-175B parameters), demonstrating an average performance lift of 10.2% over the few-shot baseline. This simple strategy enables the *open-source* GPT-J-6B model to match and exceed the performance of *few-shot* GPT3-175B on 15 of 20 popular benchmarks. Averaged across these tasks, the GPT-J-6B model outperforms few-shot GPT3-175B. We release our code here: https://github.com/HazyResearch/ama_prompting.

## 1 INTRODUCTION

Large language models (LLMs) are bringing us closer to the goal of task-agnostic machine learning (Brown et al., 2020; Bommasani et al., 2021). Rather than training models for new tasks, LLMs are applied to new tasks out-of-the box with no additional training. In this paradigm, termed *in-context learning*, LLMs are controlled through user-provided natural language specifications of the task, or *prompts*, which illustrate how to complete a task. A prompt is defined by a template which contains placeholders for in-context demonstrations of the inputs and outputs for the task.

Recent work has evaluated LLM prompting performance on a broad set of tasks and finds the process to be brittle — small changes to the prompt result in large performance variations (Zhao et al., 2021; Holtzman et al., 2021). The performance further varies depending on the chosen LLM *family* (Ouyang et al., 2022; Sanh et al., 2022, inter alia.) and model size (Wei et al., 2022c; Lampinen et al., 2022). To improve reliability, significant effort is dedicated towards designing a painstakingly *perfect prompt*. For instance, Mishra et al. (2021) and Wu et al. (2022) recommend that users manually explore large search-spaces of strategies to tune their prompts on a task-by-task basis.

This work instead considers aggregating the predictions of multiple effective, yet *imperfect prompts* to improve prompting performance over a broad set of models and tasks. Given a task input, each prompt produces a *vote* for the input's true label, and these votes are aggregated to produce a final prediction. In pursuit of high quality prompting via aggregation, we face the following challenges:

1. **Effective prompts**: High quality prompts are a precursor to improvements from aggregation. We take the original prompts which yield near-random performance using the GPT-3 model in

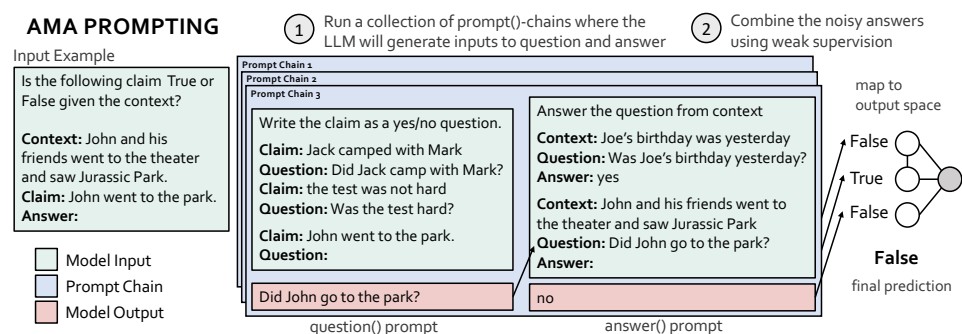

Figure 1: AMA uses the LLM itself to reformat task inputs to more effective formats. AMA creates *multiple* reformatted prompts per input. The LLM predictions from the prompts are aggregated using weak supervision.

Brown et al. (2020) for two SuperGLUE tasks (CB, RTE). Generating multiple prompts in the same format and taking majority vote prediction across prompts has a minor effect (+4% for CB) and can even hurt performance versus the average prompt performance (-2% for RTE). Many proposals for improved prompts focus on a single task type and evaluate on a single model-family and/or size (Wei et al., 2022c; Jung et al., 2022). We need a structure for prompting that improves quality across tasks and models.

2. **Scalable collection**: After identifying effective prompt formats, we need to obtain such prompts at scale. The original format of a task varies widely, and prior works manually rewrite each task input to new formats (Mishra et al., 2021; Wu et al., 2022), which is challenging to scale. Generating multiple prompts per input increases the difficulty.

3. **Prompt aggregation**: Using the prompts above (for CB and RTE), we see 9.5% average variation in accuracy and that the Jaccard index over errors is 69% higher than if prompt errors were i.i.d., suggesting highly correlated prompt outputs. Majority vote (MV) is the primary unsupervised aggregation strategy in prior prompting work (Jiang et al., 2020; Schick & Schütze, 2021), but it does not account for either property, making it unreliable. We need a strategy that accounts for the varying accuracies and dependencies.

We propose ASK ME ANYTHING PROMPTING (AMA), a simple approach that enables open-source LLMs with 30x fewer parameters to exceed the *few-shot* performance of GPT3-175B. In AMA:

1. **We identify properties of prompts that improve effectiveness across tasks, model types, and model sizes**. We study standard prompt-formats categorized by prior work (Brown et al., 2020) and find prompts that encourage open-ended answers ("Where did John go?") to be more effective than prompts that restrict the model output to particular tokens (e.g. "John went to the park. Output True or False?"). For instance, converting three SuperGLUE tasks (CB, RTE, WSC) from the original restrictive formats in (Brown et al., 2020) to open-ended formats provides a 72% performance improvement (Section 3.2). Given a task input, we find that a simple structure of (1) forming questions based on the input and (2) prompting the LLM to answer the questions applies quite generally and improves performance across diverse benchmark tasks.

2. **We propose a strategy for scalably reformatting task inputs to the effective formats found in (1).** We propose to transform task inputs to the effective open-ended question-answering format by recursively *using the LLM itself* in a task-agnostic two step pipeline. We first use question()-prompts, which contain examples of how to transform statements to various (e.g., yes-no, cloze) questions and second use answer()-prompts that demonstrate ways of answering questions (e.g., concise or lengthy answers). Applying *prompt-chains*— answer(question(x))—gives a final prediction for the input $x$.[1] Chains are (1) reused across inputs and (2) different pairs of functional prompts can be combined to create variety. We apply the varying functional prompt-chains to an input to collect multiple votes for the input's true label.

3. **We propose the use of weak supervision (WS) to reliably aggregate predictions.** We find that the errors produced by the predictions of different chains can be highly varying and correlated. While majority vote (MV) may do well on certain sets of prompts, it performs poorly in the above cases. AMA accounts for these cases by identifying dependencies among prompts and using WS, a procedure for modeling and combining noisy predictions without any labeled data (Ratner et al., 2017; Varma et al., 2019). We apply WS to prompting broadly for the first time in this work,

---

[1]We draw inspiration from Wu et al. (2022) and focus on task-agnostic and scalable prompt-chains.

showing it improves the reliability of prompting with off-the-shelf LLMs and no further training. We find that AMA can achieve up to 8.7 points of lift over MV and recovering dependencies boosts performance by up to 9.6 points.

We apply our proposed prompt-aggregation strategy, AMA, to 20 popular language benchmarks and 14 open-source LLMs from 4 model families (Neo, BLOOM, OPT, and T0) spanning 3 orders-of-magnitude (125M-175B parameters). The simple strategy provides an improvement over the few-shot ($k = 3$) baseline by an average of 10.2% $\pm$ 6.1% absolute (21.4% $\pm$ 11.2% relative) lift across models. We find the largest gains are on tasks where the knowledge required to complete the task is found in the provided context and comparatively less on closed-book tasks (e.g., factual recall). Most excitingly, ASK ME ANYTHING PROMPTING enables an open-source LLM, which is 30x parameters smaller, to match or exceed the challenging GPT3-175B *few-shot* baseline results in Brown et al. (2020) on 15 of 20 benchmarks. We hope AMA helps address painpoints in widely applying in-context learning (Arora & Ré, 2022; Narayan et al., 2022) by improving the ability to proceed with less-than-perfect prompts and encouraging the use of small, open-source LLMs.

## 2 RELATED WORK

Several existing works seek to improve the zero-to-few-shot task-transfer abilities of LLMs.

**Training based strategies**  Prior works have improved prompting performance by training larger models over more or curated data, and for longer (Kaplan et al., 2020; Chowdhery et al., 2022) — or by explicitly fine-tuning LMs over prompts (Wang et al., 2022b; Wei et al., 2022a; Sanh et al., 2022; Ouyang et al., 2022). We complementarily aim to improve the prompting performance of off-the-shelf language models with no additional fine-tuning.

**Prompt-engineering**  Prompt-engineering is the process of designing natural language specifications of a task, which are used to condition the LLM at inference time. Prior work finds that the prompt format changes the model behavior and proposes particular formats. Some formats are designed-for or evaluated-on a narrow task type, model type, or model size (Wei et al., 2022c; Jung et al., 2022). Others require users to *manually* rewrite task-inputs to the prescribed formats on a example-by-example basis in a task-specific manner (Mishra et al., 2021; Wu et al., 2022). Our recursive use of the LLM is similar to Jung et al. (2022), which focuses on commonsense reasoning. We draw inspiration from these lines of work and investigate a broader set of tasks and model sizes.

Complementary work investigates how to simplify complex tasks (e.g., logical, compositional, and multi-hop reasoning), to achieve better performance in the prompting paradigm. Creswell et al. (2022); Wu et al. (2022); Zhou et al. (2022); Yang et al. (2022) explicitly *decompose* the complex tasks into steps, which are each handled in a separate inference-pass. However, these methods draw a distinction between the explicitly compositional tasks which can be naturally decomposed into multiple steps and "single-step" language tasks. These prior works do not support the single-step tasks such as classification, NLU, QA, which are the focus of our work.

**Prompt aggregation**  Prior works note the sensitivity of prompting under slight modifications and propose strategies to improve the performance of single prompts (Zhao et al., 2021; Liu et al., 2021). Complementing this, we aggregate the results of multiple prompts. Shi et al. (2022) observes that different prompt example selections yield different results, and suggests combining the results of different prompts as an exciting future direction. Prompt aggregation has been applied in several prior works. Many works train models to perform the aggregation and/or to achieve strong results with small LMs (Jiang et al., 2020; Schick & Schütze, 2021; Cobbe et al., 2021; Zelikman et al., 2022, inter alia.). Self-Consistency Wang et al. (2022a), which requires no training, does not report improvements for small LMs (<10B parameters). We also compare AMA to Self-Consistency in Appendix B. The *unsupervised* aggregation strategy used in prior works is Majority Vote — we are the first to use Weak Supervision for unsupervised prompt aggregation.

**Weak supervision (WS)**  WS is a powerful framework that learns the accuracies and correlations of multiple noisy sources and aggregates them to produce weak labels for training data (Ratner et al., 2017; 2016; 2018; Varma et al., 2019; Fu et al., 2020). WS has been applied to prompting in the context of distilling a LLM by aggregating the outputs of hand-curated prompts into a labeled dataset and training a smaller model on it (Smith et al., 2022). In contrast, we aim to use aggregation to improve out-of-the-box LLM performance reliably, which has not previously been explored.

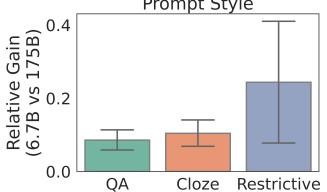

| Prompt Format | CB | WSC | RTE |
|---|---|---|---|
| Restricted 8-shot | 22.0 | 50.0 | 53.0 |
| + Calibration | 58.9 | - | 59.2 |
| Cloze-QA Format | 48.2 | 56.0 | **62.5** |
| Standard-QA Format | **83.3** | **69.2** | 62.0 |

Figure 2: Relative lift with model scale using results and prompt-styles reported in Brown et al. (2020) (Left). Ablating the prompt-style using the GPT-J-6B model. We include calibration results Zhao et al. (2021) and the "-" indicates the method cannot be applied to the task (Right).

## 3 ASK ME ANYTHING PROMPTING

We propose ASK ME ANYTHING PROMPTING (AMA), a prompting approach that uses multiple *imperfect* prompts—rather than one painstakingly crafted *perfect* prompt—and reliably aggregates their outputs. We describe and motivate AMA's prompt format (Section 3.2), how AMA scalably produces collections of prompts (Section 3.3), and AMA's aggregation method (Section 3.4).

### 3.1 PRELIMINARIES

We consider supervised tasks, $(\mathcal{X}, \mathcal{Y})$, where $x \in \mathcal{X}$ is the example and $y \in \mathcal{Y}$ is the output. We have an unlabeled dataset $\mathcal{D} = \{x_i\}_{i=1}^n$ for which we wish to predict each $y_i$. We apply LLMs to this task by using a prompt—a natural language prefix that demonstrates how to complete a task. A prompt consists of a prompt template, with placeholders for (1) zero or more in-context task demonstrations and (2) for the inference example $x$ as shown in Figure 3. Given a prompt $p$, we use $p : \mathcal{X} \to \mathcal{Y}$ to refer the output of the prompted LLM which produces a prediction $\hat{y} = p(x)$. Specifically, the LLM runs inference on $p$ with $x$ substituted for the placeholder in the template.

We denote a collection of $m$ prompts as $\mathbf{P} = [p_1, p_2, ..., p_m]$. Given input $\mathcal{D}$, we (1) apply a collection $\mathbf{P}$ to each $x \in \mathcal{D}$ and (2) aggregate their predictions, denoted as $\mathbf{P}(x) = [p_1(x), \ldots, p_m(x)]$, using an aggregator function $\phi : \mathcal{Y}^m \to \mathcal{Y}$ to produce outputs $\hat{y}$ on each $x$. We can thus express the procedure via two key components we aim to understand, the prompts $\mathbf{P}$ and aggregator $\phi$.

**Running examples** For the motivating observations in the rest of this section, we use three SuperGLUE (Wang et al., 2019) tasks—CommitmentBank (CB), Recognizing Textual Entailement (RTE), and Winograd Schema Challenge (WSC)—and the DBPedia and AGNews classification tasks (Zhang et al., 2015). We evaluate over the GPT-J-6B model. CB and RTE require determining the validity of a statement given some context (as in Figure 1), WSC requires outputting the subject corresponding to a given pronoun, and DBPedia and AGNews contain 14 and 4 classes, respectively. We use as a running example: determine if the statement "John went to the park" is valid, given the context "John invited Mark to watch Jurassic Park with his family at the theater".

**Simple baseline** We take the prompts proposed in (Brown et al., 2020) for GPT-3 and produce $\mathbf{P}$ with five prompts for each task by using different sets of in-context examples. Comparing majority vote (MV), the unsupervised aggregation strategy used in prior work, to the average performance of the prompts, MV gives 39.3% (+2.2%) for CB and 54.5% (-2%) for RTE. The delta from aggregating is minor and in the worst case, harmful. The need for effective prompts and a reliable aggregation strategy motivate our study. We provide similar results on additional tasks in Appendix B.

### 3.2 EFFECTIVE PROMPT FORMATS

First, we explore what makes an effective prompt format towards improving the quality of $\mathbf{P}(x)$.

**Standard prompt formats** We ground our analysis in three standard categories of prompts used in prior work including Brown et al. (2020); Sanh et al. (2022, inter alia.): (1) questions that **restrict** the model output particular tokens ("John invited Mark to come watch Jurassic Park. Output True or False?"); (2) **cloze-questions** which ask the model to fill in the remaining text ("John invited Mark to come watch Jurassic _" and using the LLM to fill-the-blank, "*Park*"); and (3) traditional (yes-no, *Wh*) **free-form questions** ("Where did John invite Mark?"). Compare these three formats, we see:

1. **Open-ended prompts appear to outperform restrictive-prompts.** We first group the results in Brown et al. (2020) based on the format used for the task, along the above categorizations (see Figure 2). When scaling from GPT3-6.7B to GPT3-175B, we find that the relative gain is far lower on open-ended (cloze and traditional QA) formats vs. restricted formats.

Next, CB, RTE, and WSC are originally formatted with restrictive-prompts in Brown et al. (2020), and we form copies of the tasks in the open-ended question (cloze and free-form QA) formats. This improves the performance of the small model on average from 41.7% to 71.5% (+72%) . Intuitively, the task of answering open-ended questions is aligned with the next-token prediction language modeling objective. We observe that more precise questions give larger lifts. For WSC the restrictive prompt form is: "The pronoun 'his' refers to "Mark" in the context. True or False?", given the context "Mark went to the park with his dog.". Reformatting to "What does 'his' refer to?" and evaluating whether the answer is "Mark" provides 38% lift (69.2% accuracy). Yet, further extracting the portion of the context that mentions the pronoun ("his dog"), reformatting ("Whose dog?") and prompting with *precise* questions gives 49.4% lift (74.7%).

2. **The use of open-ended questions over restrictive-prompts can increase the difficulty of mapping open-ended answers to valid output classes.** For tasks with output spaces that are likely observed during pretraining (yes-no questions, sentiment classification), we see that the LLM naturally generates valid $\hat{y} \in \mathcal{Y}$. For tasks with specialized output classes (i.e. multi-class classification), we need to map the answer to the open-ended question (e.g., "What is the document about?") to a valid output class. For example, given 'Personality and Mental Health ... is a quarterly peer-reviewed academic journal published by ...", we observe that the LLM typically outputs *semantically* correct summaries of the document topic, e.g. "journal". We find that inserting a step for the LLM to *map* the open-ended output "journal" to a valid category via the prompt "A 'journal' maps to category: written work" enables a 33.3% and 11.1% lift over the few-shot baseline on DBPedia (14-way classification) and AGNews (4-way) respectively.

**Why is the QA prompt format effective?**  We analyze the LM pretraining corpus to better understand why the proposed QA prompt template may be effective. The EleutherAI models are trained on The Pile corpus Black et al. (2021); Wang & Komatsuzaki (2021); Gao et al. (2021). Over a 2% random sample of the ~200B token Pile data, we find that open-ended QA structures (i.e., which ask the model "Is ...?", "Who ...?") appear on the order of $1000\times$ more frequently than the restrictive-prompt structures (i.e., which instruct the model to output "True or False", "Yes or No") (see Table 10). Further, when applying the few-shot restrictive prompts, we observe large imbalances in the F1-scores for different classes (Table 12) and question whether answering the restrictive prompts is challenging due to biases acquired during pretraining. We find that there are large imbalances in Pile between the frequencies of "yes" vs. "no", and "True" vs. "False". Details are in Appendix H.

**AMA's prompt format**  Motivated by the two observations above, we proceed in AMA with a two-step prompting pipeline: (1) generating questions based on the input and (2) prompting the LLM to answer the generated questions. These prompts are effective, and to further improve performance we next turn to generating and aggregating over *multiple* prompt-outputs for each input. For intuition, different questions (with our running example: "Who went to the park?", "Did John go the park?", "Where did John go?") emphasize different aspects of the input and can provide complementary information towards reasoning about the answer. Manually generating multiple prompts per input is challenging, and so we study how to do this at scale in the following section.

## 3.3 CREATING PROMPT COLLECTIONS AT SCALE

Our goal is to produce a collection of prompts, $\mathbf{P}$, that can be applied to tasks at scale. To produce prompts in the open-ended question-answering format, we recursively apply the LLM itself using a *chain* of *functional* prompts, referred to as a $\mathrm{prompt}()$-chain. We describe these prompts as functional because they apply a task-agnostic QA prompting template to all inputs in the tasks. We describe the two functional prompts used in AMA below. We use Figure 1 as a running example.

(a) $\mathrm{question}()$: $x \to q$ generates a question $q$ (such as "Did John go to the park?") from an input $x$ ("John went to the park."). $\mathrm{question}()$ prompts simply contain demonstrations of how a statement can be transformed to an open-ended question.

(b) $\mathrm{answer}()$: $q \to a$ applies the question generated by (a) to the context of $x$ to produce intermediate answers $a$ (such as "No" or "theater"). The $\mathrm{answer}()$ prompts contain demonstrations of how to answer a question (optionally) given some input context.

To create $\mathbf{P}$ for aggregation, AMA constructs different $\mathrm{prompt}()$-chains where each unique $\mathrm{prompt}()$-chain is a different *view* of the task and can emphasize different aspects of $x$. Inspired by Sanh et al. (2022) and Liu et al. (2021), we also vary chains through two key levers—the in-context demonstrations and the style of prompt questions—as shown in Figure 3. To vary the style

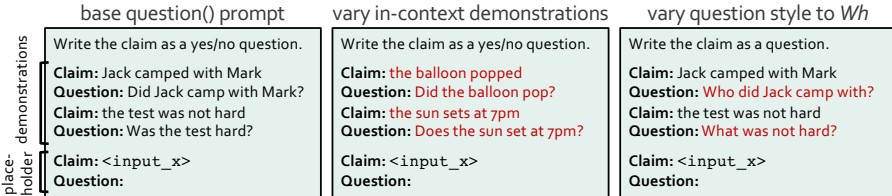

Figure 3: Example prompt with the in-context demonstrations and placeholder (Left) with two different prompt variations (Right) created by changing demonstrations and question style.

of open-ended prompt questions, we construct $\mathrm{question}()$ and $\mathrm{answer}()$ prompts that produce and answer either Yes/No, *Wh*, multiple-choice, or cloze- questions.

### 3.4 PROMPT AGGREGATION

To aggregate the prompt predictions $\mathbf{P}(x)$ into outputs $\hat{y}$ reliably, we apply tools from weak supervision, a powerful approach for learning high-quality models from weaker sources of signal *without labeled data* (Ratner et al., 2017). We first describe properties of $\mathbf{P}(x)$ that illustrate when the simple baseline of majority vote may perform poorly. We then describe our aggregator $\phi_{\mathrm{WS}}$, which explicitly identifies and then accounts for these properties.

**Baseline observations** To understand how to aggregate $\mathbf{P}(x)$, we present a set of observations on CB, RTE, and WSC. For each, we compare two baselines for constructing $\mathbf{P}$: (1) $\mathbf{P}_\mathrm{T}$: varying the prompt template with no overlap in the in-context examples, and (2) $\mathbf{P}_\mathrm{E}$: varying the in-context examples for a fixed prompt template, all with $|\mathbf{P}| = 5$. We observe the following properties on $\mathbf{P}$:

1. *Varied overall accuracies*: While prompts in $\mathbf{P}_\mathrm{E}$ may seem more similar than those in $\mathbf{P}_\mathrm{T}$, the gap between the best and worst $p_i \in \mathbf{P}$ is large in both cases — 12.1% for $\mathbf{P}_\mathrm{E}$ and 9.6% for $\mathbf{P}_\mathrm{T}$.

2. *Varied class-conditional accuracies* (Zhao et al., 2021): Beyond overall prompt accuracy, the average variance of class-conditional prompt accuracies is 9.7% across the tasks and baselines.

3. *Highly-correlated outputs*: Prompt predictions have dependencies among each other. The Jaccard index over error sets averaged across tasks is 42.2 for $\mathbf{P}_\mathrm{E}$ and 39.9 for $\mathbf{P}_\mathrm{T}$. For reference, two prompts that produce i.i.d. errors and have 60% accuracy each would have a score of 0.25.

The three observations present challenges in aggregating predictions via simple approaches like MV. MV tends to do better than using one prompt, but it weights all prompts equally and treats them independently. Such an aggregation method may be sufficient over certain collections of prompts but is not reliable across general $\mathbf{P}$ that may exhibit the three properties we have observed.

**AMA Aggregation** Given the varied accuracies and dependencies among $\mathrm{prompt}()$-chains, we draw on recent work in weak supervision (Ratner et al., 2017), which can aggregate outputs while accounting for the accuracy and dependency properties without relying on labeled data. We learn a probabilistic graphical model on $\mathrm{Pr}_{G,\theta}(y, \mathbf{P}(x))$ and define the aggregator as $\phi_{\mathrm{WS}}(x) = \arg\max_{y \in \mathcal{Y}} \mathrm{Pr}_{G,\theta}(y \mid \mathbf{P}(x))$. $G = (V, E)$ is a dependency graph where $V = \{y, \mathbf{P}(x)\}$ and $E$ is an edgeset where $(p_i(x), p_j(x)) \in E$ iff $p_i(x)$ and $p_j(x)$ are conditionally dependent given $y$. $\theta$ are the accuracy parameters for $\mathbf{P}(x)$. Since we lack labeled data $y$, we cannot estimate $G$ or $\theta$ directly from $\mathcal{D}$. Our procedure is as follows:

1. We use a structure learning approach from Varma et al. (2019) to recover the dependency structure $\hat{G}$ using $\mathbf{P}(x)$ applied to $\mathcal{D}$.

2. We use $\hat{G}$, $\mathcal{D}$, and $\mathbf{P}(x)$ to learn the accuracies $\theta$ of the prompts $\mathbf{P}$ from Ratner et al. (2018).

3. We compute $\mathrm{Pr}_{\hat{G},\hat{\theta}}(y \mid \mathbf{P}(x))$ and aggregate our predictions.

The key insight is that the inverse covariance matrix on $V$, $\Sigma^{-1}$, is *graph-structured*, meaning that $\Sigma_{ij}^{-1} = 0$ iff $p_i(x)$ and $p_j(x)$ are conditionally independent given $y$. This property yields systems of equations on $V$ from which we can recover dependencies and accuracies.

Algorithms summarizing the end-to-end AMA procedure are in Appendices D and E respectively.

## 4 INFORMATION FLOW IN AMA

Before evaluating end-to-end quality, we look at a simple information theoretic metric to understand the contributions of the individual components — $\mathbf{P}$ and $\phi$ — in the prompting procedure.

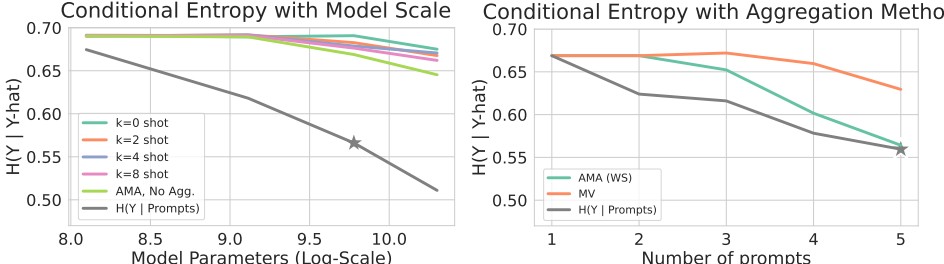

Figure 4: $H(y|\hat{y})$ as a function of model size $\in \{125M, 1.3B, 6B, 20B\}$ and prompts $p$ with $k = \{0, 2, 4, 8\}$ in-context examples (Left), and the aggregation strategy over the prompts, majority vote (MV) and weak supervision (WS) with the 6B model (Right). $H(y|\mathbf{P}(x))$ is measured in both plots. Plots are over RTE with GPT-J-6B; each k-shot point is the average of 4 seeds. Plots for BLOOM models are in Appendix C.

**Information flow metric** Specifically, we examine the conditional entropy, $H(y|\hat{y})$, which measures the amount of uncertainty remaining in the true label $y$ given a prediction $\hat{y}$. Intuitively, $H(y|\hat{y})$ will be low when $\hat{y}$ encodes information relevant to $y$. In our setting, $\hat{y} = \phi(\mathbf{P}(x))$ is dependent on the two components of the prompting procedure, the prompts $\mathbf{P}$ and aggregator $\phi$. The following simple decomposition of $H(y|\hat{y})$ enables studying the contribution of each component:

$$H(y|\hat{y}) = \underbrace{H(y|\mathbf{P}(x))}_{\text{Controlled by } \mathbf{P} \text{ prompt quality}} + \underbrace{H(y|\hat{y}) - H(y|\mathbf{P}(x))}_{\text{Controlled by aggregation method } \phi} \qquad (1)$$

The first term $H(y|\mathbf{P}(x))$, $H(y|\hat{y})$ depends on the quality and quantity of the individual prompts in $\mathbf{P}(x)$ (since $H(y|\mathbf{P}(x)) \leq H(y|p(x))$). A set of prompts that contains relevant information for $y$ contributes to a low $H(y|\hat{y})$. The second term $H(y|\hat{y}) - H(y|\mathbf{P}(x))$ shows that $H(y|\hat{y})$ depends on how the aggregation step compresses the information in $\mathbf{P}(x)$ to predict $\hat{y}$. An aggregator $\phi$ that more accurately matches the true $\Pr(y|\mathbf{P}(x))$ reduces the information loss in the compression step.

**Evaluation** We use (1) to evaluate our proposed solution AMA. First considering $H(y|\mathbf{P}(x))$, in Figure 4 (Left) we observe AMA outperforms $k$-shot baselines with expected scaling in terms of both individual $\mathrm{prompt}()$-chain quality (as shown by AMA No Agg.) and their quantity.

Next we consider the gap term $H(y|\hat{y}) - H(y|\mathbf{P}(x))$. It enables us to understand why MV is insufficient: it compresses information from $\mathbf{P}(x)$ according to a specific construction of $\Pr(y, \mathbf{P}(x))$, for which $p_i(x) \perp p_j(x)|y$ for all $i, j \in [m]$, and $\Pr(p_i(x) = c|y = c)$ for $c \in \mathcal{Y}$ is a *single* better-than-random constant across $i$ and $c$. When the true distribution is vastly different—as is common—this results in a large gap between the optimal $H(y|\mathbf{P}(x))$ and $H(y|\hat{y}_{\text{MV}})$ in Figure 4 (Right). WS can improve $\phi$ over the standard MV baseline to reduce the information loss $H(y|\hat{y}_{\text{AMA}}) - H(y|\mathbf{P}(x))$. In addition to empirical measurements, we can provide a theoretical characterization for the information flow. In Appendix F, we express $H(y|\hat{y}_{\text{AMA}})$ in terms of the individual prompt accuracies under the standard weak supervision model (i.e., Ising model on $y$ and $\mathbf{P}(x)$ (Ratner et al., 2018)).

There has been recent interest in how LLMs improve primarily along the three axes of parameter scale, training data, and compute (Kaplan et al., 2020; Hoffmann et al., 2022; Wei et al., 2022b). In Figure 4, as we increase the number of prompts to be aggregated, the conditional entropy reduces. Prompt aggregation may be another useful axis for understanding LLM scaling performance.

## 5 RESULTS

We evaluate AMA on 20 popular language benchmarks used in Brown et al. (2020); Sanh et al. (2022). We report results across 14 unique LMs including 4 model families (Neo (Black et al., 2021), OPT (Zhang et al., 2022), BLOOM, and T0 (Sanh et al., 2022)) spanning 125M-175B parameters. We aim to validate whether AMA provides consistent lift across diverse tasks (Section 5.1) and model families (Section 5.2), and reliably aggregates the predictions across prompts (Section 5.3).

**Experimental details** In Table 1, we compare to the **few-shot**-prompted GPT3-175B LM using the numbers published in Brown et al. (2020); Zhao et al. (2021), given the model's popularity and strong off-the-shelf quality. Brown et al. (2020) uses $k \in [32..70]$ and Zhao et al. (2021) uses $k = 8$ depending on the task, providing a challenging baseline. We evaluate using the same tasks on which GPT-3 was originally evaluated: SuperGLUE (Wang et al., 2019), NLI (Mostafazadeh et al., 2017; Nie et al., 2020), classification (Zhang et al., 2015; Socher et al., 2013; He & McAuley, 2016), and QA tasks (Kasai et al., 2022; Kwiatkowski et al., 2019; Berant et al., 2013; Dua et al., 2019).

| Model | Neo Few-Shot | Neo (QA) | Neo (QA + WS) | GPT-3 Few-Shot | Fine-Tuned |
|---|---|---|---|---|---|
| # Params | 6B | 6B | 6B | 175B | - |
| Natural Language Understanding | | | | | |
| BoolQ | $64.8_{(3)}$ | 64.8 | $67.2_{\pm0.0}$ | $\mathbf{77.5}_{(\mathbf{32})}$ | 91.0 |
| CB | $23.2_{(3)}$ | 83.3 | $\mathbf{83.9}_{\pm0.0}$ | $82.1_{(32)}$ | 96.9 |
| COPA | $79.0_{(3)}$ | 58.2 | $84.0_{\pm0.0}$ | $\mathbf{92.0}_{(\mathbf{32})}$ | 94.8 |
| MultiRC | $60.8_{(3)}$ | 58.8 | $63.8_{\pm0.0}$ | $\mathbf{74.8}_{(\mathbf{32})}$ | 62.3 |
| ReCoRD | $75.5_{(3)}$ | 74.5 | $74.4_{\pm0.0}$ | $\mathbf{89.0}_{(\mathbf{32})}$ | 92.5 |
| RTE | $58.0_{(3)}$ | 61.4 | $\mathbf{74.7}_{\pm0.0}$ | $72.9_{(32)}$ | 92.5 |
| StoryCloze | $56.0_{(3)}$ | 76.3 | $\mathbf{87.8}_{\pm0.0}$ | $87.7_{(70)}$ | 91.8 |
| WiC | $48.8_{(3)}$ | 59.0 | $\mathbf{61.3}_{\pm0.2}$ | $55.3_{(32)}$ | 76.1 |
| WSC | $50.0_{(3)}$ | 74.7 | $\mathbf{77.8}_{\pm0.0}$ | $75.0_{(32)}$ | 93.8 |
| Natural Language Inference | | | | | |
| ANLI R1 | $32.3_{(3)}$ | 34.5 | $\mathbf{37.8}_{\pm0.2}$ | $36.8_{(50)}$ | 73.8 |
| ANLI R2 | $33.1_{(3)}$ | 35.4 | $\mathbf{37.9}_{\pm0.2}$ | $34.0_{(50)}$ | 50.7 |
| ANLI R3 | $33.4_{(3)}$ | 37.0 | $\mathbf{40.9}_{\pm0.5}$ | $40.2_{(50)}$ | 48.3 |
| Classification | | | | | |
| AGNews | $74.5_{(3)}$ | 83.7 | $\mathbf{86.4}_{\pm0.0}$ | $79.1_{(8)}$ | 94.3 |
| Amazon | $53.1_{(3)}$ | 67.0 | $\mathbf{68.6}_{\pm0.0}$ | $41.9_{(8)}$ | 90.5 |
| DBPedia | $50.7_{(3)}$ | 81.4 | $\mathbf{83.9}_{\pm0.0}$ | $83.2_{(8)}$ | 99.0 |
| SST | $94.5_{(3)}$ | 94.5 | $\mathbf{95.7}_{\pm0.0}$ | $95.6_{(8)}$ | 97.5 |
| Question Answering | | | | | |
| DROP | $32.3_{(3)}$ | 51.0 | $\mathbf{51.6}_{\pm0.0}$ | $36.5_{(20)}$ | 88.0 |
| NQ | $13.7_{(3)}$ | 19.7 | $19.6_{\pm0.0}$ | $\mathbf{29.9}_{(\mathbf{64})}$ | 44.5 |
| RealTimeQA | $34.7_{(3)}$ | 34.7 | $\mathbf{36.0}_{\pm0.0}$ | $35.4_{(1)}$ | - |
| WebQs | $31.3_{(3)}$ | 44.2 | $\mathbf{44.1}_{\pm0.0}$ | $41.5_{(64)}$ | 45.5 |

Table 1: AMA results for the GPT-J-6B parameter model compared to the few-shot GPT3-175B. The GPT-175B numbers are as reported in Brown et al. (2020); Zhao et al. (2021), where the numbers of in-context examples is in parentheses. Note that prompts can *abstain* from predicting, which can lead to lower average numbers for **QA**, such as on COPA. For QA tasks and ReCoRD, we report the majority vote score, as applying WS is complex for open-domain tasks. Results for the BLOOM 7.1B parameter model are in Appendix 2.

For AMA we use 3-6 prompt()-chains to generate predictions per input. We model the dependencies and accuracies of each prompt-prediction per task, without using any labeled training data, to obtain the final aggregated prediction per input via weak supervision (WS). We report both the average performance over the prompt()-chains (**QA**) and with AMA's WS aggregation (**QA + WS**). We report **QA + WS** across 5 random seeds for the model. Further details are in the Appendix. [2]

## 5.1 MAIN RESULTS

We report benchmark results in Table 1 comparing the open-source GPT-J-6B and few-shot ($k \in [32..70]$) GPT3-175B. **We find that the open-source 6B parameter model exceeds the average few-shot performance of the GPT3-175B model on 15 of 20 benchmarks**. Overall, AMA gives a 37.6% improvement over the 6B parameter model's few-shot ($k = 3$) performance to achieve this.

We find that AMA provides the most lift on tasks where all requisite knowledge is included in the task input (e.g., reading comprehension) and that largely rely on model's NLU abilities. The lift is lower on tasks that rely on the LMs memorized knowledge (e.g. closed-book QA). However, on the closed-book WebQ task where the answers are likely seen during pretraining, we find that prompting the LM to generate relevant context, and then answer the original question using the generated context is effective. That being said, the closed-book NQ task shows there are limitations.

We similarly see limitations when tasks cannot rely on the latent knowledge. We observe a small performance gap between model sizes on RealTimeQA, which includes questions that have temporally changing answers that are less likely to be memorized. Similarly, for tasks requiring domain knowledge, e.g. the "Amazon Instant Video" class in the Amazon task, all model-sizes achieve near-0 performance. We provide an extended error analysis Table 1 results in Appendix I.

## 5.2 EVALUATION ACROSS MODELS

**Benchmark results** We evaluate the lift from AMA over out-of-the-box few-shot performance across different sizes of four open-source LMs (Neo, OPT, BLOOM, and T0) across 7 tasks (4

---

[2] We do not use rank-classification scoring, which is commonly used (Brown et al. (2020); Sanh et al. (2022)) to reduce task complexity, barring explicitly multiple-choice tasks (ReCORD, StoryCloze and COPA).

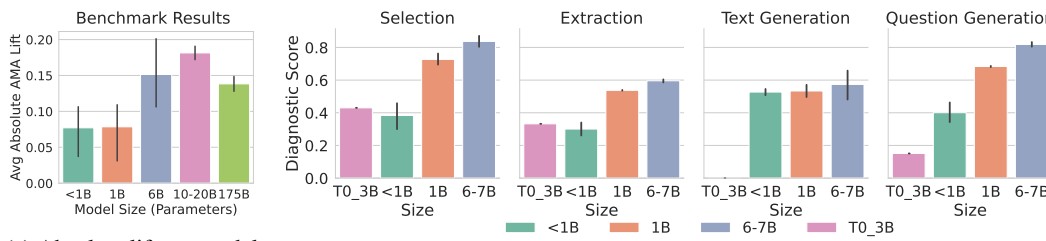

(a) Absolute lift vs. model size (parameters).

(b) Average diagnostic score vs. model size (parameters).

Figure 5: Evaluation across model sizes for diagnostics and benchmarks. We report the absolute lift from AMA over few-shot ($k = 3$) performance, averaged over 7 tasks with 95% confidence intervals (Left). Diagnostic plots are ordered by the amount of lift models of the size-category see on 7 the benchmarks (Right).

NLU, 2 NLI, 1 classification). In this analysis, we want to understand the effectiveness of AMA's prompt()-chains reformattings across models and report the average prompt performance over the 3-6 prompt()-chains used per task. Excitingly, the AMA prompt()-chains apply quite generally. We observe $10.2\% \pm 6.1\%$ absolute lift across models ($21.4\% \pm 11.2\%$ relative lift) on average across models and tasks, as shown in Figure 5a (a). We observe the absolute lift increases with model size and levels out. The average absolute (relative) lift by model family (across tasks and sizes) is 11.0% (24.4%) for Neo, 11.0% (23.4%) for BLOOM, and 11.9% (22.7%) for OPT, though only 2.9% (8.3%) for T0. T0 is a popular open-source (non-GPT) LM, which was fine-tuned on prompt-output pairs and transfers well to new tasks in a zero-shot fashion.

**Diagnostics for understanding AMA lift**   We next provide a set of diagnostics to better understand the model reasoning skills that correspond to the different degrees lift from AMA, including T0's limited benefit from AMA's prompt()-chains. The diagnostics measure four basic operations required in AMA—question generation, answer generation, answer selection, and extraction. For each operation, we create 1-3 tasks each with 50 manually-labeled samples (See Appendix G).

We measure the average performance across each operation across different sizes of 4 model types (Neo, OPT, BLOOM, and T0). We group models and sizes into four buckets of T0 (3B parameters) and GPT models ($< 1B$, $1B$, and $6 - 7B$ parameters). Figure 5b shows results where the buckets are ordered by their average AMA lift across the 7 tasks from Section 5.2, meaning T0 (3B) sees the least lift while $6 - 7B$ GPT models realize the most lift. We find that overall, models with higher performance across the four operations see more lift with AMA. T0 performs poorly on the generative tasks, indicating the importance of text and question generation for AMA.

## 5.3 Evaluation against other aggregation methods

We compare our WS aggregation approach with the standard unsupervised approach, majority vote (MV), on prompt()-chains. We find that AMA can achieve up to 8.7 points of lift over MV, and matches or outperforms MV on 16 out of 20 tasks. On the remaining 4 tasks, we perform worse than MV by at most 1.0 points. We also examine the effect of modeling dependencies in WS. We find that on 9 tasks, our approach recovers dependencies in the data (rather than assuming conditionally independent $\mathbf{P}(x)$), which improves performance by up to 9.6 points and an average of 2.2 points. We provide more details and evaluation against labeled data baselines in Table 7 (Appendix B.5).

Next, we evaluate T0 on zero-shot prompts from the public PromptSource (Bach et al., 2022), which are better aligned with how this model has been trained. Using the off-the-shelf prompts from PromptSource for 4 NLU tasks which T0 heldout during training, we see an average lift of 6.1 points when applying weak supervision over these prompts. Details are in Appendix B.3.

## 6 Conclusion

In this work we introduce Ask Me Anything Prompting which scalably obtains multiple effective prompts given a task input and combines the intermediate answers to these prompts using weak supervision to provide the final answer. The improvements of AMA stem from our observations on the effectiveness of open-ended questions over restrictive prompts, and the ability to model the varying accuracies and dependencies across a collection of prompts using weak-supervision. We hope this work highlights the importance of the prompt structure and encourages future work to improve the capabilities of small and open-source models.

## 7 REPRODUCIBILITY STATEMENT

We release prompts and code for reproducing all benchmark results for few-shot and AMA prompting, and our diagnostic evaluation splits here: https://github.com/HazyResearch/ama_prompting.

## 8 ETHICS STATEMENT

We intend for AMA to aid practitioners in their exploration and use of LLMs—especially smaller, open-source LLMs. However, we recognize that AMA could be used to perform harmful or unethical tasks. AMA is a proof-of-concept; it has error-modes and we recognize the inherent risks to using LLMs. Detailed discussions of these risks are in Bommasani et al. (2021); Weidinger et al. (2021).

## 9 ACKNOWLEDGEMENTS

The computation required in this work was provided by Together Computer (https://together.xyz/). We are grateful to the Numbers Station (https://numbersstation.ai/), Snorkel (https://snorkel.ai/), Stanford Center for Research on Foundation Models (https://crfm.stanford.edu/), and Stanford HAI (https://hai.stanford.edu/) organizations for the resources that supported this work. We thank Fred Sala, Karan Goel, Maya Varma, Joel Johnson, Sabri Eyuboglu, Kawin Ethayarajh, Niladri Chatterji, Neha Gupta, Alex Ratner, Percy Liang, and Rishi Bommasani for their helpful feedback and discussions. We gratefully acknowledge the support of NIH under No. U54EB020405 (Mobilize), NSF under Nos. CCF1763315 (Beyond Sparsity), CCF1563078 (Volume to Velocity), and 1937301 (RTML); US DEVCOM ARL under No. W911NF-21-2-0251 (Interactive Human-AI Teaming); ONR under No. N000141712266 (Unifying Weak Supervision); ONR N00014-20-1-2480: Understanding and Applying Non-Euclidean Geometry in Machine Learning; N000142012275 (NEPTUNE); NXP, Xilinx, LETI-CEA, Intel, IBM, Microsoft, NEC, Toshiba, TSMC, ARM, Hitachi, BASF, Accenture, Ericsson, Qualcomm, Analog Devices, Google Cloud, Salesforce, Total, the HAI-GCP Cloud Credits for Research program, the Stanford Data Science Initiative (SDSI), and members of the Stanford DAWN project: Facebook, Google, and VMWare. SA is supported by a Stanford Graduate Fellowship. LO is supported by an Intelligence Community Postdoctoral Fellowship. The U.S. Government is authorized to reproduce and distribute reprints for Governmental purposes notwithstanding any copyright notation thereon. Any opinions, findings, and conclusions or recommendations expressed in this material are those of the authors and do not necessarily reflect the views, policies, or endorsements, either expressed or implied, of NIH, ONR, or the U.S. Government.

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

## APPENDIX

## A EXPERIMENT DETAILS

We use A100 NVidia GPUs to run all experiments.

## A.1 MODELS

We evaluate over 4 model families: T0, BLOOM, Neo, OPT, and GPT3. In our evaluations, we use the following model family variants: Neo (GPT-Neo-125M, GPT-Neo-1.3B, GPT-Neo-6B, GPT-Neo-20B), BLOOM (BLOOM-560M, BLOOM-1.7B, BLOOM-7.1B, BLOOM-176B), OPT(OPT-125M, OPT-1.3B, OPT-6.7B, OPT-13B, OPT-175B), T0 (T0-3B), and GPT-3 (davinci). We download T0, BLOOM, OPT, and Neo models from the HuggingFace Model Hub (HuggingFace, 2021). All inference calls to the OpenAI Davinci model were made using the OpenAI API `davinci` endpoint (OpenAI, 2021), the original GPT-3 175B parameter model used in (Brown et al., 2020). We access these models by passing our input prompts to the endpoint for a per-sample fee.

## A.2 METRICS

For RealTimeQA, the reported GPT-3 performance in Kasai et al. (2022) is reported over the `text-davinci-002` API endpoint. Given that all our GPT-3 evaluations are over `davinci`, we re-evaluate the GPT-3 performance on RealTimeQA using the `davinci` endpoint and the few-shot prompt from RealTimeQA.[3]

We follow the metrics used in Brown et al. (2020). All tasks are scored using matching accuracy except for DROP/RealTimeQA that use text f1, WebQ/NQ that use span overlap accuracy, and MultiRC that uses f1a accuracy.

## A.3 WEAK SUPERVISION

For each task, we use an unlabeled dataset constructed from the test set as well as 1000 additional unlabeled samples from the training set. The additional data is used for less noisy parameter estimation, although in section B.5 we find that this data may not be necessary. We run the structure learning part of the weak supervision algorithm (for $\hat{G}$) with the default parameters from Varma et al. (2019). If the recovered sparse matrix has all entries greater than 1, we pass in an empty edgeset to the next step of learning $\hat{\theta}$ (e.g., data is too noisy to learn structure from); otherwise, we pass in the edge with the highest value in the sparse matrix.

# B ADDITIONAL RESULTS

## B.1 BLOOM MODEL RESULTS

In Table 2, we provide results using the BLOOM-7.1B parameter model over all 20 benchmarks. We observe consistent lift over few-shot performance using AMA, though the performance remains below that of the comparably sized GPT-J-6B parameter model reported in Table 5.1. We note that the few-shot results for BLOOM 7.1B are also often lower than the few-shot results for GPT-J-6B.

## B.2 LARGE MODEL RESULTS

In Table 3, we provide results using the large open-source language models including the 175B parameter BLOOM and OPT models. We observe that even at the large model scale, AMA provides performance improvements over the few-shot baseline.

## B.3 T0 COMPARISON

While we observe that T0 performs poorly on synthetic tasks correlated with the $\mathrm{prompt}()$-functions (see Figure 5), we find that aggregating over zero-shot instructions in Prompt-Source provides lift Bach et al. (2022). Specifically, when evaluating over 10 unique prompts for CB, WIC, WSC and RTE respectively, we find that aggregating with MV yields an average lift of 3.7 accuracy points and aggregating with WS gives an average lift of 6.15 accuracy points (see Table 5).

---

[3]https://github.com/realtimeqa/realtimeqa_public

| Model | BLOOM Few-Shot | BLOOM (QA) | BLOOM (QA + WS) | GPT-3 Few-Shot |
|---|---|---|---|---|
| # Params | 7.1B | 7.1B | 7.1B | 175B |
| Natural Language Understanding | | | | |
| BoolQ | $61.9_{(3)}$ | 66.0 | $67.9_{\pm0.0}$ | $\mathbf{77.5}_{(32)}$ |
| CB | $41.1_{(3)}$ | 67.3 | $77.6_{\pm0.9}$ | $\mathbf{82.1}_{(32)}$ |
| COPA | $65.0_{(3)}$ | 57.8 | $74.0_{\pm0.5}$ | $\mathbf{92.0}_{(32)}$ |
| MultiRC | $21.3_{(3)}$ | 56.9 | $59.7_{\pm0.0}$ | $\mathbf{74.8}_{(32)}$ |
| ReCoRD | $71.8_{(3)}$ | 70.0 | $69.8_{\pm0.0}$ | $\mathbf{89.0}_{(32)}$ |
| RTE | $53.1_{(3)}$ | 61.2 | $67.5_{\pm0.0}$ | $\mathbf{72.9}_{(32)}$ |
| StoryCloze | $49.4_{(3)}$ | 70.4 | $80.8_{\pm0.0}$ | $\mathbf{87.7}_{(70)}$ |
| WiC | $50.8_{(3)}$ | 57.8 | $\mathbf{60.0}_{\pm0.0}$ | $55.3_{(32)}$ |
| WSC | $61.5_{(3)}$ | 62.2 | $60.6_{\pm0.0}$ | $\mathbf{75.0}_{(32)}$ |
| Natural Language Inference | | | | |
| ANLI R1 | $34.9_{(3)}$ | 32.9 | $34.6_{\pm0.0}$ | $\mathbf{36.8}_{(50)}$ |
| ANLI R2 | $33.6_{(3)}$ | 34.1 | $34.6_{\pm0.0}$ | $\mathbf{34.0}_{(50)}$ |
| ANLI R3 | $32.3_{(3)}$ | 35.0 | $36.6_{\pm0.7}$ | $\mathbf{40.2}_{(50)}$ |
| Classification | | | | |
| AGNews | $68.3_{(3)}$ | 74.4 | $\mathbf{81.7}_{\pm0.3}$ | $79.1_{(8)}$ |
| Amazon | $58.1_{(3)}$ | 62.8 | $\mathbf{65.2}_{\pm0.0}$ | $41.9_{(8)}$ |
| DBPedia* | $72.3_{(3)}$ | 62.8 | $69.8_{\pm0.0}$ | $\mathbf{83.2}_{(8)}$ |
| SST2 | $56.4_{(3)}$ | 90.2 | $91.0_{\pm0.0}$ | $\mathbf{95.6}_{(8)}$ |
| Question Answering | | | | |
| DROP | $30.5_{(3)}$ | 67.9 | $\mathbf{67.9}_{\pm0.0}$ | $36.5_{(20)}$ |
| NQ | $12.1_{(3)}$ | 15.2 | $15.1_{\pm0.0}$ | $\mathbf{29.9}_{(64)}$ |
| RealTimeQA | $21.8_{(3)}$ | 27.7 | $29.0_{\pm0.0}$ | $\mathbf{35.4}_{(1)}$ |
| WebQs | $28.0_{(3)}$ | 34.8 | $34.8_{\pm0.0}$ | $\mathbf{41.5}_{(64)}$ |

Table 2: AMA results for the BLOOM-7.1B parameter model compared to the few-shot GPT3-175B. The GPT-175B numbers are as reported in Brown et al. (2020), where the numbers of shots is in parentheses, and the classification task baselines are from from Zhao et al. (2021). *: DBPedia results are on a randomly subsampled test set of 10k points.

| Model | BLOOM Few-Shot | BLOOM AMA | OPT Few-Shot | OPT AMA | Fine-Tuned |
|---|---|---|---|---|---|
| # Params | 175B | 175B | 175B | 175B | - |
| Natural Language Understanding | | | | | |
| CB | 64.3 | 82.1 | 64.3 | 80.1 | 96.9 |
| RTE | 57.4 | 66.5 | 53.8 | 67.7 | 92.5 |
| WSC | 68.3 | 71.8 | 49.0 | 76.0 | 93.8 |
| WiC | 51.7 | 59.4 | 51.9 | 58.4 | 76.1 |
| Natural Language Inference | | | | | |
| ANLI R1 | 37.3 | 37.4 | 37.9 | 37.9 | 73.8 |
| ANLI R2 | 34.4 | 36.1 | 37.2 | 37.2 | 50.7 |
| Classification | | | | | |
| SST | 45.6 | 95.2 | 55.6 | 95.8 | 97.5 |

Table 3: AMA results for the large open source models. These are the raw results corresponding to Figure 5a.

## B.4 ABLATING THE PROMPT REFORMATTING AND AGGREGATION COMPONENTS OF AMA

Here we study the degree to which *both* prompt re-formatting and aggregation are required to achieve high quality, extending the observations in Section 3. We produce 3 few-shot prompts that follow the same template by varying the $k = 3$ in-context prompt examples. In this process, the proposed AMA prompt reformatting is *not* applied. We apply each of the few-shot prompts and aggregate the results using majority vote MV and WS. We observe aggregation alone provides lift over the average prompt performance in Table 6. However the gap to AMA performance remains large. Aggregation and re-formatting are both critical and complementary to the end-to-end solution.

## B.5 WEAK SUPERVISION ABLATIONS

**Comparison to other aggregation baselines**  Table 7 compares AMA's aggregation method against several other baselines for aggregating prompt()-chains, including majority vote. We compare against weighted majority vote (WMV), where we use labeled data to weight according to each

| Model | T0 3B AMA | T0 PP AMA |
|-------|-----------|-----------|
| CB | 50.0 | 44.6 |
| RTE | 51.8 | 58.7 |
| WSC | 66.0 | 62.5 |
| WiC | 49.7 | 51.9 |
| SST | 47.4 | 47.2 |
| ANLI-R1 | 33.3 | 32.1 |
| ANLI-R2 | 32.9 | 33.6 |

Table 4: AMA results for T0 models.

| Task | CB | WIC | WSC | RTE |
|------|-----|-----|-----|-----|
| T0 (3B) Mean | 45.4 | 50.7 | 65.1 | 64.6 |
| T0 (3B) 10 Formats MV | 60.7 | 50.5 | 68.3 | 60.6 |
| T0 (3B) AMA MV | 50.0 | 49.5 | 64.4 | 49.5 |
| T0 (3B) 10 Formats WS | 60.7 | 50.8 | 69.2 | 69.7 |
| T0 (3B) AMA WS | 50.0 | 51.4 | 66.4 | 59.2 |

Table 5: Performance of T0 as reported in Sanh et al. (2022) compared to majority vote (MV) and weak supervision (WS) over 10 different prompt formats in Prompt-Source. When using the Prompt-Source prompts, the average lift across tasks is 3.6 points for MV and 6.1 points for WS.

prompt's accuracy by constructing $\phi_{\text{WMV}}(\mathbf{P}(x)) = \sum_{i=1}^{m} \exp(-\eta \varepsilon_i) \mathbb{1}\{p_i(x) = y\}$. $\varepsilon_i$ is the error of prompt $p_i$ on a training set of 1000 examples, and $\eta$ is a temperature hyperparameter, for which we perform a sweep over $[0.25, 0.5, 1, 2, 4, 8, 16, 32]$ using a 20% validation split. We also compare against a simple strategy of using the prompt that performs the best on the labeled set of data (Pick Best). Finally, AMA (no deps) is our method when we pass in an empty edgeset to the algorithm in Ratner et al. (2018).

**Varying amount of additional data**  We study the effect of varying the amount of additional unlabeled training data that is used in learning the probabilistic graphical model on $y, \mathbf{P}(x)$. On three tasks (RTE, WSC, and AGNews) averaged over 5 runs, we run AMA with $100\%, 50\%, 20\%, 10\%$, and $0\%$ of the additional dataset while still evaluating on the fixed test dataset. Figure 6 shows AMA's accuracy versus the amount of additional unlabeled data used. We find that even without any of the additional data, average accuracy does not decrease on WSC or AGNews, and only decreases by 0.4 points on RTE, still outperforming GPT3-175B few-shot. This suggests that the additional data is not necessary for AMA's performance.

**Latency of Weak Supervsion**  Over RTE, WSC, and AGNews, we find that WS (both learning the graphical model and aggregating outputs) takes an average of 13.0 seconds when dependencies are not modeled. When dependencies are modeled in RTE (as dependencies are ignored in WSC and AGNews because they both exhibit dense recovered structured matrices), the algorithm takes an average of 84.3 seconds to run. As a point of comparison, we include Table 8 which shows the time in seconds for running inference with the GPT-J-6B model on the same tasks. The latency introduced by running weak supervision is comparatively low.

### B.6 ADDITIONAL AMA BASELINES

Here we compare AMA to Self-Consistency Wang et al. (2022a), which is particularly relevant in that it also aggregates over multiple prompt outputs without requiring any additional supervised training. Self-Consistency builds on Chain-of-Thought prompting Wei et al. (2022c), which proposes to guide the LM to generate reasoning paths in addition to the final prediction. We use the exact prompts and overlapping benchmark tasks provided in the Appendix of Wang et al. (2022a), using GPT-J-6B and report the results in Table 9. For Self-Consistency, we use temperature based sampling as discussed in Wang et al. (2022a), using temperatures $\in \{0.0, 0.3, 0.5, 0.6, 0.7\}$.

Overall, we observe AMA outperforms Self-Consistency at this model scale. This agrees with the results in Wang et al. (2022a) and Wei et al. (2022c), which report limited performance improvements for small LMs (<10B).

| Model | GPT-J Few-Shot | GPT-J Few-Shot | GPT-J Few-Shot | GPT-J AMA |
|---|---|---|---|---|
| Aggregation | Average | Majority Vote | Weak Supervision | Weak Supervision |
| Natural Language Understanding | | | | |
| CB | 23.8 | 17.9 | 50.0 | 83.9 |
| RTE | 53.5 | 53.1 | 54.2 | 75.1 |
| WSC | 46.2 | 38.5 | 38.5 | 77.9 |
| COPA | 80.0 | 81.0 | 81.0 | 84.0 |
| Natural Language Inference | | | | |
| ANLI R1 | 33.4 | 33.5 | 33.5 | 37.8 |
| ANLI R2 | 33.2 | 32.9 | 32.2 | 37.9 |
| ANLI R3 | 35.4 | 36.5 | 34.6 | 40.2 |
| Classification | | | | |
| AGNews | 70.3 | 70.7 | 75.0 | 86.4 |
| Amazon | 61.9 | 62.4 | 62.5 | 68.2 |

Table 6: Results from applying prompt aggregation via Majority Vote and Weak Supervision to 3 random few-shot ($k = 3$) prompts. Here we apply *no* prompt reformatting to the proposed AMA QA template.

| | # Prompts | Avg | MV | WMV | Pick Best | AMA (no dep) | AMA (WS) |
|---|---|---|---|---|---|---|---|
| No labels: | | | ✓ | | | ✓ | ✓ |
| Natural Language Understanding | | | | | | | |
| WSC | 3 | 74.7 | 77.8 | 77.8 | 75.0 | $77.8_{\pm0.0}$ | $\mathbf{77.8}_{\pm0.0}$ |
| WiC | 5 | 59.0 | 61.3 | 60.9 | 60.0 | $60.8_{\pm0.0}$ | $\mathbf{61.3}_{\pm0.2}$ |
| RTE | 5 | 61.4 | 66.0 | 71.4 | 62.0 | $65.1_{\pm0.5}$ | $\mathbf{74.7}_{\pm0.0}$ |
| CB | 3 | 83.3 | 82.1 | 82.1 | 83.9 | $82.1_{\pm0.0}$ | $\mathbf{83.9}_{\pm0.0}$ |
| MultiRC | 3 | 58.8 | 63.8 | 63.4 | 63.4 | $63.7_{\pm0.0}$ | $\mathbf{63.8}_{\pm0.0}$ |
| BoolQ | 5 | 64.9 | 65.9 | 67.2 | $\mathbf{68.3}$ | $65.9_{\pm0.0}$ | $67.2_{\pm0.0}$ |
| COPA | 4 | 58.3 | $\mathbf{85.0}$ | 82.0 | 82.0 | $84.0_{\pm0.0}$ | $84.0_{\pm0.0}$ |
| Natural Language Inference | | | | | | | |
| ANLI R1 | 5 | 34.6 | 37.6 | 36.1 | 36.8 | $37.4_{\pm1.0}$ | $\mathbf{37.8}_{\pm0.2}$ |
| ANLI R2 | 5 | 35.4 | 36.3 | 36.0 | 36.0 | $\mathbf{38.7}_{\pm0.4}$ | $37.9_{\pm0.2}$ |
| ANLI R3 | 5 | 37.0 | 39.0 | 38.4 | 38.4 | $39.6_{\pm0.9}$ | $\mathbf{40.9}_{\pm0.5}$ |
| StoryCloze | 6 | 76.3 | $\mathbf{87.9}$ | 81.8 | 81.8 | $82.2_{\pm0.0}$ | $87.8_{\pm0.0}$ |
| Classification | | | | | | | |
| DBPedia | 3 | 81.4 | $\mathbf{84.1}$ | 83.9 | 82.2 | $83.9_{\pm0.0}$ | $83.9_{\pm0.0}$ |
| SST2 | 3 | 94.5 | 95.7 | 95.7 | 95.2 | $95.7_{\pm0.0}$ | $\mathbf{95.7}_{\pm0.0}$ |
| Amazon | 3 | 67.0 | 68.6 | 68.6 | 67.3 | $68.6_{\pm0.0}$ | $\mathbf{68.6}_{\pm0.0}$ |
| AGNews | 3 | 83.7 | $\mathbf{86.5}$ | 84.2 | 83.8 | $86.4_{\pm0.0}$ | $86.4_{\pm0.0}$ |

Table 7: AMA Aggregation method ablation for the GPT-Neo-6B parameter model, as well as the number of $\mathrm{prompt}()$-chains used for each task. For ReCoRD, and QA tasks (DROP, WebQs, RealTimeQA, NQ), we use 3 prompts each and use majority vote as our aggregation strategy reported in the (QA + WS) columns of Table 1 and Table 2.

| Task | Number of Examples | Total Inference Cost (seconds) |
|---|---|---|
| RTE | 277 | 8310 |
| WSC | 104 | 3141 |
| AGNews | 7600 | 53200 |

Table 8: Total inference cost in applying the AMA prompt chains to achieve the results in Table 5.1, using the GPT-J-6B model.

## C  CONDITIONAL ENTROPY ABLATIONS

In addition to the plots for the Neo models which we present in Section 4, we measure the conditional entropy metric across 4 BLOOM models (see Figure 7 (Left)) and for aggregation on the BLOOM 7.1B model (Right). We evaluate the $\{560\mathrm{M}, 1.3\mathrm{B}, 7.1\mathrm{B}, 175\mathrm{B}\}$ parameter BLOOM models on RTE and the $k$-shot points are each the average of 4 random seeds.

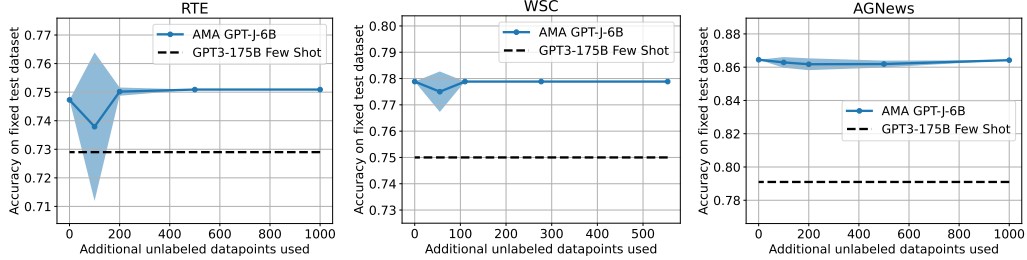

Figure 6: Performance on RTE, WSC, and AGNews averaged over 5 runs when using varying amounts of additional unlabeled training data for estimating $\Pr(y, \mathbf{P}(x))$ in WS.

| Baseline | Self-Consistency with Chain-of-Thought | Ask Me Anything |
|---|---|---|
| Aggregation over # Outputs | 5 | 5 |
| RTE | 47.3 | 75.1 |
| BoolQ | 63.1 | 67.2 |
| ANLI-R1 | 33.4 | 37.8 |

Table 9: Comparison between Self-Consistency Wang et al. (2022a) and AMA using GPT-J-6B and the same number of prompts.

# D  AMA PROCEDURE

## D.1  USER INPUTS TO AMA

To use AMA, the requirement from the user is to provide $\mathrm{question}()$ and $\mathrm{answer}()$ prompts. However, in this work, we provide many pre-written prompts for users to adopt off-the-shelf. We also note that many tasks reuse the exact same prompts (with the exact same in-context examples) to achieve the reported results. Because there is a large search space in natural language prompt design, we hope by providing users with the QA template, AMA helps reduce the required effort.

## D.2  STEPS

Here we summarize the AMA procedure that was introduced in Section 3. The end-to-end procedure is outlined in Algorithm 1. We further describe the prodedure line-by-line, using the following example input $x_i$, which contains a textual statement and passage, and requires determining if the statement is valid based on the information in the passage. Let $x_i$ include:

- Statement: John went to the park.
- Passage: John and his family went to the theater to watch Jurassic Park.

Following step 2 in Algorithm 1, a $\mathrm{question}()$ prompt is applied to the statement via the LM, to convert the statement to a question such as the yes-no question "Did John go to the park?" or the open-ended question, "Where did John go?". Example $\mathrm{question}()$ prompts, which contain generic in-context demonstrations of statements and question-versions of those statements, are listed in Appendix K.

Next, an $\mathrm{answer}()$ prompt is applied (step 3 in Algorithm 1). The $\mathrm{answer}()$ prompts are constructed by having the LM answer the previously generated question (e.g., "Did John go to the park?") using the passage in $x_i$. See Appendix K for example $\mathrm{answer}()$ prompts. Applying the prompt, the LM generates an answer $a_i$, for instance "No" to the yes-no question "Did John go to the park?", or "theater" to the open-ended question "Where did John go?".

Next, we discuss how the answer $a_i$ is mapped to a prediction $\hat{y}_i$ in the task output space, for input $x_i$. Note that there is no specialized mapping process in AMA— we simply score the exact match between the LM-generated $a_i$ and the gold label $y_i$. The approach is detailed below:

- **Yes-no questions** For yes-no questions, the ultimate answers are ultimately "yes" or "no" and we can also fit in-context examples of questions with each of these two answer choices in the $\mathrm{answer}()$ prompt.

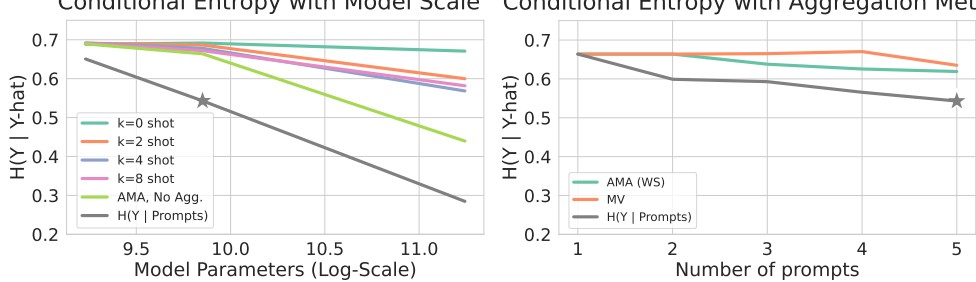

Figure 7: BLOOM model plots for the conditional entropy metric $H(y|\hat{y})$ as a function of model size $\in \{560M, 1.7B, 7.1B, 175B\}$ and prompts $p$ with $k = \{0, 2, 4, 8\}$ in-context examples (Left), and the aggregation strategy over the prompts (majority vote and weak supervision) with the 7.1B model (Right). Plots are over RTE benchmark, and each $k$-shot point is the average of 4 random seeds.

---

**Procedure 1:** AMA End-to-End Procedure

1: **Input:** Dataset $\mathcal{D} = \{x_i\}_{i=1}^n$, collection of $m$ prompt()-chains $\mathbf{P}$. A prompt()-chain contains a question() and an answer() prompt.
  **Output:** Predictions $\{\hat{y}_i\}_{i=1}^n$.
2: Given the input $x_i \in \mathcal{D}$, for each prompt()-chain $P_i \in \mathbf{P}$, prompt the LM with question() to produce a question about $x_i$. Across the $m$ prompt()-chains, AMA produces $m$ questions, $q_{i1}, ..., q_{im}$ for each input $x_i$.
3: For each input $x_i$, for each question $q_{ij}$, apply the answer() prompt to produce the LM prediction $a_{ij}$ for $x_i$. Across the $m$ prompt()-chains, $m$ answers $a_{i1}, ..., a_{im}$ are produced for each $x_i$.
4: Construct the weak-supervision based aggregation function (as specified in Algorithm 2), which takes the $m$ LM generated answers $a_{i1}, ..., a_{im}$ for the task inputs in $\mathcal{D}$, and returns a single prediction $\hat{y}_i$ for each $x_i$.
5: **Returns:** $\hat{y}_i$ for all $x_i \in \mathcal{D}$.

---

- **Open-ended questions, Open-ended output space** For open-ended questions, the answers are open-ended. To facilitate scoring the answers to open-ended questions, our approach is to, when the LM generates the question "Where did John go?" using the question() prompt, to *also* have the LM generate the hypothesized answer, i.e. "theater", *at the same time*. See the RTE task section in Appendix K for an example of such a prompt.

- **Open-ended questions, Restricted output space** Tasks such as classification require outputting a "restricted" label for the input $x_i$. For instance, is the article about "Sports" or "Technology" — answers such as "Athletics" are invalid. For such tasks, we include the viable outputs in the answer() prompt to encourage the LM to select from the list.

Finally, the LM generated answer $a_i$ is scored simply given an exact-match comparison to the gold label. In AMA, we aggregate over multiple answers per input. The different answers for the same input $x_i$ are obtained by applying different combinations of question() and answer() prompts, i.e. differing in their in-context demonstrations. In this work, we use 3-6 pairs of question() and answer() prompts per task. The weak-supervision based aggregation step is applied once multiple predictions are collected for all inputs in the task, to produce a single prediction per input. The aggregation algorithm is detailed in the next section.

## E   WEAK SUPERVISION ALGORITHM

We briefly explain the weak supervision algorithm used for constructing $\phi_{WS}$. Weak supervision models learn the latent variable graphical model on the distribution $\Pr(y, \mathbf{P}(x))$ using the dataset $\mathcal{D}$, and aggregate votes using the learned distribution by setting $\phi(x) = \arg\max_y \Pr(y|\mathbf{P}(x))$. Our key insight in our aggregation approach is to parametrize $\Pr(y, \mathbf{P}(x))$ so that we can capture variations in accuracy as well as dependencies if they exist. The overall procedure of our aggregation is in Algorithm 2. Formally, we model $\Pr(y, \mathbf{P}(x))$ as a probabilistic graphical model with dependency graph $G = (V, E)$, where $V = \{y, \mathbf{P}(x)\}$. If $p_i(x)$ and $p_j(x)$ are not conditionally independent

---

**Procedure 2:** AMA Aggregation Method

1: **Input:** Dataset $\mathcal{D} = \{x_i\}_{i=1}^n$, collection of $\mathrm{prompt}()$-chains $\mathbf{P}$. **Output:** Predictions $\{\hat{y}_i\}_{i=1}^n$.
2: Prompt the LLM with $\mathbf{P}$ to produce $m$ predictions $\mathbf{P}(x)$ per input $x \in \mathcal{D}$, constructing dataset $\mathcal{D}_{\mathbf{P}} \in \mathbb{R}^{n \times m}$.
3: Learn $\hat{G} = (V, \hat{E})$ via structure learning on $\mathcal{D}_{\mathbf{P}}$ (Algorithm 1 in Varma et al. (2019)).
4: Learn $\mathrm{Pr}_{\hat{G},\hat{\theta}}(y, \mathbf{P}(x))$ using $\mathcal{D}_{\mathbf{P}}$ and $\hat{G}$ (Algorithm 1 in Ratner et al. (2018)).
5: Construct aggregator $\phi_{\mathrm{WS}}(\mathbf{P}(x)) = \arg\max_{y \in \mathcal{Y}} \mathrm{Pr}_{\hat{G},\hat{\theta}}(y|\mathbf{P}(x))$.
6: **Returns:** $\hat{y}_{\mathrm{AMA}} = \phi_{\mathrm{WS}}(x)$ for all $x \in \mathcal{D}$.

---

given $y$ and the other $\mathrm{prompt}()$-chains, then $(p_i(x), p_j(x)) \in E$. $E$ also contains edges $(p_i(x), y)$ for each $i \in [m]$.

The algorithm uses $\mathbf{P}(x)$ and $\mathcal{D}$ to first learn the dependency structure $\hat{G}$ among prompts using the approach from Varma et al. (2019). The key insight from that work is that the inverse covariance matrix $\Sigma^{-1}$ over $y$ and $\mathbf{P}(x)$ is graph-structured, meaning that $\Sigma_{ij}^{-1} = 0$ iff $p_i(x)$ and $p_j(x)$ are conditionally independent given $y$. The graph structure means that the inverse covariance over just $\mathbf{P}(x)$ decomposes into sparse and low-rank matrices, which can hence be estimated together using RobustPCA (Candès et al., 2011), and the sparse matrix can be used to recover the graph. Next, the algorithm uses the recovered $\hat{G}$ along with $\mathbf{P}(x)$ and $\mathcal{D}$ to learn the accuracies of the prompts with the approach from Ratner et al. (2018). The key insight from that work is to use the sparsity of $\Sigma^{-1}$ to construct a system of equations set equal to $0$ that recover the latent accuracy parameters. Once the parameters of the distribution are learned, we can compute $\mathrm{Pr}_{\hat{G},\hat{\theta}}(y|\mathbf{P}(x))$ and aggregate our predictions.

## F    INFORMATION-FLOW THEORETICAL RESULT

In equation 1, we decompose $H(y|\hat{y})$ into $H(y|\mathbf{P}(x))$ and $H(y|\hat{y}) - H(y|\mathbf{P}(x))$. For AMA, suppose that the weak supervision algorithm exactly recovers $\mathrm{Pr}(y, \mathbf{P}(x))$. That is, $\hat{y}_{\mathrm{AMA}}$ is drawn from $\mathrm{Pr}(\cdot|\mathbf{P}(x))$. Then, the second term $H(y|\hat{y}) - H(y|\mathbf{P}(x))$ can be thought of as an irreducible error corresponding to how much information about $y$ is lost in converting $\mathbf{P}(x)$ into an i.i.d. $y'$ randomly drawn from $\mathrm{Pr}(\cdot|\mathbf{P}(x))$. Since $y'$ is more likely to change values when this distribution has high entropy, the second term is correlated with our first term $H(y|\mathbf{P}(x))$, the amount of randomness in $\mathrm{Pr}(y|\mathbf{P}(x))$. We thus focus on obtaining an expression for $H(y|\mathbf{P}(x))$ in terms of individual prompt accuracies.

We assume that $\mathcal{Y} = \{-1, 1\}$. We model $\mathrm{Pr}(y, \mathbf{P}(x))$ as a probabilistic graphical model with dependency graph $G = (V, E)$, where $V = \{y, \mathbf{P}(x)\}$. The density of $\mathrm{Pr}(y, \mathbf{P}(x))$ follows the following Ising model commonly used in weak supervision (Ratner et al., 2017; Fu et al., 2020):

$$\Pr_{G,\theta}(y, \mathbf{P}(x)) = \frac{1}{Z} \exp\left( \theta_y y + \sum_{i=1}^m \theta_i p_i(x) y + \sum_{(i,j) \in E} \theta_{ij} p_i(x) p_j(x) \right), \tag{2}$$

where $Z$ is the partition function for normalization and $\{\theta_y, \theta_i \ \forall \ i \in [m], \theta_{ij} \ \forall \ (i, j) \in E\}$. Each $\theta_i$ can be viewed as the strength of the correlation between $y$ and $p_i(x)$, while each $\theta_{ij}$ can be viewed as the strength of the dependence between $p_i(x)$ and $p_j(x)$. We assume that $\theta_y = 0$, which corresponds to $\mathrm{Pr}(y = 1) = \frac{1}{2}$.

We present our expression for $H(y|\mathbf{P}(x))$. Define $\Theta = [\theta_1, \ldots, \theta_m]$ to be the vector of canonical parameters corresponding to the strength of correlation between $y$ and each $p_i(x)$. Define $\mu = \mathbb{E}[p_i(x)]$, which can be written as $2\mathrm{Pr}(p_i(x) = y) - 1$, a notion of accuracy scaled to $[-1, 1]$.

Note that the above form of the distribution is in terms of canonical parameters $\theta$. This distribution can also be parametrized in terms of the mean parameters corresponding to $\theta$, which are $\mathbb{E}[y], \mathbb{E}[p_i(x)y]$ for $i \in [m]$, and $\mathbb{E}[p_i(x)p_j(x)]$ for $(p_i(x), p_j(x)) \in E$.

**Theorem 1.** *Assume* $\Pr(y, \mathbf{P}(x))$ *follows equation* 2 *above. Then, the conditional entropy* $H(y|\mathbf{P}(x))$ *can be expressed as*

$$H(y|\mathbf{P}(x)) = H(y) - \left( \Theta^\top \mu - \mathbb{E}_{\mathbf{P}(x)} \left[ \log \cosh \Theta^\top \mathbf{P}(x) \right] \right) \qquad (3)$$

The quantity being subtracted from $H(y)$ corresponds to the reduction in entropy of $y$ given that we observe $\mathbf{P}(x)$. Within this expression, there are two terms. First, $\Theta^\top \mu$ is correlated with how much signal each $p_i(x)$ contains about $y$. Note that this quantity is symmetric—if $p_i(x)$ is negatively correlated with $y$, it still provides information since both $\theta_i$ and $\mathbb{E}\left[ p_i(x)y \right]$ will be negative. The second term, $\mathbb{E}_{\mathbf{P}(x)} \left[ \log \cosh \Theta^\top \mathbf{P}(x) \right]$, is for normalization (otherwise, the first term can grow arbitrarily large with $\Theta$). Note that this quantity is independent of $\theta_{ij}$, the interactions between prompts.

*Proof.* We can write $H(y|\mathbf{P}(x))$ as $H(y, \mathbf{P}(x)) - H(\mathbf{P}(x))$, and $H(y, \mathbf{P}(x))$ as $H(\mathbf{P}(x)|y) + H(y)$. Therefore, $H(y|\mathbf{P}(x)) = H(y) - \left( H(\mathbf{P}(x)) - H(\mathbf{P}(x)|y) \right)$. We focus on simplifying $H(\mathbf{P}(x)) - H(\mathbf{P}(x)|y)$:

$$H(\mathbf{P}(x)) - H(\mathbf{P}(x)|y) = -\sum_{\mathbf{P}(x)\in\{-1,1\}^m} \Pr(\mathbf{P}(x)) \log \Pr(\mathbf{P}(x)) + \sum_{\mathbf{P}(x)\in\{-1,1\}^m,y} \Pr(y, \mathbf{P}(x)) \log \Pr(\mathbf{P}(x)|y) \quad (4)$$

$$= -\sum_{\mathbf{P}(x)\in\{-1,1\}^m,y} \Pr(\mathbf{P}(x), y)\Big( \log \Pr(\mathbf{P}(x)) - \log \Pr(\mathbf{P}(x)|y) \Big)$$

$$= -\sum_{\mathbf{P}(x)\in\{-1,1\}^m} \Big( \Pr(\mathbf{P}(x), y = -1)\Big( \log \Pr(\mathbf{P}(x)) - \log \Pr(\mathbf{P}(x)|y = -1) \Big)$$

$$+ \Pr(\mathbf{P}(x), y = 1)\Big( \log \Pr(\mathbf{P}(x)) - \log \Pr(\mathbf{P}(x)|y = 1) \Big) \Big).$$

We now write $\Pr(\mathbf{P}(x))$, $\Pr(\mathbf{P}(x)|y = -1)$ and $\Pr(\mathbf{P}(x)|y = 1)$ according to our Ising model in equation 2. Let $A_{\mathbf{P}(x)} = \sum_{i=1}^m \theta_i p_i(x)$, and let $B_{\mathbf{P}(x)} = \sum_{(i,j)\in E} \theta_{ij} p_i(x) p_j(x)$, so that $\Pr(y, \mathbf{P}(x)) = \frac{1}{Z} \exp(A_{\mathbf{P}(x)}y + B_{\mathbf{P}(x)})$:

$$\Pr(\mathbf{P}(x)) = \Pr(\mathbf{P}(x), y = -1) + \Pr(\mathbf{P}(x), y = 1)$$

$$= \frac{1}{Z} \exp(A_{\mathbf{P}(x)} + B_{\mathbf{P}(x)}) + \frac{1}{Z} \exp(-A_{\mathbf{P}(x)} + B_{\mathbf{P}(x)}))$$

$$= \frac{1}{Z} \exp(B_{\mathbf{P}(x)})\big( \exp(A_{\mathbf{P}(x)}) + \exp(-A_{\mathbf{P}(x)}) \big)$$

$$\Pr(\mathbf{P}(x)|y = -1) = 2 \Pr(\mathbf{P}(x), y = -1) = \frac{2}{Z} \exp(-A_{\mathbf{P}(x)} + B_{\mathbf{P}(x)}))$$

$$\Pr(\mathbf{P}(x)|y = 1) = 2 \Pr(\mathbf{P}(x), y = 1) = \frac{2}{Z} \exp(A_{\mathbf{P}(x)} + B_{\mathbf{P}(x)}))$$

Therefore, we have that

$$\log \Pr(\mathbf{P}(x)) - \log \Pr(\mathbf{P}(x)|y = -1) = -\log Z + B_{\mathbf{P}(x)} + \log\big( \exp(A_{\mathbf{P}(x)}) + \exp(-A_{\mathbf{P}(x)}) \big)$$

$$- \log 2 + \log Z + A_{\mathbf{P}(x)} - B_{\mathbf{P}(x)} = -\log 2 + A_{\mathbf{P}(x)} + \log\big( \exp(A_{\mathbf{P}(x)}) + \exp(-A_{\mathbf{P}(x)}) \big)$$

$$\log \Pr(\mathbf{P}(x)) - \log \Pr(\mathbf{P}(x)|y = 1) = -\log Z + B_{\mathbf{P}(x)} + \log\big( \exp(A_{\mathbf{P}(x)}) + \exp(-A_{\mathbf{P}(x)}) \big)$$

$$- \log 2 + \log Z - A_{\mathbf{P}(x)} - B_{\mathbf{P}(x)} = -\log 2 - A_{\mathbf{P}(x)} + \log\big( \exp(A_{\mathbf{P}(x)}) + \exp(-A_{\mathbf{P}(x)}) \big)$$

Plugging this back into equation 4, we have

$$\sum_{\mathbf{P}(x)\in\{-1,1\}^m,y} \Pr(\mathbf{P}(x), y)A_{\mathbf{P}(x)}y - \Pr(\mathbf{P}(x))\Big( \log\big( \exp(A_{\mathbf{P}(x)}) + \exp(-A_{\mathbf{P}(x)}) \big) - \log 2 \Big)$$

$$= \sum_{\mathbf{P}(x)\in\{-1,1\}^m,y} \Pr(\mathbf{P}(x), y)A_{\mathbf{P}(x)}y - \Pr(\mathbf{P}(x)) \log \cosh A_{\mathbf{P}(x)}$$

$$= \mathbb{E}\left[ A_{\mathbf{P}(x)}y \right] - \mathbb{E}\left[ \log \cosh A_{\mathbf{P}(x)} \right].$$

Substituting in our definitions of $\Theta$ and $\mu$ give us our desired expression for $H(y|\mathbf{P}(x))$.  □

## G  AMA DIAGNOSTICS

We present a suite of $8$ diagnostic tasks, which can be categorized into four task types: question generation, answer generation, answer selection and extraction. We provided details about the tasks and scoring below.

**Question Generation**: We measure the ability of the model to transform a statement to a question. We construct 3 question generation tasks which evaluate the models ability to transform a statement to a yes/no question (see Question Generation (Yes/No)), transform a statement to a *wh-* question (see Question Generation (wh-)) and finally, transform statement about a placeholder entity to a question about the placeholder (see Question Generation (@placeholder)). All question generation tasks are scored using the ROUGE score (Lin, 2004).

---

**Question Generation (Yes/No)**

Input

```
Rewrite the statement as a yes/no question.

Statement: The father and son went camping to California.
Question:
```

Output

```
Did the father and son go camping?
```

---

**Question Generation (wh-)**

Input

```
Convert statement to a question.

Statement: Aristide kills Prime Minister Robert Malval
Question:
```

Output

```
Who killed Prime Minister Robert Malval?
```

---

**Question Generation (@placeholder)**

Input

```
Rewrite the statement as a question about the [at]placeholder.

Statement: Most of the light comes from the [at]placeholder
Question:
```

Output

```
Where does most of the light come from?
```

---

**Answer Selection**: We construct 2 answer selection tasks which measure the model's ability to generate an answer that is faithful to a set of provided answer choices. Concretely, we measure the models ability to select object categories from a fixed set of options specified in the context (see Answer Selection (category)). Further, we measure the model's ability to complete a sentence when provided with a context and set of sentence completion candidates (see Answer Selection (completion)). In both tasks, an answer is marked as correct if the generated response is one of the candidates provided in the context.

---

**Answer Selection (category)**

Input

```
Select the correct category.

"Categories":
- company
- educational institution
- artist
- athlete
- office holder
- mean of transportation
- building
- natural place
- village
- animal
- plant
- album
- film
- written work

Example: A "journal" fits "Category":
```

Output

```
written work
```

---

**Answer Selection (sentence)**

Input

```
Select one choice from the passage.

Select One Choice:
1. consumer electronics
2. Play Stations
3. cameras

Passage: Microsoft Corporation produces computer software, consumer electronics, and
    personal computers. It is headquartered at the Microsoft Redmond campus located
    in Redmond, Washington, United States.

The passage "Passage" states: Microsoft Corporation sells: "Choice":.
```

Output

```
consumer electronics
```

---

**Answer Generation**: We construct 1 answer generation task which measures the model's ability to generate candidate sentence completions given a context and portion of a statement (see Answer Generation). Here, a generated answer is marked as correct if the model generates 2 candidate answers.

---

**Answer Generation**

Input

```
Output a list of unique alternatives for each example.

Example: Barrack Obama believes the:

List alternatives:
- best novel is Harry Potter
```

---

Output

```
- worst book is Harry Potter
- United States is great
```

**Extraction**: We construct 2 extraction tasks which evaluate the ability of the model to extract spans from a given context. The first, and easier task, tests the model's ability to extract an attribute value from a wikibio (see Extraction (Span)). The second, more difficult task, tests the model's ability to extract the sentence from the context that mentions a specified entity (see Extraction (Sentence)). For both tasks, we use the Text-F1 score introduced in SQuAD (Rajpurkar et al., 2018).

---

**Extraction (Span)**

Input

```
Wiki Bio:
name: robert king
article_title: robert king (photojournalist)
birth_place: memphis , tn , usa
occupation: war correspondent photojournalist filmmaker creative director art director
     birthname: robert whitfield king
birth_date: may 25th

Question: What is the birthname?
Answer:
```

Output

```
robert whitfield king
```

---

**Extraction (Sentence)**

Input

```
Context: Caracas, Venezuela (CNN) -- It's been more than 180 years since Venezuelans
    saw Simon Bolivar's face. But the revolutionary leader's thick sideburns, bushy
    eyebrows and steely gaze popped out from behind picture frames Tuesday in new 3-D
     images unveiled by President Hugo Chavez. Researchers used several software
    programs to reconstruct the face of the man who liberated Bolivia, Colombia,
    Ecuador, Panama, Peru and Venezuela from the Spanish crown. Scans of Bolivar's
    skeletal remains, which investigators exhumed two years ago, factored into their
    calculations. So did historical paintings, photos of restored uniforms Bolivar
    wore and images of middle-aged Venezuelans, officials said.

Extract the sentence containing "Simon Bolivar":
```

Output

```
Caracas, Venezuela (CNN) -- It's been more than 180 years since Venezuelans saw Simon
    Bolivar's face.
```

---

# H  UNDERSTANDING THE EFFECTIVENESS OF THE QUESTION-ANSWERING TEMPLATE

We analyze the LM pretraining corpus to better understand why the proposed QA prompt template may be effective. The EleutherAI models are trained on The Pile corpus Black et al. (2021); Wang & Komatsuzaki (2021); Gao et al. (2021).

**Prompt patterns**  We compute the frequency of regular expression matches that correspond to the restrictive prompts (i.e., which instruct the model to output "True or False", "Yes or No") versus open-ended questions (i.e., which ask the model "Is . . . ?", "Who . . . ?") in a 2% random sample of

the ˜200B token Pile corpus. The restrictive prompt-patterns appear frequently in the original GPT-3 prompts Brown et al. (2020). The frequencies are in Table 10.

We observe that question patterns appear more frequently than the restrictive prompts. Further, we find several instances of yes-no questions followed by "yes" or "no", which mimics the AMA format (Table 11). Overall, we find that QA structured text appears much more frequently in the pretraining corpus, which may help explain why the language models perform better on QA.

| Category | Regular Expressions | Count |
|---|---|---|
| Restrictive Patterns | ".* true or false\?",
".* true or false\.",
".* true, false, or neither\?",
".* true, false, or neither\.",

".* yes or no\?",
".* yes or no\.",
".* yes, no, or maybe\?"
".* yes, no, or maybe\."

".* correct or incorrect\?"
".* correct or incorrect\. "
".* correct, incorrect or inconclusive\?"
".* correct, incorrect or inconclusive\."

"choose between:"
"pick one from:" | 222 |
| Yes-No
Question Patterns | "Is .*\?"
"Was .*\?"
"Did .*\?"
"Do .*\?"
"Are .*\?"
"Will .*\?" | 378379 |
| Open-Ended
Question Patterns | "When .*\?",
"Where .*\?",
"Why .*\?",
"Who .*\?",
"What .*\?",
"How many .*\?" | 639573 |

Table 10: Frequency of each category of regular expressions in the Pile sample.

| Category | Count |
|---|---|
| Yes/No Question & Answer Pattern | "is .*\? yes": 536, "was .*\? yes": 248, "did .*\? yes": 109, "do .*\? yes": 210, "are .*\? yes": 233, "were .*\? yes": 91, "will .*\? yes": 121, "is .*\? no": 2356, "was .*\? no": 983, "did .*\? no": 534, "do .*\? no": 935, "are .*\? no": 978, "were .*\? no": 422, "will .*\? no": 423 |

Table 11: Yes/No question patterns followed by "Yes" or "No" tokens.

**Word frequencies** When applying the few-shot restrictive prompts, we observe large imbalances in the F1-scores for different classes (Table 12). Therefore, we next ask if answering the restrictive prompts is challenging due to biases acquired during pretraining. Over the same Pile sample as before, the mean word count is $25.3 \pm 7309$ occurrences. We compute the frequency of individual words in the "restrictive" and "open-ended question" patterns from Table 10. This leads to two hypotheses about why QA prompts perform well:

1. First we see that there are imbalances between the occurrence of "yes" vs. "no", and "true" vs. "neither" for instance. This may bias the model towards certain answer choices. Indeed

Zhao et al. (2021) also hypothesizes, but does not provide any analysis over the pretraining corpus, that pretraining may instill particular biases in the model.

2. The frequency of the words in the "question words" categories is typically an order of magnitude larger than those in the "restrictive words" category. We hypothesize that the representations for the "question words" will be the most context-specific, which is useful for the prompting tasks we consider. Findings in Ethayarajh (2019) support this hypothesis — Ethayarajh (2019) finds that frequently occurring words (e.g. stop-words) have the most context-specific representations. In other words, for the more frequently occurring stop-words the embedding produced by the transformer-based LM changes more significantly depending on the co-occurring words in the context.

| Benchmark | Output Space | F1-Score 0-shot | F1-Score Few-shot *two in-context examples per class* | F1-Score AMA QA *single prompt-chain with no aggregation* |
|---|---|---|---|---|
| CB | True, False, Neither | True: 36.8 False: 0.0 Neither: 21.7 | True: 55.6 False: 0.0 Neither: 12.5 | True: 95.7 False: 92.3 Neither: 28.6 |
| RTE | True, False | True: 40.4 False: 58.3 | True: 70.6 False: 31.3 | True: 58.5 False: 64.9 |
| WSC | Yes, No | Yes: 53.5 No: 0.0 | Yes: 53.5 No: 13.7 | Yes: 61.3 No: 78.2 |

Table 12: F1-Score by class for three benchmarks with three different prompting templates each: 1) 0-shot, 2) few-shot with the original GPT-3 restrictive prompts Brown et al. (2020), and 3) AMA prompts. We observe large imbalances in the scores across classes under the 0-shot and few-shot prompting.

| Category | Word Counts |
|---|---|
| Restrictive Prompt Words | true: 69658 false: 41961 neither: 20891 

 yes: 12391 no: 452042 maybe: 36569 |
| Yes-No Question Prompt Words | Is: 3580578 Was: 1926273 Did: 200659 Do: 394140 Are: 1441487 Will: 619490 |
| Open-Ended Question Prompt Words | When: 583237 Where: 303074 Why: 97324 Who: 417798 What: 548896 How: 298140 |

Table 13: Frequency of each category of regular expressions in the Pile sample.

Overall, designing prompting templates for an LM based on analysis of the LM pretraining corpus may be a promising path forward for future work.

## I ERROR ANALYSIS

We bucket the common error modes of AMA into three categories: knowledge, instruction-following, and long-context.

**Knowledge errors**. We find that AMA yields the most gains when the knowledge required to complete the task is explicitly provided in the context (e.g., reading comprehension, extractive QA). We find that AMA provides comparatively less lift on tasks where the model needs to (1) recall encoded factual knowledge or (2) apply common-sense / real-world knowledge to a given context. We provide concrete examples from the NaturalQuestions dataset (see Knowledge (Factual) below) in which the GPT-Neo-6B model wrongly answers the question due to a lack of latent factual knowledge. We additionally provide case-examples from the BoolQ dataset where the model's limited real-world knowledge limits its ability to correctly answer the questions where the model's failure to recognize that food that is smoked is cooked, leads it to incorrectly answer the question (see Knowledge (Commonsense) below).

---

**Knowledge (Factual)**

Input

```
Question: what's the dog's name on tom and jerry
Answer:
```

Prediction

```
The dog's name is "Fido"
```

Ground Truth

```
Spike
```

---

**Knowledge (Commonsense)**

Input

```
Passage: A Philadelphia roll is a makizushi (also classified as a kawarizushi) type of
    sushi generally made with smoked salmon, cream cheese, and cucumber. It can also
    include other ingredients, such as other types of fish, avocado, scallions, and
    sesame seed.
Question: is the salmon cooked in a philadelphia roll
Answer:
```

Prediction

```
false
```

Ground Truth

```
true
```

---

**Instruction-following errors**. We find that on tasks with more restrictive output spaces (e.g., multiway classification tasks), a common failure mode is to generate an answer that is not in the desired output space of the AMA prompt, despite being *explicitly* prompted to do so. In Listing 3 and 4, we provide sample instances from the DBPedia classification task where GPT-Neo-6B does not correctly map a descriptive adjective (e.g., automobile or singer) to a valid class specified in the prompt.

---

**Instruction Following (1)**

Input

---

```
Pick one category for the following text.

"Categories":
- company
- educational institution
- artist
- athlete
- office holder
- mean of transportation
- building
- natural place
- village
- animal
- plant
- album
- film
- written work

Passage: Monteverdi High Speed -  The Monteverdi High Speed was a grand tourer
    automobile built by Monteverdi in Basel Switzerland from 1967 to 1970.
    Contemporary rivals included the British Jensen Interceptor (which was also
    powered by a Chrysler V8).This car was designed by the Italian design house Frua
    and was actually built by Fissore of Italy from 1969. They redesigned the car in
    1972 and again in 1975.The convertible version of the High Speed 375 was known as
     the Palm Beach.

Summary: This passage is about a automobile.

The summary "Summary" fits "Category":
```

### Prediction

```
automobile
```

### Ground Truth

```
mean of transportation
```

---

**Instruction Following (2)**

### Input

```
Pick one category for the following text.

"Categories":
- company
- educational institution
- artist
- athlete
- office holder
- mean of transportation
- building
- natural place
- village
- animal
- plant
- album
- film
- written work

Passage: Passage: Patricia Bennett -  Patricia Bennett (born 7 April 1947 in The Bronx
     New York) was an original member of the American singing girl group the Chiffons
     .

Summary: This passage is about a singer.

The summary "Summary" fits "Category":
```

Prediction

```
singer
```

Ground Truth

```
artist
```

**Long-context errors**. We find that the AMA question() functions struggle to generate accurate statement-question transformations when the input is long or contains complex sentence structures (e.g. compound sentences). We provide sample instances from the SuperGLUE record task where GPT-Neo-6B fails to transform a sentence with a placeholder subject to a question about the placeholder subject (see Long-context (question()) below). Additionally, we find that the AMA answer() functions struggle to extract the correct span in long contexts (greater than 6 sentences). We show a sample instance from the DROP QA task where GPT-Neo-6B fails to extract the correct span from the long provided context (see Long-context (answer()) below).

---

**Long-context (question())**

Input

```
Rewrite the statement as a question about the [at]placeholder.

Statement: Most of the light comes from the [at]placeholder
Question: Where does most of the light come from?

Statement: The [at]placeholder was not hard
Question: What was not hard?

Statement: [at]placeholder went to the mall with her mom to buy a backpack
Question: Who went to the mall with her mom to buy a backpack?

Statement: Rossello warned the public on Sunday that the island could feel [at]
    placeholder's wrath around noon Wednesday.
Question:
```

Prediction

```
Who warned the public on Sunday that the island could feel [at]placeholder's wrath
    around noon Wednesday?
```

Ground Truth

```
Who's wrath could be felt around noon on Wednesday?
```

---

**Long-context (answer())**

Input

---

```
Context:  Looking to avoid back-to-back divisional losses, the Patriots traveled to
    Miami to face the 6-4 Dolphins at Dolphin Stadium. After Carpenter's kickoff was
    returned from the 29-yard line by Matthew Slater, the Patriots began their first
    possession at their own 40-yard line. Cassel's first two passes were both
    completed for first downs, putting the Patriots in Dolphins territory and
    eventually their red zone. However, a holding penalty on Neal pushed the Patriots
     back 10 yards, forcing a 30-yard Gostkowski field goal four plays later that
    gave the Patriots a 3-0 lead. Following a Dolphins three-and-out, the Patriots'
    second drive ended when a Cassel pass to Moss was bobbled by both Moss and
    cornerback Jason Allen to keep the ball in the air until Renaldo Hill intercepted
     it; a 17-yard return gave the Dolphins the ball at the Patriots' 42-yard line.
    On the next play, a 29-yard David Martin reception moved the Dolphins into the
    Patriots' red zone, where the Dolphins used their "Wildcat" formation on the next
     two plays
    ....

Question: Which team scored first?
```

Prediction
```
Patriots
```

Ground Truth
```
Dolphins
```

## J    AMA PSEUDOCODE

## K    DATASETS AND PROMPTS

We evaluate over 20 datasets which fall into 4 categories: SuperGLUE (BoolQ (Clark et al., 2019),
CB (De Marneffe et al., 2019), COPA (Roemmele et al., 2011), MultiRC (Khashabi et al., 2018),
ReCoRD (Zhang et al., 2018), RTE (Wang et al., 2019), WiC (Pilehvar & Camacho-Collados,
2018), WSC (Levesque et al., 2012)), NLI (ANLI R1, ANLI R2, ANLI R3 (Nie et al., 2020), Sto-
ryCloze (Mostafazadeh et al., 2017)), Classification (DBPedia (Zhang et al., 2015), AGNews (Zhang
et al., 2015), SST2 (Socher et al., 2013), Amazon (He & McAuley, 2016)), and Question-Answering
(RealTimeQA (Kasai et al., 2022), DROP (Dua et al., 2019), NaturalQuestion (Kwiatkowski et al.,
2019), WebQuestions (Berant et al., 2013)). We provide dataset details along with few shot and
AMA prompts for the dataset below. The purple highlighted part is the input example. The rest of
the prompt is fixed for all examples in the dataset.

## K.1 AGNEWS

*Description:* News article classification dataset with 4 topics. Zhang et al. (2015)
*Train Size:* 120000, *Test Size:* 76000

---

**AGNews Few Shot**

Input

```
Pick the correct category for the passage.

Categories:
- World News
- Sports
- Business
- Technology and Science

Passage: Wedding cad comes clean over invite sale (Reuters). Reuters - A wedding guest
    who sparked a bidding frenzy when he offered for sale a pair of invitations to a
    wedding he did not want to attend has admitted that the bride was a former girl
    friend.
Category: World News

Passage: Tennis: Serena Williams Reaches Finals of China Open. Top seed Serena
    Williams of the United States has powered her way into the finals of the China
    Open tennis tournament in Beijing with a straight sets (6-2, 6-3) victory over
    fourth-seeded Vera Zvonareva of Russia.
Category: Sports

Passage: Abramovich faces rich list challenge. Lakshmi Mittal, the Indian-born steel
    magnate, yesterday staked a claim to overtake Roman Abramovich as Britain's
    richest man with a 10bn deal to create the world's largest steelmaker.
Category: Business

Passage: The Race is On: Second Private Team Sets Launch Date for Human Spaceflight (
    SPACE.com). SPACE.com - TORONTO, Canada -- A second team of rocketeers competing
    for the #36;10 million Ansari X Prize, a contest for privately funded suborbital
    space flight, has officially announced the first launch date for its manned
    rocket.
Category:
```

Gold Output

```
Technology and Science
```

---

**AGNews AMA** prompt()**-chain Example**

question()

```
Summarize the passage.

Passage: China overtakes United States as top destination for foreign investment (AFP)
    . AFP - China overtook the United States as a top global destination for foreign
    direct investment (FDI) in 2003 while the Asia-Pacific region attracted more
    investment than any other developing region, a UN report said.
Summarize: the passage "Passage": The passage is about foreign direct investment.

Passage: Colangelo resigns as CEO of D-Backs. Jerry Colangelo has resigned his
    position as chief executive officer of the Arizona Diamondbacks, effective
    immediately, handing the reins of the organization to CEO Elect Jeff Moorad.
Summarize: the passage "Passage": The passage is about the Arizona Diamondbacks.

Passage: 3 injured in plant fire in Japan. TOKYO, Aug. 20 (Xinhuanet) -- Fire broke
    out Friday at a tire plant belonging to Bridgestone Corp. in Amagi, western
    Fukuoka Prefecture of Japan, leaving 13 people injured.
Summarize: the passage "Passage": The passage is about a plant fire.

Passage: The Race is On: Second Private Team Sets Launch Date for Human Spaceflight (
    SPACE.com). SPACE.com - TORONTO, Canada -- A second team of rocketeers competing
    for the #36;10 million Ansari X Prize, a contest for privately funded suborbital
    space flight, has officially announced the first launch date for its manned
    rocket.
Summarize: the passage "Passage":
```

## Model Output

```
The passage is about a rocket.
```

## answer()

```
Pick the correct category for the passage.

"Categories":
- World News
- Sports
- Business
- Technology and Science

Passage: China overtakes United States as top destination for foreign investment (AFP)
    . AFP - China overtook the United States as a top global destination for foreign
    direct investment (FDI) in 2003 while the Asia-Pacific region attracted more
    investment than any other developing region, a UN report said.
Summary: The passage is about foreign direct investment.
The summary "Summary" fits "Category": Business

Passage: Colangelo resigns as CEO of D-Backs. Jerry Colangelo has resigned his
    position as chief executive officer of the Arizona Diamondbacks, effective
    immediately, handing the reins of the organization to CEO Elect Jeff Moorad.
Summary: The passage is the Arizona Diamondbacks.
The summary "Summary" fits "Category": Sports

Passage: 3 injured in plant fire in Japan. TOKYO, Aug. 20 (Xinhuanet) -- Fire broke
    out Friday at a tire plant belonging to Bridgestone Corp. in Amagi, western
    Fukuoka Prefecture of Japan, leaving 13 people injured.
Summary: The passage is about a plant fire.
The summary "Summary" fits "Category": World News

Passage: The Race is On: Second Private Team Sets Launch Date for Human Spaceflight (
    SPACE.com). SPACE.com - TORONTO, Canada -- A second team of rocketeers competing
    for the #36;10 million Ansari X Prize, a contest for privately funded suborbital
    space flight, has officially announced the first launch date for its manned
    rocket.
Summary: The passage is about a rocket.
The summary "Summary" fits "Category":
```

## Gold Output

```
technology and science
```

## K.2    AMAZON

*Description:* Amazon product classification dataset with 9 classes. He & McAuley (2016)
*Train Size:* 9000, *Test Size:* 9000

---

**Amazon Few Shot**

Input

```
Pick the correct category for the product.
Categories:
- Amazon Instant Video
- Books
- Clothing Shoes and Jewelry
- Electronics
- Kindle Store
- Movies and TV
- Musical Instruments
- Office Products
- Tools and Home Improvement

Product: This bra is extremely comfortable, affordable and pretty too! My only
    complaint, and the reason for 4 stars is that the straps can't be adjusted very
    much. Right now as it is I'm at maximum shortening of the straps, so as I wear it
     and it stretches I can see some adjustments in my future. Nothing my sewing
    machine can't fix though. All in all I'd recommend to someone like me whose been
    searching for a comfortable yet pretty bra for their girls (I'm 32GG).
Category: Clothing Shoes and Jewelry

Product: 1/8/10 Have been using this drill and am very pleased.  It has tons of torque
     and the handle comes in handy.  Just great to have a corded drill with this type
     of power for when the cordless 18V wont do the job or if you have a lot of
    screws to driveHavent used this drill yet but am looking forward to putting it to
     work.  I bought it because I only had cordless drills, aside from a great,
    rugged Milwaukee Hammer drill, was also confident because of great reviews from
    others, It is very solid and I like the quick change cord.  I expect to have this
     drill for years
Category: Tools and Home Improvement

Product: Lindsey is tired of being used by men so she swears off them until she meets
    Brad. Lindsey was offed a position to renovate a room for sex in a hotel. She has
     no limits on what she can do with the room. Lindsey is ready to create her
    ultimate fantasy and Brad is just the man to help her. He is hired to help with
    the construction portion of the room but one look at Lindsey and he would like to
     be renovating other things as well like her and him together. Brad has his own
    insecurities about woman and most people in general he has big dreams that he has
     never gotten any backing in. This is a great book to read about an in charge
    woman giving up control to a man who doesn't usually take it because he is not
    sure if he will be rejected. The are a great pare that burn up the pages and
    learn that if you open up just a little then true love an truly find you.
Category: Kindle Store

Product: I first read THE PROPHET in college back in the 60's. The book had a revival
    as did anything metaphysical in the turbulent 60's. It had a profound effect on
    me and became a book I always took with me. After graduation I joined the Peace
    Corps and during stressful training in country (Liberia) at times of illness and
    the night before I left, this book gave me great comfort. I read it before I
    married, just before and again after my children were born and again after two
    near fatal illnesses. I am always amazed that there is a chapter that reaches out
     to you, grabs you and offers both comfort and hope for the future.Gibran offers
    timeless insights and love with each word. I think that we as a nation should
    read AND learn the lessons here. It is definitely a time for thought and
    reflection this book could guide us through.
Category:
```

Gold Output

```
Books
```

**Amazon AMA** prompt()**-chain Example**

question()

```
Summarize the product.

Product: Was unsure when I purchased the DVD what to expect.  With real joy I can say
    that it was worth every cent and I have already watched it several times. The
    Storyline kept me interested.
Summarize: the product "Product": The product is a DVD.

Product: These are the best headphones I've ever owned. I recently purchased a
    replacement pair, as my original set died after several years of intensive use.
Summarize: the product "Product": The product is headphones.

Product: So these tights are tighter than most tights I own and when I take these off,
     they leave my legs feeling like they've been squeezed to death.
Summarize: the product "Product": The product is tights.

Product: I first read THE PROPHET in college back in the 60's. The book had a revival
    as did anything metaphysical in the turbulent 60's. It had a profound effect on
    me and became a book I always took with me. After graduation I joined the Peace
    Corps and during stressful training in country (Liberia) at times of illness and
    the night before I left, this book gave me great comfort. I read it before I
    married, just before and again after my children were born and again after two
    near fatal illnesses. I am always amazed that there is a chapter that reaches out
     to you, grabs you and offers both comfort and hope for the future.Gibran offers
    timeless insights and love with each word. I think that we as a nation should
    read AND learn the lessons here. It is definitely a time for thought and
    reflection this book could guide us through.
Summarize: the product "Product":
```

Model Output

```
The product is a book.
```

answer()

```
Pick the correct category for the product.

"Categories":
- Amazon Instant Video
- Books
- Clothing Shoes and Jewelry
- Electronics
- Kindle Store
- Movies and TV
- Musical Instruments
- Office Products
- Tools and Home Improvement

Product: Was unsure when I purchased the DVD what to expect.  With real joy I can say
    that it was worth every cent and I have already watched it several times.  The
    Storyline kept me interested.
Summary: The product is a DVD.
The summary "Summary" fits "Category": Amazon Instant Video

Product: These are the best headphones I've ever owned. I recently purchased a
    replacement pair, as my original set died after several years of intensive use.
Summary: The product is headphones.
The summary "Summary" fits "Category": Electronics

Product: So these tights are tighter than most tights I own and when I take these off,
     they leave my legs feeling like they've been squeezed to death.
Summary: The product is tights.
The summary "Summary" fits "Category": Clothing Shoes and Jewelry

Product: I first read THE PROPHET in college back in the 60's. The book had a revival
    as did anything metaphysical in the turbulent 60's. It had a profound effect on
    me and became a book I always took with me. After graduation I joined the Peace
    Corps and during stressful training in country (Liberia) at times of illness and
    the night before I left, this book gave me great comfort. I read it before I
    married, just before and again after my children were born and again after two
    near fatal illnesses. I am always amazed that there is a chapter that reaches out
     to you, grabs you and offers both comfort and hope for the future.Gibran offers
    timeless insights and love with each word. I think that we as a nation should
    read AND learn the lessons here. It is definitely a time for thought and
    reflection this book could guide us through.
Summary: The product is a book.
The summary "Summary" fits "Category":
```

Gold Output

```
Book
```

## K.3  ANLI R1

*Description:* Adversarially mined natural language inference dataset from Wikipedia. Nie et al. (2020)
*Train Size:* 16946, *Test Size:* 1000

**ANLI R1 Few Shot**

Input

Robert L. Hass (born March 1, 1941) is an American poet. He served as Poet Laureate of the United States from 1995 to 1997. He won the 2007 National Book Award and shared the 2008 Pulitzer Prize for the collection "Time and Materials: Poems 1997-2005." In 2014 he was awarded the Wallace Stevens Award from the Academy of American Poets.
Question: Robert L. Hass was Poet Laureate of the United States for two years. True, False, or Neither? True

Randall Park (born March 23, 1974) is an American actor, comedian, writer, and director. He played Kim Jong-Un in the 2014 film "The Interview", Minnesota governor Danny Chung in "Veep", and beginning in 2015 he portrayed Eddie Huang's father, American restaurateur Louis Huang, in ABC's television show "Fresh Off the Boat".
Question: Randall Park is dead True, False, or Neither? False

Fragaria x vescana is a hybrid strawberry cultivar that was created in an effort to combine the best traits of the garden strawberry ("Fragaria" x "ananassa"), which has large berries and vigorous plants, with the woodland strawberry ("Fragaria vesca"), which has an exquisite flavour, but small berries.
Question: Extensive testing went on to produce this berry True, False, or Neither? Neither

Daniel Zolnikov (born January 29, 1987) is a Republican member of the Montana Legislature. He was elected to House District 47 which represents Billings, Montana After redistricting, he now represents House District 45. He has made a name for himself pursuing pro-privacy legislation.
Question: There is no information indicating whether Daniel Zolnikov is a good legislator or not. True, False, or Neither?

## Gold Output

neither

## ANLI R1 AMA prompt()-chain Example

question()

Rewrite the statement as a yes/no question.

Statement: most of the light comes from the sun
Question: Does most of the light come from the sun?

Statement: the test was not hard
Question: Was the test not hard?

Statement: it is a good idea to buy your parents gifts
Question: Is it a good idea to buy your parents gifts?

Statement: the balloon popped
Question: Did the balloon pop?

Statement: The father and son went camping to California.
Question: Did the father and son go camping?

Statement: There is no information indicating whether Daniel Zolnikov is a good legislator or not.
Question:

## Model Output

Is there information indicating whether Daniel Zolnikov is a good legislator?

answer()

```
Answer the question. If there is no evidence in the context, return "Unknown".

Context: According to Biraben, the plague was present somewhere in Italy and affected
    1,200 people.
Question: Based on the context, Did the plague affect people in Europe?
Answer: yes, people in Italy, Europe

Context: Policies aiming at controlling unemployment and in particular at reducing its
    inequality-associated effects support economic growth.
Question: Based on the context, Is confidence a factor in increasing self-esteem?
Answer: unknown

Context: The term "matter" is used throughout physics in a bewildering variety of
    contexts: for example, one refers to "condensed matter physics", "elementary
    matter", "partonic" matter, "dark" matter, "anti"-matter, "strange" matter, and "
    nuclear" matter.
Question: Based on the context, Is anti-matter made of electrons?
Answer: Unknown

Context: Daniel Zolnikov (born January 29, 1987) is a Republican member of the Montana
    Legislature. He was elected to House District 47 which represents Billings,
    Montana After redistricting, he now represents House District 45. He has made a
    name for himself pursuing pro-privacy legislation.
Question: Based on the context, Is there information indicating whether Daniel
    Zolnikov is a good legislator?
Answer:
```

## Gold Output

```
neither
```

## K.4 ANLI R2

*Description:* Adversarially mined natural language inference dataset from Wikipedia. Nie et al. (2020)
*Train Size:* 45460, *Test Size:* 1000

**ANLI R2 Few Shot**

Input

```
Matter was a London music venue and nightclub that opened in September 2008, after
    three years of planning. A 2,600 capacity live music venue and nightclub, it was
    the second project for owners Cameron Leslie and Keith Reilly, founders of the
    London club Fabric. Matter is the third venue to open at The O in south-east
    London.
Question: The owners own more than one London club. True, False, or Neither? True

Whitechapel is a British television drama series produced by Carnival Films, in which
    detectives in London's Whitechapel district dealt with murders which replicated
    historical crimes. The first series was first broadcast in the UK on 2 February
    2009 and depicted the search for a modern copycat killer replicating the murders
    of Jack the Ripper.
Question: Some of the victims depicted in Whitechapel were women True, False, or
    Neither? Neither

Nannina de' Medici (14 February 1448 – 14 May 1493), born Lucrezia de' Medici, was the
     second daughter of Piero di Cosimo de' Medici and Lucrezia Tornabuoni. She was
    thus the elder sister of Lorenzo de' Medici. She married Bernardo Rucellai. Her
    father's name was Piero, so she is sometimes known as Lucrezia di Piero de'
    Medici.
Question: Nannina de' Medici is sometimes known as Ivanka Trump True, False, or
    Neither? False

There is a little Shia community in El Salvador. There is an Islamic Library operated
    by the Shia community, named "Fatimah Az-Zahra". They published the first Islamic
     magazine in Central America: "Revista Biblioteca Islamica". Additionally, they
    are credited with providing the first and only Islamic library dedicated to
    spreading Islamic culture in the country.
Question: The community is south of the United States. True, False, or Neither?
```

### Gold Output

```
true
```

### ANLI R2 AMA prompt()-chain Example

question()

```
Rewrite the statement as a yes/no question.

Statement: most of the light comes from the sun
Question: Does most of the light come from the sun?

Statement: the test was not hard
Question: Was the test not hard?

Statement: it is a good idea to buy your parents gifts
Question: Is it a good idea to buy your parents gifts?

Statement: the balloon popped
Question: Did the balloon pop?

Statement: The father and son went camping to California.
Question: Did the father and son go camping?

Statement: The community is south of the United States.
Question:
```

### Model Output

```
Is the community south of the United States?
```

answer()

```
Answer the question. If there is no evidence in the context, return "Unknown".

Context: According to Biraben, the plague was present somewhere in Italy and affected
    1,200 people.
Question: Based on the context, Did the plague affect people in Europe?
Answer: yes, people in Italy, Europe

Context: Policies aiming at controlling unemployment and in particular at reducing its
    inequality-associated effects support economic growth.
Question: Based on the context, Is confidence a factor in increasing self-esteem?
Answer: unknown

Context: The term "matter" is used throughout physics in a bewildering variety of
    contexts: for example, one refers to "condensed matter physics", "elementary
    matter", "partonic" matter, "dark" matter, "anti"-matter, "strange" matter, and "
    nuclear" matter.
Question: Based on the context, Is anti-matter made of electrons?
Answer: Unknown

Context: There is a little Shia community in El Salvador. There is an Islamic Library
    operated by the Shia community, named "Fatimah Az-Zahra". They published the
    first Islamic magazine in Central America: "Revista Biblioteca Islamica".
    Additionally, they are credited with providing the first and only Islamic library
     dedicated to spreading Islamic culture in the country.
Question: Based on the context, Is the community south of the United States?
Answer:
```

Gold Output

```
true
```

## K.5    ANLI R3

*Description:* Adversarially mined natural language inference dataset from Wikipedia, News and other data sources. Nie et al. (2020)
*Train Size:* 100459, *Test Size:* 1200

**ANLI R3 Few Shot**

Input

```
And that means that the local law enforcement officials need help at the federal level
    . Programs like Project Exile where the federal government intensifies arresting
    people who illegally use guns. And we haven't done a very good job of that at the
     federal level recently.
Question: There are only federal enforcement officials. True, False, or Neither? False

Scary Dream
Tom woke up in a cold sweat. He was shaking and scared. He realized he
    had just had a scary dream. Tom was too afraid to fall back asleep. Instead he
    stayed up all night.
Question: Tom experienced a bad nightmare that kept him from sleeping. True, False, or
     Neither? True

Wayman Lawrence Tisdale (June 9, 1964 – May 15, 2009) was an American professional
    basketball player in the NBA and a smooth jazz bass guitarist. A three-time All
    American at the University of Oklahoma, he was elected to the National Collegiate
     Basketball Hall of Fame in 2009.
Question: Wayman Tisdale played smooth jazz bass guitar at the University of Oklahoma
    True, False, or Neither? Neither

For one night, all of Clinton's non-varsity squads achieved perfection sweeping Altus
    in seventh, eighth and ninth grade basketball at ... PLEASE LOG IN FOR PREMIUM
    CONTENT. Our website requires visitors to log in to view the best local news from
     Clinton Daily News. Not yet a subscriber? Subscribe today! Thank you!
Question: This headline leads to more information that is behind a paywall. True,
    False, or Neither?
```

Gold Output

```
true
```

---

**ANLI R3 AMA** prompt()**-chain Example**

question()

```
Rewrite the statement as a yes/no question.

Statement: most of the light comes from the sun
Question: Does most of the light come from the sun?

Statement: the test was not hard
Question: Was the test not hard?

Statement: it is a good idea to buy your parents gifts
Question: Is it a good idea to buy your parents gifts?

Statement: the balloon popped
Question: Did the balloon pop?

Statement: The father and son went camping to California.
Question: Did the father and son go camping?

Statement: This headline leads to more information that is behind a paywall.
Question:
```

Model Output

```
Does this headline lead to more information that is behind a paywall?
```

---

answer()

```
Answer the question. If there is no evidence in the context, return "Unknown".

Context: According to Biraben, the plague was present somewhere in Italy and affected
    1,200 people.
Question: Based on the context, Did the plague affect people in Europe?
Answer: yes, people in Italy, Europe

Context: Policies aiming at controlling unemployment and in particular at reducing its
    inequality-associated effects support economic growth.
Question: Based on the context, Is confidence a factor in increasing self-esteem?
Answer: unknown

Context: The term "matter" is used throughout physics in a bewildering variety of
    contexts: for example, one refers to "condensed matter physics", "elementary
    matter", "partonic" matter, "dark" matter, "anti"-matter, "strange" matter, and "
    nuclear" matter.
Question: Based on the context, Is anti-matter made of electrons?
Answer: Unknown

Context: For one night, all of Clinton's non-varsity squads achieved perfection
    sweeping Altus in seventh, eighth and ninth grade basketball at ... PLEASE LOG IN
     FOR PREMIUM CONTENT. Our website requires visitors to log in to view the best
    local news from Clinton Daily News. Not yet a subscriber? Subscribe today! Thank
    you!
Question: Based on the context, Does this headline lead to more information that is
    behind a paywall?
Answer:
```

Gold Output

```
true
```

## K.6   BOOLQ

*Description:* Yes/no QA task over small wikipedia passages. Clark et al. (2019)
*Train Size:* 9427, *Test Size:* 3245

---

**BoolQ Few Shot**

Input

```
Answer the question.

Context: Red River (1948 film) -- The film's supporting cast features Walter Brennan,
    Joanne Dru, Coleen Gray, Harry Carey, John Ireland, Hank Worden, Noah Beery Jr.,
    Harry Carey Jr. and Paul Fix. Borden Chase and Charles Schnee wrote the
    screenplay, based on Chase's original story (which was first serialized in The
    Saturday Evening Post in 1946 as ''Blazing Guns on the Chisholm Trail'').
Question: is the movie red river based on a true story
Answer: No

Context: Legal drinking age -- Kazakhstan, Oman, Pakistan, Qatar, Sri Lanka,
    Tajikistan, Thailand, United Arab Emirates, Federated States of Micronesia, Palau
    , Paraguay, Solomon Islands, India (certain states), the United States (except U.
    S. Virgin Islands and Puerto Rico), Yemen (Aden and Sana'a), Japan, Iceland,
    Canada (certain Provinces and Territories), and South Korea have the highest set
    drinking ages; however, some of these countries do not have off-premises drinking
     limits. Austria, Antigua and Barbuda, Belgium, Bermuda, British Virgin Islands,
    Cuba, Ethiopia, Gibraltar, Luxembourg and Nicaragua have the lowest set drinking
    ages.
Question: is america the only country with a drinking age of 21
Answer: No

Context: Drinking in public -- Drinking in public is legal in England and Wales -- one
     may carry a drink from a public house down the street (though it is preferred
    that the user requests a plastic glass to avoid danger of breakage and because
    the taking of the glass could be considered an offence of Theft as only the drink
     has been purchased), and one may purchase alcohol at an off-licence and
    immediately begin drinking it outside. Separately, one may drink on aeroplanes
    and on most National Rail train services, either purchasing alcohol on-board or
    consuming one's own.
Question: is it legal to drink in public in london
Answer: Yes

Context: Harry Potter and the Escape from Gringotts -- Harry Potter and the Escape
    from Gringotts is an indoor steel roller coaster at Universal Studios Florida, a
    theme park located within the Universal Orlando Resort. Similar to dark rides,
    the roller coaster utilizes special effects in a controlled-lighting environment
    and also employs motion-based 3-D projection of both animation and live-action
    sequences to enhance the experience. The ride, which is themed to the Gringotts
    Wizarding Bank, became the flagship attraction for the expanded Wizarding World
    of Harry Potter when it opened on July 8, 2014.
Question: is harry potter and the escape from gringotts a roller coaster ride? True or
     False?
Answer:
```

Gold Output

```
True
```

---

**BoolQ AMA** prompt()-**chain Example**

answer()

```
Answer the question using the context.

Context: Tonic water -- Tonic water (or Indian tonic water) is a carbonated soft drink
     in which quinine is dissolved. Originally used as a prophylactic against malaria
     , tonic water usually now has a significantly lower quinine content and is
     consumed for its distinctive bitter flavor. It is often used in mixed drinks,
     particularly in gin and tonic.
Question: does tonic water still have quinine in it?
Answer: yes

Context: Northern bobwhite -- The northern bobwhite, Virginia quail or (in its home
     range) bobwhite quail (Colinus virginianus) is a ground-dwelling bird native to
     the United States, Mexico, and the Caribbean. It is a member of the group of
     species known as New World quails (Odontophoridae). They were initially placed
     with the Old World quails in the pheasant family (Phasianidae), but are not
     particularly closely related. The name ''bobwhite'' derives from its
     characteristic whistling call. Despite its secretive nature, the northern
     bobwhite is one of the most familiar quails in eastern North America because it
     is frequently the only quail in its range. Habitat degradation has likely
     contributed to the northern bobwhite population in eastern North America
     declining by roughly 85% from 1966-2014. This population decline is apparently
     range-wide and continuing.
Question: is a quail the same as a bobwhite?
Answer: yes

Context: United States Department of Homeland Security -- In fiscal year 2017, it was
     allocated a net discretionary budget of $40.6 billion. With more than 240,000
     employees, DHS is the third largest Cabinet department, after the Departments of
     Defense and Veterans Affairs. Homeland security policy is coordinated at the
     White House by the Homeland Security Council. Other agencies with significant
     homeland security responsibilities include the Departments of Health and Human
     Services, Justice, and Energy
Question: is department of homeland security part of dod?
Answer: no

Context: Harry Potter and the Escape from Gringotts -- Harry Potter and the Escape
     from Gringotts is an indoor steel roller coaster at Universal Studios Florida, a
     theme park located within the Universal Orlando Resort. Similar to dark rides,
     the roller coaster utilizes special effects in a controlled-lighting environment
     and also employs motion-based 3-D projection of both animation and live-action
     sequences to enhance the experience. The ride, which is themed to the Gringotts
     Wizarding Bank, became the flagship attraction for the expanded Wizarding World
     of Harry Potter when it opened on July 8, 2014.
Question: is harry potter and the escape from gringotts a roller coaster ride?
Answer:
```

### Model Output

```
true
```

## K.7 CB

*Description:* Three-class textual entailement task. Wang et al. (2019)
*Train Size:* 250, *Test Size:* 56

**CB Few Shot**

Input

```
B: Now see I. A: I'm intrigued by it, but I'm not sure I want to go see it yet. B:
    Yeah, I don't think I want to see that either.
Question: she wants to see that True, False, or Neither? False

A: Yeah. The radio doesn't really have much news sometimes. The stations I listen to
    are just mainly music. B: Yeah, I think you pretty much have to listen to all
    news station to get any news at all. A: Yeah. Do you think that TV is, uh, pretty
     accurate.
Question: TV is pretty accurate True, False, or Neither? Neither

It is part of their religion, a religion I do not scoff at as it holds many elements
    which match our own even though it lacks the truth of ours. At one of their great
     festivals they have the ritual of driving out the devils from their bodies.
    First the drummers come on - I may say that no women are allowed to take part in
    this ritual and the ladies here will perhaps agree with me that they are
    fortunate in that omission.
Question: no women are allowed to take part in this ritual True, False, or Neither?
    True

Modify the arachnids, said the researchers. Change their bodies and conditions, and
    you could get fibres like glass, still monofilament, but with logarithmic
    progressions of possibilities of strength and flexibility, and the ability to
    resonate light-particles or sound-waves undistorted, scarcely weakened over
    thousands of miles. Who said the arachnids had to be totally organic?
Question: arachnids had to be totally organic. True, False, or Neither?
```

## Gold Output

```
False
```

**CB AMA** prompt()**-chain Example**

question()

```
Rewrite the statement as a yes/no question.

Statement: most of the light comes from the sun
Question: Does most of the light come from the sun?

Statement: the test was not
Question: Was the test hard?

Statement: it is a good idea to buy your parents gifts
Question: Is it a good idea to buy your parents gifts?

Statement: the balloon popped
Question: Did the balloon pop?

Statement: The father and son went camping to California.
Question: Did the father and son go camping?

Statement: arachnids had to be totally organic
Question:
```

## Model Output

```
Did arachnids have to be totally organic?
```

answer()

```
Provide the answer to the question from the passage.

Passage: When Judy and Jack went to school, they got in trouble with their teacher for
     being late. I didn't think it was very fair.
Question: Did she think it was fair?
Answer: No

Passage: If inflation is occurring, leading to higher prices for basic necessities
     such as gas by 2 dollars. Do you think that inflation is good for society?
Question: Is inflation good for society?
Answer: Maybe

Passage: Put yourself out there. The more time you spend dating and socializing, the
     more likely you will find a boyfriend you like.
Question: Does socializing help you find a boyfriend?
Answer: Yes

Passage: Modify the arachnids, said the researchers. Change their bodies and
     conditions, and you could get fibres like glass, still monofilament, but with
     logarithmic progressions of possibilities of strength and flexibility, and the
     ability to resonate light-particles or sound-waves undistorted, scarcely weakened
      over thousands of miles. Who said the arachnids had to be totally organic?
Question: Did arachnids have to be totally organic?
Answer:
```

Gold Output

```
false
```

## K.8 COPA

*Description:* Casual reasoning dataset where task is to select the alternative that more plausibly has a causal relation with the premise. Wang et al. (2019)
*Train Size:* 400, *Test Size:* 100

**COPA Few Shot**

Input

```
Pick the more likely continuation to the following sentence.

Context: The truck crashed into the motorcycle on the bridge so The motorcyclist died.

Context: The customer came into the boutique because The window display caught her eye
    .

Context: The print on the brochure was tiny so The man put his glasses on.

Context: The
```

Model Choices

```
- woman became famous so
- photographers followed her
```

Gold Output

```
photographers followed her
```

**COPA AMA** prompt()**-chain Example**

answer()

```
Pick the correct ending for the example.

Question: (because 'she took medicine', because 'she got expelled') My roommate was
    feeling better because?
Answer: 'she took medicine'

Question: (because 'he does not practice', because 'he is fast') Matt is not good at
    soccer because?
Answer: 'he does not practice'

Question: (because 'she was smart', because 'she never did her homework') The girl
    went to college and graduated with honors because?
Answer: 'she was smart'

Question: (and so 'her family avoided her.', and so 'photographers followed her.') The
    woman became famous and so?
Answer:
```

### Model Choices

```
- woman became famous so
- photographers followed her
```

### Gold Output

```
photographers followed her
```

## K.9 DBPEDIA

*Description:* Ontology classification dataset with 14 classes. Zhang et al. (2015)
*Train Size:* 560000, *Test Size:* 70000

**DBPedia Few Shot**

Input

```
Pick the correct category for the passage.
Categories:
- company
- educational institution
- artist
- athlete
- office holder
- mean of transportation
- building
- natural place
- village
- animal
- plant
- album
- film
- written work

Passage: Garrison Cadet College Kohat – Garrison Cadet College Kohat is Situated in
    Kohat. Foundation stone was laid by the then Prime Minister of Islamic Republic
    of Pakistan Late Mohtarama Benazir Bhutto in 1992. Lieutenant General Arif
    Bangash Lieutenant General K.K Afridi Major General Shirendil Niazi and Colonel
    Idreesm(founder Principal) Dr.
Category: Educational Institution

Passage: River Ingrebourne – The River Ingrebourne is a tributary of the River Thames
    27 miles (43.3 km) in length. It is considered a strategic waterway in London
    forming part of the Blue Ribbon Network. It flows through the London Borough of
    Havering roughly from north to south joining the Thames at Rainham.
Category: Natural Place

Passage: USS Patrol No. 4 (SP-8) – USS Patrol No. 4 (SP-8) often rendered as USS
    Patrol #4 was an armed motorboat that served in the United States Navy as a
    patrol vessel from 1917 to 1919.Patrol No. 4 was built as a private motorboat of
    the same name in 1915 by Britt Brothers at Lynn Massachusetts. She was one of
    five motorboats built to the same design for private owners by Britt Brothers as
    part of the civilian Preparedness Movement program with an understanding that
    they would enter U.S.
Category: Mean Of Transportation

Passage: TY KU – TY KU is an American alcoholic beverage company that specializes in
    sake and other spirits. The privately-held company was founded in 2004 and is
    headquartered in New York City New York. While based in New York TY KU's
    beverages are made in Japan through a joint venture with two sake breweries.
    Since 2011 TY KU's growth has extended its products into all 50 states.
Category:
```

Gold Output

```
company
```

**DBPedia AMA** prompt()**-chain Example**

question()

```
Summarize the passage.

"Categories":
- company
- educational institution
- artist
- athlete
- office holder
- mean of transportation
- building
- natural place
- village
- animal
- plant
- album
- film
- written work

Passage: Personality and Mental Health - Personality and Mental Health:
    Multidisciplinary Studies from Personality Dysfunction to Criminal Behaviour is a
     quarterly peer-reviewed academic journal published by Wiley-Blackwell on behalf
    of the Centre for Health and Justice.
Summarize: the passage "Passage": The passage is about a journal.

Passage: RNLB Mona (ON 775) - RNLB Mona (ON 775) was a Watson Class lifeboat based at
    Broughty Ferry in Scotland that capsized during a rescue attempt with the loss of
     her entire crew of eight men. The Mona was built in 1935 and in her time saved
    118 lives.
Summarize: the passage "Passage": The passage is about a lifeboat.

Passage: Sayonara mo Ienakatta Natsu - Sayonara mo Ienakatta Natsu is an album by
    Mikuni Shimokawa released on July 4 2007 by Pony Canyon.This album consists of
    eleven songs; several new songs and some songs which were previously released as
    singles.
Summarize: the passage "Passage": The passage is about a album.

Passage: TY KU - TY KU is an American alcoholic beverage company that specializes in
    sake and other spirits. The privately-held company was founded in 2004 and is
    headquartered in New York City New York. While based in New York TY KU's
    beverages are made in Japan through a joint venture with two sake breweries.
    Since 2011 TY KU's growth has extended its products into all 50 states.
Summarize: the passage "Passage":
```

Model Output

```
The passage is about a company.
```

answer()

```
Pick one category for the following text.

"Categories":
- company
- educational institution
- artist
- athlete
- office holder
- mean of transportation
- building
- natural place
- village
- animal
- plant
- album
- film
- written work

Passage: Personality and Mental Health - Personality and Mental Health:
    Multidisciplinary Studies from Personality Dysfunction to Criminal Behaviour is a
     quarterly peer-reviewed academic journal published by Wiley-Blackwell on behalf
    of the Centre for Health and Justice.
Summary: The passage is about a journal.
The summary "Summary" fits "Category": written work

Passage: RNLB Mona (ON 775) - RNLB Mona (ON 775) was a Watson Class lifeboat based at
    Broughty Ferry in Scotland that capsized during a rescue attempt with the loss of
     her entire crew of eight men. The Mona was built in 1935 and in her time saved
    118 lives.
Summary: The passage is about a lifeboat.
The summary "Summary" fits "Category": mean of transportation

Passage: Sayonara mo Ienakatta Natsu - Sayonara mo Ienakatta Natsu is an album by
    Mikuni Shimokawa released on July 4 2007 by Pony Canyon.This album consists of
    eleven songs; several new songs and some songs which were previously released as
    singles.
Summary: The passage is about a album.
The summary "Summary" fits "Category": album

Passage: TY KU - TY KU is an American alcoholic beverage company that specializes in
    sake and other spirits. The privately-held company was founded in 2004 and is
    headquartered in New York City New York. While based in New York TY KU's
    beverages are made in Japan through a joint venture with two sake breweries.
    Since 2011 TY KU's growth has extended its products into all 50 states.
Summary: The passage is about a company.
The summary "Summary" fits "Category":
```

Gold Output

```
company
```

## K.10   DROP

*Description:*   A reading comprehension benchmark requiring discrete reasoning over paragraphs. Dua et al. (2019)
*Train Size:* 77409, *Test Size:* 9536

**DROP Few Shot**

Input

```
Passage: As of the 2010 United States Census, there were 16,589 people, 6,548
    households, and 4,643 families residing in the county. The population density was
     . There were 7,849 housing units at an average density of . The racial makeup of
     the county was 96.8% white, 0.7% black or African American, 0.6% American Indian
    , 0.2% Asian, 0.2% from other races, and 1.5% from two or more races. Those of
    Hispanic or Latino origin made up 0.6% of the population. In terms of ancestry,
    23.4% were Germans, 22.3% were Americans, 13.6% were Irish people, and 11.0% were
     English people.
Question: How many percent of people were not Asian?
Answer: unknown

Passage: The health sector comprises 17 specialized hospitals and centers, 4 regional
    diagnostic and treatment centers, 9 district and 21 aimag general hospitals, 323
    soum hospitals, 18 feldsher posts, 233 family group practices, 536 private
    hospitals, and 57 drug supply companies/pharmacies. In 2002, the total number of
    health workers was 33,273, of whom 6823 were doctors, 788 pharmacists, 7802
    nurses, and 14,091 mid-level personnel. At present, there are 27.7 physicians and
     75.7 hospital beds per 10,000 inhabitants.
Question: What profession had more health workers, doctors or nurses?
Answer: nurses

Passage: The exact number of peasant deaths is unknown, and even the course of events
    are not clear, because the government, to hide the size of the massacre, ordered
    the destruction of all documents relating to the uprising. Historian Markus Bauer
     mentions a greatly underestimated official figure of 419 deaths, while an
    unofficial figure, circulated by the press and widely accepted, of about 10,000
    peasants killed, has never been proven to be true. The same figure of 419 deaths
    was mentioned by Ion I. C. Bratianu in the Romanian Parliament. The data
    available to the Prime Minister Dimitrie Sturdza indicated 421 deaths between 28
    March and 5 April 1907. Likewise, about 112 were injured and 1,751 detained.
    Newspapers patronized by Constantin Mille, Adevarul and Dimineata, gave a figure
    of 12,000-13,000 victims. In a conversation with the British ambassador in
    Bucharest, King Carol I mentioned a figure of "several thousand". According to
    figures given by Austrian diplomats, between 3,000-5,000 peasants were killed,
    while the French Embassy mentioned a death toll ranging between 10,000-20,000.
    Historians put the figures between 3,000-18,000, the most common being 11,000
    victims.
Question: Which organizations said the death toll to be beyond 10,000?
Answer: Newspapers patronized by Constantin Mille

Passage: Still searching for their first win, the Bengals flew to Texas Stadium for a
    Week 5 interconference duel with the Dallas Cowboys. In the first quarter,
    Cincinnati trailed early as Cowboys kicker Nick Folk got a 30-yard field goal,
    along with RB Felix Jones getting a 33-yard TD run. In the second quarter, Dallas
     increased its lead as QB Tony Romo completed a 4-yard TD pass to TE Jason Witten
    . The Bengals would end the half with kicker Shayne Graham getting a 41-yard and
    a 31-yard field goal. In the third quarter, Cincinnati tried to rally as QB
    Carson Palmer completed an 18-yard TD pass to WR T. J. Houshmandzadeh. In the
    fourth quarter, the Bengals got closer as Graham got a 40-yard field goal, yet
    the Cowboys answered with Romo completing a 57-yard TD pass to WR Terrell Owens.
    Cincinnati tried to come back as Palmer completed a 10-yard TD pass to
    Houshmandzadeh (with a failed 2-point conversion), but Dallas pulled away with
    Romo completing a 15-yard TD pass to WR Patrick Crayton.
Question: Which team scored the final TD of the game?
Answer:
```

Gold Output

```
dallas
```

**DROP AMA** prompt()**-chain Example**

answer()

```
Answer the question. If there is no evidence in the context, return "Unknown".

Context: According to Biraben, the plague was present somewhere in Europe in every
    year between 1346 and 1671
Question: Where was the plague present?
Answer: somewhere in Europe

Context: Policies aiming at controlling unemployment and in particular at reducing its
    inequality-associated effects support economic growth.
Question: What's one factor in increasing self-esteem?
Answer: Unknown

Context: The term "matter" is used throughout physics in a bewildering variety of
    contexts: for example, one refers to "condensed matter physics", "elementary
    matter", "partonic" matter, "dark" matter, "anti"-matter, "strange" matter, and "
    nuclear" matter.
Question: What is another name for anti-matter?
Answer: Unknown

Context: Still searching for their first win, the Bengals flew to Texas Stadium for a
    Week 5 interconference duel with the Dallas Cowboys. In the first quarter,
    Cincinnati trailed early as Cowboys kicker Nick Folk got a 30-yard field goal,
    along with RB Felix Jones getting a 33-yard TD run. In the second quarter, Dallas
     increased its lead as QB Tony Romo completed a 4-yard TD pass to TE Jason Witten
    . The Bengals would end the half with kicker Shayne Graham getting a 41-yard and
    a 31-yard field goal. In the third quarter, Cincinnati tried to rally as QB
    Carson Palmer completed an 18-yard TD pass to WR T. J. Houshmandzadeh. In the
    fourth quarter, the Bengals got closer as Graham got a 40-yard field goal, yet
    the Cowboys answered with Romo completing a 57-yard TD pass to WR Terrell Owens.
    Cincinnati tried to come back as Palmer completed a 10-yard TD pass to
    Houshmandzadeh (with a failed 2-point conversion), but Dallas pulled away with
    Romo completing a 15-yard TD pass to WR Patrick Crayton.
Question: Which team scored the final TD of the game?
Answer:
```

Model Output

```
dallas
```

## K.11  MULTIRC

*Description:* Multi-sentence reading comprehension dataset. Wang et al. (2019)
*Train Size:* 27243, *Test Size:* 953

**MultiRC Few Shot**

Input

```
Answer if the possible answer is a correct answer to the question.

Passage: While this process moved along, diplomacy continued its rounds. Direct
    pressure on the Taliban had proved unsuccessful. As one NSC staff note put it, "
    Under the Taliban, Afghanistan is not so much a state sponsor of terrorism as it
    is a state sponsored by terrorists." In early 2000, the United States began a
    high-level effort to persuade Pakistan to use its influence over the Taliban. In
    January 2000, Assistant Secretary of State Karl Inderfurth and the State
    Department's counterterrorism coordinator, Michael Sheehan, met with General
    Musharraf in Islamabad, dangling before him the possibility of a presidential
    visit in March as a reward for Pakistani cooperation. Such a visit was coveted by
     Musharraf, partly as a sign of his government's legitimacy. He told the two
    envoys that he would meet with Mullah Omar and press him on Bin Laden. They left,
     however, reporting to Washington that Pakistan was unlikely in fact to do
    anything," given what it sees as the benefits of Taliban control of Afghanistan."
     President Clinton was scheduled to travel to India. The State Department felt
    that he should not visit India without also visiting Pakistan....
Question: Based on the previous passage, What did President Clinton's visit with
    Pakistan include?
Is "Discussing Bin Laden" a correct answer?
Answer: Yes

Passage: While this process moved along, diplomacy continued its rounds. Direct
    pressure on the Taliban had proved unsuccessful. As one NSC staff note put it, "
    Under the Taliban, Afghanistan is not so much a state sponsor of terrorism as it
    is a state sponsored by terrorists." In early 2000, the United States began a
    high-level effort to persuade Pakistan to use its influence over the Taliban. In
    January 2000, Assistant Secretary of State Karl Inderfurth and the State
    Department's counterterrorism coordinator, Michael Sheehan, met with General
    Musharraf in Islamabad, dangling before him the possibility of a presidential
    visit in March as a reward for Pakistani cooperation. Such a visit was coveted by
     Musharraf, partly as a sign of his government's legitimacy....
Question: Based on the previous passage, Where did President Clinton visit on March
    25, 2000?
Is "Parkistan" a correct answer?
Answer: Yes

Passage: The Agencies Confer When they learned a second plane had struck the World
    Trade Center, nearly everyone in the White House told us, they immediately knew
    it was not an accident. The Secret Service initiated a number of security
    enhancements around the White House complex. The officials who issued these
    orders did not know that there were additional hijacked aircraft, or that one
    such aircraft was en route to Washington. These measures were precautionary steps
     taken because of the strikes in New York. The FAA and White House
    Teleconferences. The FAA, the White House, and the Defense Department each
    initiated a multiagency teleconference before 9:30. Because none of these
    teleconferences-at least before 10:00- included...
Question: Based on the previous passage, To what did the CIA and FAA begin
    participating in at 9:40?
Is "Coffee hour" a correct answer?
Answer: No

Passage: What causes a change in motion? The application of a force. Any time an
    object changes motion, a force has been applied. In what ways can this happen?
    Force can cause an object at rest to start moving. Forces can cause objects to
    speed up or slow down. Forces can cause a moving object to stop. Forces can also
    cause a change in direction. In short, forces cause changes in motion. The moving
     object may change its speed, its direction, or both. We know that changes in
    motion require a force. We know that the size of the force determines the change
    in motion. How much an objects motion changes when a force is applied depends on
    two things. It depends on the strength of the force. It also depends on the
    objects mass. Think about some simple tasks you may regularly do. You may pick up
     a baseball. This requires only a very small force.
Question: Based on the previous passage, Would the mass of a baseball affect how much
    force you have to use to pick it up?
Is "Yes" a correct answer?
Answer:
```

Gold Output

```
yes
```

**MultiRC AMA** prompt()**-chain Example**

answer()

```
Answer if the possible answer is a correct answer to the question.

Passage: Sara wanted to play on a baseball team. She had never tried to swing a bat
    and hit a baseball before. Her Dad gave her a bat and together they went to the
    park to practice. Sara wondered if she could hit a ball. She wasn't sure if she
    would be any good. She really wanted to play on a team and wear a real uniform.
    She couldn't wait to get to the park and test out her bat. When Sara and her Dad
    reached the park, Sara grabbed the bat and stood a few steps away from her Dad.
    Sara waited as her Dad pitched the ball to her. Her heart was beating fast. She
    missed the first few pitches. She felt like quitting but kept trying. Soon she
    was hitting the ball very far. She was very happy and she couldn't wait to sign
    up for a real team. Her Dad was very proud of her for not giving up.
Question: Based on the previous passage, Who pitched the ball to Sara and where did it
    occur?
Is "Her dad did in the park" a correct answer?
Answer: yes

Passage: The Vice President stated that he called the President to discuss the rules
    of engagement for the CAP. He recalled feeling that it did no good to establish
    the CAP unless the pilots had instructions on whether they were authorized to
    shoot if the plane would not divert. He said the President signed off on that
    concept. The President said he remembered such a conversation, and that it
    reminded him of when he had been an interceptor pilot. The President emphasized
    to us that he had authorized the shootdown of hijacked aircraft. The Vice
    President's military aide told us he believed the Vice President spoke to the
    President just after entering the conference room, but he did not hear what they
    said. Rice, who entered the room shortly after the Vice President and sat next to
     him, remembered hearing him inform the President, "Sir, the CAPs are up. Sir,
    they're going to want to know what to do." Then she recalled hearing him say, "
    Yes sir." She believed...
Question: Based on the previous passage, Why was the Secret Service's information
    about United 93 flawed?
Is "The Secret Service Didn't have access to FAA information" a correct answer?
Answer: no

Passage: Patricia Cross and her boyfriend Larry Osborne , two students in a San
    Francisco school , become expelled for the publication of an off-campus
    underground paper . As a result , a philosophy professor , Dr. Jonathon Barnett ,
     resigns his teaching position and decides to become an advocate for the
    counterculture youth movement and , specifically , the use of LSD . The hippies
    of the Haight-Ashbury district first see him as a hero and then as something even
     more . Dr. Barnett even makes an appearance on the Joe Pyne TV show to voice his
     support of the hippie community and the use of LSD . One scheming young man sees
     the opportunity to build Dr. Barnett as the head of a cult centered around the
    use of LSD . He hopes to earn profit from the users , Dr. Barnett's speeches
    known as `` happenings , '' and their lifestyles . At a massive LSD-fueled dance
    , Patricia begins to have a bad trip Which leads to an argument between her and
    Pat , ultimately splitting the couple up...
Question: Based on the previous passage, Why did Dr. Barnett resign from teaching?
Is "Patricia expulsion" a correct answer?
Answer: yes

Passage: I wondered if that were my case--if I rode out for honour, and not for the
    pure pleasure of the riding. And I marvelled more to see the two of us, both
    lovers of one lady and eager rivals, burying for the nonce our feuds, and with
    the same hope serving the same cause. We slept the night at Aird's store, and
    early the next morning found Ringan. A new Ringan indeed, as unlike the buccaneer
     I knew as he was unlike the Quaker. He was now the gentleman of Breadalbane,
    dressed for the part with all the care of an exquisite. He rode a noble roan, in
    his Spanish...
Question: Based on the previous passage, Who is described as both buccaneer and
    cavalier?
Is "Quaker" a correct answer?
Answer: no

Passage: What causes a change in motion? The application of a force. Any time an
    object changes motion, a force has been applied. In what ways can this happen?
    Force can cause an object at rest to start moving. Forces can cause objects to
    speed up or slow down. Forces can cause a moving object to stop. Forces can also
    cause a change in direction. In short, forces cause changes in motion. The moving
     object may change its speed, its direction, or both. We know that changes in
    motion require a force. We know that the size of the force determines the change
    in motion. How much an objects motion changes when a force is applied depends on
    two things. It depends on the strength of the force. It also depends on the
    objects mass. Think about some simple tasks you may regularly do. You may pick up
     a baseball. This requires only a very small force.
Question: Based on the previous passage, Would the mass of a baseball affect how much
    force you have to use to pick it up?
Is "Yes" a correct answer?
```

Model Output

```
yes
```

## K.12  NATURAL QUESTIONS (NQ)

*Description:* Open-domain question answering that contains questions from real users. Kwiatkowski et al. (2019)
*Train Size:* 307373, *Test Size:* 7830

---

**Natural Questions (NQ) Few Shot**

Input

```
Question: who sings does he love me with reba
Answer: Linda Davis

Question: where do the great lakes meet the ocean
Answer: the Saint Lawrence River

Question: when does the new my hero academia movie come out
Answer: July 5 , 2018

Question: what is the main mineral in lithium batteries
Answer:
```

Gold Output

```
lithium
```

---

**Natural Questions (NQ) AMA** prompt()**-chain Example**

question()

```
Produce distinct questions.

Question: who plays Carrie Bradshaw in sex and the city?
Answer: Caroline "Carrie" Bradshaw is a fictional character from the HBO franchise Sex
    and the City, portrayed by Sarah Jessica Parker.

Question: what are the elements in air?
Answer: By mole fraction (i.e., by number of molecules), dry air contains 78.08%
    nitrogen, 20.95% oxygen, 0.93% argon, 0.04% carbon dioxide, and small amounts of
    other gases

Question: what is HP company?
Answer: HP Inc. is an American multinational information technology company
    headquartered in Palo Alto, California, that develops personal computers (PCs)

Question: when was the last season of FRIENDS released?
Answer: The series finale aired on May 6, 2004, and was watched by around 52.5 million
     American viewers, making it the fifth-most-watched series finale in television
    history

Question: what is the main mineral in lithium batteries
Answer:
```

Model Output

```
Lithium is the most common element in the periodic table, and is the lightest metal.
    It
```

---

answer()

```
Answer the question.

Context: The nearest airport to Palm Springs is Indio/Palm Springs (PSP) Airport which
     is 2.1 miles away.
Question: what airport is closest to palm springs?
Answer: Palm Springs International Airport

Context: Martin Luther King earned his Bachelor of Divinity degree from Crozer
     Theological Seminary, followed by a doctorate in Systematic Theology from Boston
     University.
Question: what degree did martin luther king get?
Answer: Bachelor of Divinity

Context: The Niger river runs in a crescent through Libya, Mali, Niger, on the border
     with Benin and then through Nigeria.
Question: what countries does the niger river flow through?
Answer: Libya

Context: Puerto Rico is a territory of the United States and uses the U.S. dollar.
Question: what type of currency is used in puerto rico?
Answer: United States dollar

Context: kitt was voice most often by William daniels.
Question: who played kitt in knight rider?
Answer: William Daniels

Context: Lithium is the most common element in the periodic table, and is the lightest
     metal. It
Question: what is the main mineral in lithium batteries
Answer:
```

Gold Output

```
lithium
```

## K.13   RTE

*Description:* Dataset where the task is to predict whether a proposed premise sentence entails a given hypothesis sentence. Wang et al. (2019)
*Train Size:* 2490, *Test Size:* 277

---

**RTE Few Shot**

Input

---

```
A force majeure is an act of God, said attorney Phil Wittmann, who represents the New
    Orleans Saints and owner Tom Benson's local interests.
Question: New Orleans Saints are property of Tom Benson. True or False? True

Scientists at the Genome Institute of Singapore (GIS) have discovered the complete
    genetic sequence of a coronavirus isolated from a Singapore patient with SARS.
Question: Singapore scientists reveal that SARS virus has undergone genetic changes.
    True or False? False

Frye says, that he (a homeopathy expert) and Iris Bell recently studied homeopathic
    treatment of fibromyalgia. A new analysis – comparing published studies of
    homeopathic drugs to matched, randomly selected studies of medical drugs –
    suggests that these apparent homeopathic drug effects are merely placebo effects.
Question: What really irks Frye and other doctors of homeopathy, however, is that
    homeopathic remedies are not supposed to be used like medical drugs. True or
    False?False

Security forces were on high alert after an election campaign in which more than 1,000
    people, including seven election candidates, have been killed.
Question: Security forces were on high alert after a campaign marred by violence. True
    or False?
```

**Gold Output**

```
True
```

**RTE AMA** prompt()**-chain Example**

question()

```
Rewrite the statement as a question.

Statement: most of the light comes from the sun
Question: Does most of the light come from the sun?

Statement: the test was not hard
Question: Was the test hard?

Statement: it was a good idea to buy your parents gifts
Question: Was it a good idea to buy your parents gifts?

Statement: The 20 cans will arrive in the grocery store tomorrow.
Question: Will the 20 cans arrive in the grocery store tomorrow?

Statement: the balloon popped
Question: Did the balloon pop?

Statement: The father and son went camping to California.
Question: Did the father and son go camping?

Statement: Security forces were on high alert after a campaign marred by violence.
Question:
```

**Model Output**

```
Were security forces on high alert after a campaign marred by violence?
```

answer()

```
Answer the question. If there is no evidence in the context, return "Unknown".

Context: Jenna's 10th birthday was yesterday evening and at least 10 of her friends
    attended the party.
Question: Did 10 friends attend Jenna's party?
Answer: Unknown, at least 10

Context: The bullies attacked John when he was walking through the elementary school
    parking lot and then got sent to the teacher's office.
Question: Did the bullies attack John in the teacher's office?
Answer: No, parking lot

Context: WISS discovered a new monkey disease in a remote tribe in the Amazon
    rainforrest last week. It was highly contagious.
Question: Did WISS discover a new disease?
Answer: Yes, new monkey disease

Context: Security forces were on high alert after an election campaign in which more
    than 1,000 people, including seven election candidates, have been killed.
Question: Were security forces on high alert after a campaign marred by violence?
Answer:
```

Gold Output

```
True
```

**RTE AMA Example (Open-Ended Question Generation)**

question()

```
Rewrite the statement as a question.

Statement: she prefers kittens over puppies
Question: What does she prefer over puppies?
Answer: kittens

Statement: Max and his wife went on a trip to Europe
Question: Where did Max and his wife go on a trip
Answer: Europe

Statement: jared was born during the war in 1942
Question: Jared was born during a war in which year?
Answer: 1942

Statement: it took jenna 7 attempts to solve the problem
Question: How many attempts did it take Jenna to solve the problem?
Answer: 7

Statement: Security forces were on high alert after a campaign marred by violence.
Question:
```

Model Output

```
Security forces wer on high alert after a campaign marred by what?
Answer: violence
```

### K.14 ReCoRD

*Description:* Reading comprehension dataset which requires commonsense reasoning. Wang et al. (2019)
*Train Size:* 100730, *Test Size:* 10000

**ReCoRD Few Shot**

Input

Context: The University of Pennsylvania has been named America's top party school by
    Playboy in the first time the Ivy League institution has made the list. Believe
    it or not the magazine gave the top spot to the college by declaring that 'UPenn
    puts other Ivies to shame with its union of brains, brewskies and bros.' In the
    magazine's ninth annual ranking of universities around the country, the
    University of Wisconsin-Madison scored the runner up slot. The University of
    Pennsylvania (pictured) has been named America's top party school by Playboy in
    the first time the Ivy League institution has made the list. The University of
    Wisconsin-Madison scored the runner up slot. Last year's winner West Virginia
    University slipped down to third place. It is the magazine's ninth annual ranking
     of universities around the country
Answer: Playboy writes: 'Boasting a notorious underground frat scene that school
    officials have deemed a nuisance, these renegades pony up thousands of dollars'
    worth of liquor for their parties-and competition among the houses means a balls-
    out war of debauchery.

Context: Garita Palmera, El Salvador (CNN) -- She talks to the pictures as if they
    could make her voice travel thousands of miles and reach her son's ears. "Oh, my
    son," Julia Alvarenga, 59, says in a tender voice at her home in this coastal
    town. And then she says, "I'm going to see him again." The past two weeks have
    been an emotional roller coaster for the Salvadoran woman. First, she learned her
     son had been missing for 13 months. Then she was told he had turned up half a
    world away. And now she's getting news he might be back home soon.. It's been an
    emotional time for the parents of castaway Jose Salvador Alvarenga. His mother,
    Julia, said her son didn't keep up, and they didn't even know he was missing. "I
    would pray to God, and I won't lie to you, I was crying," she says. For the
    excited residents of his town in El Salvador, Alvarenga is a hero
Answer: Even though their son has yet to return home, he's already a celebrity in
    Garita Palmera and neighboring towns.

Context: (CNN) -- Members of a well-known hacking group -- according to a statement
    and Twitter messages -- took credit Sunday for an online attack targeting San
    Francisco's embattled transit system. Anonymous -- in a news release attributed
    to the group, and backed up by related Twitter pages -- said it would take down
    the website of the Bay Area Rapid Transit System, known as BART, between noon and
     6 p.m. PT Sunday. This is in response to the system's decision to cut off
    cellphone signals at "select" subway stations in response to a planned protest
    last week. "By (cutting cell service), you have not only threatened your citizens
    ' safety, you have also performed an act of censorship," a seemingly computer-
    generated voice -- speaking over dramatic music and images -- said in a video
    posted online Sunday afternoon. "By doing this, you have angered Anonymous.". NEW
    : A video urges protesters Monday to wear red shirts and record the event.
    Statements attributed to Anonymous promised an online attack Sunday on BART.
    MyBART.gov appears Sunday to have been hacked. The system said it was prepared
    for hacks, as well as a planned protest Monday
Answer: "We're doing what we can to defend against any attack on the BART website,"
    the system said..

Context: Tracy Morgan hasn't appeared on stage since the devastating New Jersey crash
    that nearly ended his life last summer, but all that will change this fall when
    he returns to host Saturday Night Live. NBC announced on Twitter Monday that
    Morgan, an SNL alum with seven seasons as a cast member under his belt, will
    headline the third episode of Season 41 airing October 17. For Morgan, 46, it
    will be a second time hosting the long-running variety show, the first since the
    June 2014 pileup on the New Jersey Turnpike that killed his friend and mentor
    James 'Jimmy Mack' McNair.. Morgan, 46, will host third episode of season 41 of
    SNL airing October 17. He tweeted to his fans: 'Stoked to be going home...#SNL'.
    For the SNL alum who had spent seven years as cast member, it will be a second
    time hosting the show. Morgan has been sidelined by severe head trauma suffered
    in deadly June 2014 crash on New Jersey Turnpike that killed his friend. First
    episode of new SNL season will be hosted by Miley Cyrus, followed by Amy Schumer
Answer: 'On October 10, acclaimed comedian and star of the summer box office hit
    Trainwreck Amy Schumer will make her SNL debut, followed by

Model Choices

```
– Amy Schumer a week later
– James a week later
– Jimmy Mack a week later
– McNair a week later
– Miley Cyrus a week later
– Morgan a week later
– NBC a week later
– New Jersey a week later
– New Jersey Turnpike a week later
– Night Live a week later
– SNL a week later
– Season 41 a week later
– Tracy Morgan a week later
– Twitter a week later
```

Gold Output

```
Morgan, Tracy Morgan
```

**ReCoRD AMA** prompt()**-chain Example**

answer()

```
Complete the paragraph.

Context: Barack Hussein Obama is an American politician who served as the 44th
     president of the United States from 2009 to 2017. A member of the Democratic
     Party, he was the first African-American president of the United States. Obama
     previously served as a U.S. senator from Illinois from 2005 to 2008 and as an
     Illinois state senator from 1997 to 2004. Obama was senator of the state of
     Illinois prior to becoming a US president.

Context: (CNN) -- Saif al-Islam Gadhafi, 38, has never lived a day in which his father
      Moammar didn't rule Libya -- as its undisputed leader inside the country and an
     enigmatic, controversial voice for the world. And yet, as the Libyan government
     faced a stiff popular uprising, it was Moammar Gadhafi's second eldest son -- and
      not the Leader of the Revolution himself -- who was first to talk to the nation
     about the unrest and detail a plan to address it. The speech, made early Monday
     on Libyan state television, does not mean that Saif Gadhafi has usurped power
     from his father: Senior U.S. officials said there's no indication the elder
     Gadhafi is losing his grip.Saif al-Islam Gadhafi, 38, gives Libya's first public
     speech acknowledging unrest. There's been no public indication why he, and not
     his father Moammar, talked. Even while some may see the son as more open to
     change, there's little question that his loyalty remains first with Moammar and
     that his father has given little indication publicly that he's ready to let go
     and calls the shots.

Context: The Beatles were an English rock band, formed in Liverpool in 1960, that
     comprised John Lennon, Paul McCartney, George Harrison and Ringo Starr. They are
     regarded as the most influential band of all time and were integral to the
     development of 1960s counterculture and popular music's recognition as an art
     form. They were led by primary songwriters Lennon and McCartney. It is without a
     doubt that the Beatles were influential in rock and roll.

Context: Tracy Morgan hasn't appeared on stage since the devastating New Jersey crash
     that nearly ended his life last summer, but all that will change this fall when
     he returns to host Saturday Night Live. NBC announced on Twitter Monday that
     Morgan, an SNL alum with seven seasons as a cast member under his belt, will
     headline the third episode of Season 41 airing October 17. For Morgan, 46, it
     will be a second time hosting the long-running variety show, the first since the
     June 2014 pileup on the New Jersey Turnpike that killed his friend and mentor
     James 'Jimmy Mack' McNair.. Morgan, 46, will host third episode of season 41 of
     SNL airing October 17. He tweeted to his fans: 'Stoked to be going home...#SNL'.
     For the SNL alum who had spent seven years as cast member, it will be a second
     time hosting the show. Morgan has been sidelined by severe head trauma suffered
     in deadly June 2014 crash on New Jersey Turnpike that killed his friend. First
     episode of new SNL season will be hosted by Miley Cyrus, followed by Amy Schumer.
      'On October 10, acclaimed comedian and star of the summer box office hit
     Trainwreck Amy Schumer will make her SNL debut, followed by
```

## Model Choices

```
- Amy Schumer a week later
- James a week later
- Jimmy Mack a week later
- McNair a week later
- Miley Cyrus a week later
- Morgan a week later
- NBC a week later
- New Jersey a week later
- New Jersey Turnpike a week later
- Night Live a week later
- SNL a week later
- Season 41 a week later
- Tracy Morgan a week later
- Twitter a week later
```

## Model Output

```
Morgan, Tracy Morgan
```

## K.15  RealTime QA

*Description:* Dynamic question answering dataset that asks questions about current world facts. Kasai et al. (2022)
*Train Size:* 90, *Test Size:* 187

---

**RealTime QA Few Shot**

Input

```
Question: What is the capital city of Japan?
Answer: Tokyo

Article: 5 things to know for June 13: Gun laws, January 6, Covid, White ... If your
    day doesn't start until you're up to speed on the latest headlines, then let us
    introduce you to your new favorite morning fix. Sign up here for the '5 Things'
    newsletter. (CNN) Just imagine what a relief it would be if you could use the
    same charging cable for all of your devices -- your phone, laptop, earbuds,
    camera, tablet, portable speaker, etc. Well, in a huge step to reduce cable
    clutter and waste, European regulators say that Apple and other smartphone makers
     will be required to support a single common charging standard for all mobile
    devices as early as the fall of 2024. But Apple hates the idea (shocker) because
    that means about a billion devices will become obsolete.
Article: 5 things to know for March 11: Ukraine, Pandemic, MLB, North ... If your day
    doesn't start until you're up to speed on the latest headlines, then let us
    introduce you to your new favorite morning fix. Sign up here for the '5 Things'
    newsletter. (CNN) America, the "land of the free," is getting quite costly.
    Prices for gas, food and housing -- which are all necessary expenses -- are
    spiking across the country. Gas prices have risen 38% over the past year , and
    rising prices in pandemic-related sectors, such as travel and dining, are also
    expected as the US recovers from the Omicron wave of Covid-19. Here's what you
    need to know to Get Up to Speed and On with Your Day .
Article: Wi-Charge / consists of a transmitter and a receiver. Transmitter connects to
     a standard power outlet and converts electricity into infrared laser beam.
    Receivers use a miniature photo-voltaic cell to convert transmitted light into
    electrical power. Receivers can be embedded into a device or connected into an
    existing charging port. The transmitter automatically identifies chargeable
    receivers and start charging. Several devices can charge at the same time.
    According to Wi-Charge it can deliver several watts of power to a device at
    several meters away. The core technology is based on a "distributed laser
    resonator" which is formed by the retroreflectors within the
Article: Mobile broadband / added in 2005. CDPD, CDMA2000 EV-DO, and MBWA are no
    longer being actively developed. In 2011, 90% of the world's population lived in
    areas with 2G coverage, while 45% lived in areas with 2G and 3G coverage, and 5%
    lived in areas with 4G coverage. By 2017 more than 90% of the world's population
    is expected to have 2G coverage, 85% is expected to have 3G coverage, and 50%
    will have 4G coverage. A barrier to mobile broadband use is the coverage provided
     by the mobile service networks. This may mean no mobile network or that service
    is limited to
Article: Mobile edge computing / Combining elements of information technology and
    telecommunications networking, MEC also allows cellular operators to open their
    radio access network (RAN) to authorized third-parties, such as application
    developers and content providers. Technical standards for MEC are being developed
     by the European Telecommunications Standards Institute, which has produced a
    technical white paper about the concept. MEC provides a distributed computing
    environment for application and service hosting. It also has the ability to store
     and process content in close proximity to cellular subscribers, for faster
    response time. Applications can also be exposed to real-time radio access network
     (RAN) information. The key element is
Question: To help reduce cable clutter and waste, which continent will soon require
    Apple and other smartphone makers to support a single common charging standard
    for all mobile devices?
Answer:
```

Gold Output

```
Europe
```

---

**RealTime QA AMA** prompt()**-chain Example**

answer()

```
Answer the question given the articles.Article 1: Walmart is slashing prices on
    clothing and other products - CNN New York(CNN Business) Many shoppers have
    pulled back on buying clothing and other discretionary items as the highest
    inflation in four decades pinches their pocketbooks.
Article 2: Retail slowdown: Target cuts vendor orders, slashes prices as it ...
    Associated Press NEW YORK.
Article 3: Stores have too much stuff. That means discounts are coming | CNN ... New
    York(CNN Business).
Article 4: GM reports strong sales but says it's prepared for possible recession ...
    New York (CNN Business).
Article 5: Target is ramping up discounts. Here's why - CNN New York(CNN Business).
Question: Which major US retailer announced this week it is slashing prices on
    clothing and other products?
Answer: "Walmart"

Article 1: Article 1: JetBlue announces a deal to buy Spirit Airlines. Fares could
    surge.
Article 2: JetBlue-Spirit merger: Airlines have complaints over flights and fees
    Christopher Elliott Special to USA TODAY.
Article 3: JetBlue announces a deal to buy Spirit Airlines | CNN Business The
    announcement comes a day after Spirit pulled the plug on a deal to merge with
    Frontier.
Article 4: Spirit and Frontier pull plug on deal, setting stage for JetBlue to buy ...
     New York (CNN Buiness).
Article 5: Frontier Airlines, Spirit Airlines announce budget airline merger Budget
    airlines Frontier Airlines and Spirit Airlines.
Question: Which airline announced a deal this week to buy Spirit Airlines?
Answer: "JetBlue"

Article 1: Oak Fire: California's fast-moving wildfire burns 14,000 acres and ... (CNN
    ) A wildfire raging for a third day Sunday in central California's Mariposa
    County outside Yosemite National Park has burned more than 14, 000 acres and
    forced thousands to evacuate from rural communities.
Article 2: California Oak Fire: Rapidly-growing fire engulfs homes near ... For more
    on the fires, " United Shades of America with W. Kamau Bell " heads to California
     to discover how communities are learning to coexist with the frequent
    destruction.
Article 3: 5 things to know for July 25: Wildfires, Ukraine, Monkeypox, Volcano ... If
     your day doesn't start until you're up to speed on the latest headlines, then
    let us introduce you to your new favorite morning fix.
Article 4: Wildfires in US: 2 firefighting helicopter pilots die in Idaho ... Multiple
     wildfires raged across the U.S. Saturday, causing deaths, destruction and
    thousands of forced evacuations.
Article 5: Boulder wildfires: Hundreds of homes burn evacuations ordered BOULDER, Colo
    . - A ferocious wind-driven wildfire on Thursday destroyed hundreds of homes and
    businesses near Denver, forcing tens of thousands to flee and blanketing the area
     in smoke.
Question: A raging wildfire this week forced thousands of people to evacuate
    communities near which national park?
Answer: "Yosemite National Park"

Article 1: 5 things to know for June 13: Gun laws, January 6, Covid, White ... If your
     day doesn't start until you're up to speed on the latest headlines, then let us
    introduce you to your new favorite morning fix. Sign up here for the '5 Things'
    newsletter. (CNN) Just imagine what a relief it would be if you could use the
    same charging cable for all of your devices -- your phone, laptop, earbuds,
    camera, tablet, portable speaker, etc. Well, in a huge step to reduce cable
    clutter and waste, European regulators say that Apple and other smartphone makers
Article 2: 5 things to know for March 11: Ukraine, Pandemic, MLB, North ... If your
    day doesn't start until you're up to speed on the latest headlines, then let us
    introduce you to your new favorite morning fix. Sign up here for the '5 Things'
    newsletter. (CNN) America, the "land of the free," is getting quite costly.
    Prices for gas, food and housing -- which are all necessary expenses -- are
    spiking across the country. Gas prices have risen 38% over the past year , and
    rising prices in pandemic-related sectors, such as travel and dining, are also
    expected as
Article 3: Wi-Charge / consists of a transmitter and a receiver. Transmitter connects
    to a standard power outlet and converts electricity into infrared laser beam.
    Receivers use a miniature photo-voltaic cell to convert transmitted light into
    electrical power. Receivers can be embedded into a device or connected into an
    existing charging port. The transmitter automatically identifies chargeable
    receivers and start charging. Several devices can charge at the same time.
    According to Wi-Charge it can deliver several watts of power to a device at
    several meters away. The core technology is based on a "distributed laser
    resonator" which is formed by the
Article 4: Mobile broadband / added in 2005. CDPD, CDMA2000 EV-DO, and MBWA are no
    longer being actively developed. In 2011, 90% of the world's population lived in
    areas with 2G coverage, while 45% lived in areas with 2G and 3G coverage, and 5%
    lived in areas with 4G coverage. By 2017 more than 90% of the world's population
```

Gold Output

```
europe
```

## K.16  SST2

*Description:* Movie review binary sentiment classification dataset. Socher et al. (2013)
*Train Size:* 6920, *Test Size:* 1821

---

**SST2 Few Shot**

Input

```
For each snippet of text, label the sentiment of the text as positive or negative.

Text: in terms of execution this movie is careless and unfocused .
Sentiment: negative

Text: ... a pretentious and ultimately empty examination of a sick and evil woman .
Sentiment: negative

Text: the film 's plot may be shallow , but you 've never seen the deep like you see
    it in these harrowing surf shots .
Sentiment: positive

Text: a gob of drivel so sickly sweet , even the eager consumers of moore 's
    pasteurized ditties will retch it up like rancid creme brulee .
Sentiment:
```

Gold Output

```
negative
```

---

**SST2 AMA** prompt()**-chain Example**

answer()

```
For each snippet of text, label the sentiment of the text as positive or negative.

Text: watching `` ending '' is too often like looking over the outdated clothes and
    plastic knickknacks at your neighbor 's garage sale .
Sentiment: negative

Text: naipaul fans may be disappointed .
Sentiment: negative

Text: scott delivers a terrific performance in this fascinating portrait of a modern
    lothario .
Sentiment: positive

Text: you 'll probably love it .
Sentiment: positive

Text:a gob of drivel so sickly sweet , even the eager consumers of moore 's
    pasteurized ditties will retch it up like rancid creme brulee .
Sentiment:
```

Model Output

```
negative
```

---

## K.17 STORY CLOZE

*Description:* Commonsense reasoning task that requires choosing the correct ending to a four-sentence story. Mostafazadeh et al. (2017)
*Train Size:* 1871, *Test Size:* 1871

---

**Story Cloze Few Shot**

Input

```
Given two possible next sentences A) and B), choose the best next sentence to complete
    the story. Answer with A or B.

Peter was excited to go to the Sanders rally in New Hampshire. As Peter entered the
    arena it was full of thousands of people. When Peter saw Bernie he cheered as
    loudly as possible. He felt thrilled to be there.
A) He couldn't wait to vote for him.
B) He was a staunch republican.
Answer: He couldn't wait to vote for him.

My roommate was sick. She stayed home from work and school. She slept all day long. By
    the end of the day, she was feeling better.
A) She decided rest has helped.
B) She hoped she would soon be sick again.
Answer: She decided rest has helped.

My aunt is a nurse. She often talks about long hours at work. Last week was especially
    bad. She didn't have a single day where she didn't work late.
A) It was easy work.
B) It was hard work.
Answer: It was hardwork.

My friends all love to go to the club to dance. They think it's a lot of fun and
    always invite. I finally decided to tag along last Saturday. I danced terribly
    and broke a friend's toe.
A) My friends decided to keep inviting me out as I am so much fun.
B) The next weekend, I was asked to please stay home.
Answer:
```

Model Choices

```
- My friends decided to keep inviting me out as I am so much fun.
- The next weekend, I was asked to please stay home.
```

Gold Output

```
The next weekend, I was asked to please stay home.
```

---

**Story Cloze AMA** prompt()**-chain Example**

question()

```
Rewrite the statement as a yes/no question.

Statement: Jonathan Samuels was born in the 70's.
Question: Was Jonathan Samuels born in the 70's?

Statement: Jerry bullied him and called him names
Question: Did Jerry bully him and call him names?

Statement: Sam and jade were going to go to the movies
Question: Did did Sam and jade go to the movies?

Statement: Chocolate is tasty, when I am feeling hungry.
Question: Does chocolate taste good when you are hungry?

Statement: Mark ran fast.
Question: Did mark run fast?

Statement: The next weekend, I was asked to please stay home.
Question:
```

## Model Output

```
Did I stay home the next weekend?
```

## answer()

```
Passage: My aunt is a nurse and she often talks about long hours at work. Last week
    was especially bad and she was constantly working many hours.
Question: Was her work easy?
Answer: No, it was hard work.

Passage: My roommate was sick. She stayed home from work and school. She slept all day
      long and by the end of the day, she was feeling better.
Question: Did the rest help her?
Answer: Yes, she slept and felt better.

Passage: Andy had always wanted a big kids bike. When he turned six Year's old he
    asked for a bike for his birthday. He did not know how to ride a bike. On Andy's
    birthday his mother gave him a bike.
Question: Did he cry all night?
Answer: No, Andy was happy because he got a bike.

Passage: My friends all love to go to the club to dance. They think it's a lot of fun
    and always invite. I finally decided to tag along last Saturday. I danced
    terribly and broke a friend's toe.
Question: Did I stay home the next weekend?
```

## Model Choices

```
- My friends decided to keep inviting me out as I am so much fun.
- The next weekend, I was asked to please stay home.
```

## Gold Output

```
The next weekend, I was asked to please stay home.
```

### K.18   WSC

*Description:* Task that requires readining a sentence with a pronoun and selecting the referent of that pronoun from a list of choices. Wang et al. (2019)
*Train Size:* 554, *Test Size:* 104

**WSC Few Shot**

### Input

```
Passage: Mark was close to Mr. Singer 's heels. He heard him calling for the captain ,
     promising him, in the jargon everyone talked that night, that not one thing
     should be damaged on the ship except only the ammunition, but the captain and all
     "his" crew had best stay in the cabin until the work was over.
Question: In the passage above, does the pronoun "his" refer to Mark?
Answer: No

Passage: Tom gave Ralph a lift to school so "he" wouldn't have to walk.
Question: In the passage above, does the pronoun "he" refer to Ralph?
Answer: Yes

Passage: This book introduced Shakespeare to Ovid ; it was a major influence on "his"
     writing.
Question: In the passage above, does the pronoun "his" refer to Shakespeare?
Answer: Yes

Passage: The large ball crashed right through the table because "it" was made of
     styrofoam.
Question: In the passage above, does the pronoun "it" refer to the table?
Answer:
```

### Gold Output

```
Yes
```

### WSC AMA prompt()-chain Example

question()

```
Extract the phrase containing the pronoun.

Passage: Jane's mom went to the shop to buy Jane a backpack for "her" first day of
     kindergarten.
Extract: phrase containing "her": "her" first day

Passage: The musicians performed in the park and the crowd loved "them". The crowd
     cheered for them.
Extract: phrase containing "them": crowd loved "them"

Passage: Jeff gave his son some money because "he" wanted to buy lunch.
Extract: phrase containing "he": "he" wanted to buy

Passage: The large ball crashed right through the table because "it" was made of
     styrofoam.
Extract: phrase containing "it":
```

### Model Output

```
was made of styrofoam
```

question()

```
Rewrite the input as a question.

Input: it was made of glass
Question: What was made of glass?

Input: they are funny
Question: Who or what are funny?

Input: he drowned
Question: Who drowned?

Input: wrap around them
Question: Wrap around who or what?

Input: his cat is black
Question: Whose cat is black?

Input: laugh at them
Question: Laugh at who?

Input: her friend jennfier
Question: Whose friend Jennifer?

Input: it was made of styrofoam
Question:
```

## Model Output

```
What was made of styrofoam?
```

## answer()

```
Answer the question.

Passage: Jane's mom went to the shop to buy Jane a backpack for her first day of
    kindergarten.
Question: Whose first day?
Answer: Jane

Passage: Mrs. Jenna told Fred she loved him.
Question: Who loved him?
Answer: Mrs. Jenna

Passage: Joe gave Mark some money so he could buy lunch.
Question: Who could buy lunch?
Answer: Mark

Passage: The large ball crashed right through the table because it was made of
    styrofoam.
Question: What was made of styrofoam?
Answer:
```

## Gold Output

```
True
```

### K.19   WEBQUESTIONS (WQ)

*Description:* Question answering dataset with questions that can be answered using Freebase, a large knowledge graph. Berant et al. (2013)
*Train Size:* 3778, *Test Size:* 2032

**WebQuestions (WQ) Few Shot**

**Input**

```
Question: what character did natalie portman play in star wars?
Answer: Padme Amidala

Question: what country is the grand bahama island in?
Answer: Bahamas

Question: who does joakim noah play for?
Answer: Chicago Bulls

Question: who is governor of ohio 2011?
Answer:
```

**Gold Output**

```
John Kasich
```

**WebQuestions (WQ) AMA prompt()-chain Example**

question()

```
Produce distinct questions.

Question: who plays Carrie Bradshaw in sex and the city?
Answer: Caroline "Carrie" Bradshaw is a fictional character from the HBO franchise Sex
     and the City, portrayed by Sarah Jessica Parker.

Question: what are the elements in air?
Answer: By mole fraction (i.e., by number of molecules), dry air contains 78.08%
     nitrogen, 20.95% oxygen, 0.93% argon, 0.04% carbon dioxide, and small amounts of
     other gases

Question: what is HP company?
Answer: HP Inc. is an American multinational information technology company
     headquartered in Palo Alto, California, that develops personal computers (PCs)

Question: when was the last season of FRIENDS released?
Answer: The series finale aired on May 6, 2004, and was watched by around 52.5 million
     American viewers, making it the fifth-most-watched series finale in television
     history

Question: who is governor of ohio 2011?
Answer:
```

**Model Output**

```
John Kasich is the current governor of Ohio.
```

answer()

```
Answer the question.

Context: The nearest airport to Palm Springs is Indio/Palm Springs (PSP) Airport which
    is 2.1 miles away.
Question: what airport is closest to palm springs?
Answer: Palm Springs International Airport

Context: Martin Luther King earned his Bachelor of Divinity degree from Crozer
    Theological Seminary, followed by a doctorate in Systematic Theology from Boston
    University.
Question: what degree did martin luther king get?
Answer: Bachelor of Divinity

Context: The Niger river runs in a crescent through Libya, Mali, Niger, on the border
    with Benin and then through Nigeria.
Question: what countries does the niger river flow through?
Answer: Libya

Context: Puerto Rico is a territory of the United States and uses the U.S. dollar.
Question: what type of currency is used in puerto rico?
Answer: United States dollar

Context: kitt was voice most often by William daniels.
Question: who played kitt in knight rider?
Answer: William Daniels

Context: John Kasich is the current governor of Ohio.
Question: who is governor of ohio 2011?
Answer:
```

### Gold Output

```
John Kasich
```

## K.20   WIC

*Description:* Word sense disambiguation task cast as binary classification over sentence pairs. Wang
et al. (2019)
*Train Size:* 5428, *Test Size:* 638

**WiC Few Shot**

Input

```
These aspects of civilization do not find expression or receive an interpretation.
Find the product of two numbers.
Question: Is the word 'find' used in the same sense in the two sentences above? No

Cut my hair.
Cut the engine.
Question: Is the word 'cut' used in the same sense in the two sentences above? No

The pit floor showed where a ring of post holes had been.
The floor of a cave served the refugees as a home.
Question: Is the word 'floor' used in the same sense in the two sentences above? Yes

An emerging professional class.
Apologizing for losing your temper, even though you were badly provoked, showed real
    class.
Question: Is the word 'class' used in the same sense in the two sentences above?
```

### Gold Output

```
no
```

**WiC AMA** prompt()**-chain Example**

answer()

```
Give synonyms of the word in the sentence.

In "She heard the sound of voices in the hall.", synonyms for the word "sound" are:
- noise

In "Enter the secret code.", synonyms for the word "code" are:
- password

In "She acted in a play on Broadway", synonyms for the word "play" are:
- show

In "An emerging professional class.", synonyms for the word "'class'" are:
```

Model Output

