# OpenReview forum: "Ask Me Anything: A simple strategy for prompting language models"
_ICLR.cc/2023/Conference — ICLR 2023 notable top 25%_

### Official Review · Reviewer_SsWm · 2022-10-23

**Confidence:** 3
**Correctness:** 3
**Technical Novelty And Significance:** 3
**Empirical Novelty And Significance:** 4
**Recommendation:** 6

**Clarity, Quality, Novelty And Reproducibility:**

The proposed method is novel in that it is effective and scalable. However, this paper contains some unclearness, especially on weak supervision and how the predictions are mapped to the output space. See Weakness part for details.

**Strength And Weaknesses:**

Strength:
1. The improvement of AMA is significant (GPT Neo 6B outperforming GPT-3 175B). Also, the experiments are extensive, evaluating on various model families such as OPT and BLOOM. This shows the effectiveness of AMA is universal.
2. This approach is scalable since AMA uses LLM itself to generate open-ended questions.

Weakness:
1. To apply AMA in real use cases, AMA needs 1000 training examples from the training set of the target task (specified in Appendix A.3.). This is an unrealistic setting because only a few training instances might be present depending on the task. What if only a few training examples are present for a task (similar to few-shot setting)? Also, we need to train the probabilistic graphical model for each task. This introduces additional latency for task adaptation.
2. For the AMA prompts on Appendix H, it seems that some tasks (DROP, SST, BoolQ, COPA) only do either prompt() or answer() stage, not both. How can we decide "automatically" when to do only prompt() or answer() or both depending on the target task?
3. This paper is quite unclear on how the predictions are mapped to the output space. Are the predictions mapped to the output space using a predefined rule? (ex) no -> False for Figure 1)
4. Are there any results on a large-scale LM evaluation (175 OPT, GPT, BLOOM, etc,) comparing using AMA or not on the evaluation setting of Table 1 and 2?
2. The structure of Section 3.2 and Section3.4 is divided into observation and solution (AMA). However, the observation experiments are not extensive enough. Is there a reason for selecting 3 tasks (CB, WSC, RTE)? Observing the findings on more tasks would make the findings more general and the motivation of AMA stronger.
5. I think that StoryCloze should be included in Natural Language Understanding instead of NLI for Table 1 and 2.


**Summary Of The Paper:**

This paper proposes a simple strategy to prompt language models: Ask Me Anything. From the findings that open-ended questions outperform restrictive prompts, AMA first encourages the LMs to generate open-ended questions, which is a scalable approach. From a few open-ended questions generated by the model, the LM answers the question for each prompt chain. From a set of different predictions, AMA uses weak supervision to aggregate the noisy answers. The result shows that AMA applied on 6B GPT-Neo outperforms 175B GPT-3.

**Summary Of The Review:**

This paper proposes Ask Me Anything (AMA), a simple prompting method that outperforms LMs that are 30x times bigger. By making the LMs generate open-ended questions and aggregating predictions based on weak supervision, AMA is scalable and shows improved performance. Although I recommend this paper be accepted, the authors should answer and clarify the questions on the weakness part (especially Question 1 and 2).

---

> ### Author Response · Authors · 2022-11-15
> **Response Overview and Weak Supervision Clarifications**
>
> Thank you for your review! We appreciate that you find the improvements from AMA to be significant and scalable, and the experiments to be exhaustive.
>
> We provide responses to your questions and concerns below. We include (1) clarifications and new experiments on weak supervision in AMA, (2) several new experiments to bolster the observations in Section 3 including a new analysis of the Pile pre training corpus (Section 3.2, Appendix H, Appendix B), (3) the AMA results on large scale models (Appendix B.2), and (4) clarifications to your questions about particular steps of the end-to-end AMA procedure.
>
> # AMA weak supervision
> To examine the role of additional unlabeled training examples and the latency of learning the graphical model, we include new ablations over three tasks (RTE, WSC, and AGNews) with 5 random seeds per task.
>
> ***Training data:***
> ***We clarify that we do not use labeled training data to perform weak supervision.*** We only utilized the training data’s x inputs and AMA model outputs to reduce the noise in the estimation of the WS graphical model parameters (the aggregation step at inference uses these parameters but is only performed on the fixed test dataset). We run AMA with 100%, 50%, 20%, 10%, and 0% of the 1000 training examples and have updated Appendix B.5 with our results. We find that even without any of the additional unlabeled examples, AMA’s average accuracy does not decrease on WSC or AGNews, and only decreases by 0.4 points on RTE, still outperforming GPT3-175B few shot. This suggests that the additional unlabeled examples are not necessary for AMA’s performance.
>
> ***Latency of training the probabilistic graphical model:***
> Here we report on the time required to train the probabilistic graphical model, averaging across the three tasks (RTE, WSC, and AGNews). The weak supervision algorithm for learning the probabilistic graphical model takes 13.0 seconds without modeling dependencies and 84.3 with modeling dependencies on average (across 5 random seeds per task). As a point of comparison, running inference with the 6B parameter GPT-J model on a single A100 GPU takes the following amounts of time (excluding the time to load the model). The weak supervision step is relatively fast. We have updated Appendix B.5 with this information.
>
> | Benchmark | Number of Examples | Total Inference Cost with AMA prompt chains (seconds) |
> | --------- | ------------------ | ----------------------------------------------------- |
> | RTE       | 277                | 8310                                                  |
> | WSC       | 104                | 3141                                                  |
> | AGNews    | 7600               | 53200                                                 |

---

> > ### Author Response · Authors · 2022-11-15
> > **Additional Experiments for AMA with Large Models, and to Support Section 3 Observations**
> >
> > # AMA Results at Large Model Scales
> > Thanks for raising the question of AMA performance at large scales. Yes, we evaluate AMA with 175B OPT and 175B BLOOM, the available open-source LMs, and reported the lift in Figure 5a. We also include the 20B EleutherAI and 175B BLOOM models in Figure 4.
> >
> > To improve the clarity and transparency of these results, we additionally add Appendix B.2, which contains a table of the raw results by task on the large-scale models. Overall, we find AMA to be effective across model scale. We include the table from Appendix B.2 below as well.
> >
> > | Task    | BLOOM 175B Few Shot | BLOOM 175B Ask Me Anything | OPT 175B Few Shot | OPT 175B Ask Me Anything |
> > | ------- | ------------------- | -------------------------- | ----------------- | ------------------------ |
> > | ANLI R1 | 37.3                | 37.4                       | 37.9              | 37.9                     |
> > | ANLI R2 | 34.4                | 36.1                       | 37.2              | 37.2                     |
> > | RTE     | 57.4                | 66.5                       | 53.8              | 67.7                     |
> > | CB      | 64.3                | 82.1                       | 64.3              | 81.0                     |
> > | WiC     | 51.7                | 59.4                       | 51.9              | 58.4                     |
> > | WSC     | 68.3                | 71.8                       | 49.0              | 76.0                     |
> > | SST2    | 45.6                | 95.2                       | 55.6              | 95.8                     |
> >
> > # Experiments on the Observations in Section 3
> > You raised a concern that the observations in Section 3 are not demonstrated on a sufficient number of tasks. Thank you for raising this concern! We have included extended results on 4 NLU tasks, 3 NLI tasks, and 2 Classification tasks to address this point. These experiments investigate the effects of performing the aggregation step, without any prompt reformatting, and compare restrictive and open-ended prompt formats.  We find that the original findings hold true. Please find the new results and discussion in Appendix B.4.
> >
> > | Benchmark | GPT-J Few-Shot (k=3) Average | GPT-J Few-Shot (k=3) Majority Vote | GPT-J Few-Shot (k=3) Weak Supervision | GPT-J AMA |
> > | --------- | ---------------------------- | ---------------------------------- | ------------------------------------- | --------- |
> > | CB        | 23.8                         | 17.9                               | 50.0                                  | 83.9      |
> > | RTE       | 53.5                         | 53.1                               | 54.2                                  | 75.1      |
> > | WSC       | 46.2                         | 38.5                               | 38.5                                  | 77.9      |
> > | COPA      | 80.0                         | 81.0                               | 81.0                                  | 84.0      |
> > | Amazon    | 61.9                         | 62.4                               | 62.5                                  | 68.2      |
> > | AGNews    | 70.3                         | 70.7                               | 75.0                                  | 86.4      |
> > | ANLI-R1   | 33.4                         | 33.5                               | 33.5                                  | 37.8      |
> > | ANLI-R2   | 33.2                         | 32.9                               | 32.2                                  | 37.9      |
> > | ANLI-R3   | 35.4                         | 36.5                               | 34.6                                  | 40.2      |
> >
> > Next, we provide a **new analysis of why the QA prompt template may be effective** by investigating the popular Pile pretraining corpus, which is used to train the EleutherAI models. Please find the new results and discussion in Section 3.2 and in Appendix H.

---

> > > ### Author Response · Authors · 2022-11-15
> > > **Responses to Additional Questions**
> > >
> > > **Automatically deciding whether to do question(), answer(), or both.**
> > > This is a very interesting question!
> > >
> > > In light of your question, we ran a new analysis where we apply a question() prompt, which contains demonstrations that convert statements to questions, to text that is already a question. I.e. the inference text is mismatched with the in-context demonstrations, which are all statements and not questions. We repurpose our diagnostic question-generation tasks for this experiment. We find that the model generates back the input question achieving a RougeL score of 98.1. This suggests, that the question() prompts behave as an identity function for tasks that already contain questions. Therefore, we may not need to insert a complicated strategy for deciding whether a task requires a question() step! Meanwhile, all tasks require an answer step.
> > >
> > > Complementing your original question, another exciting question is how to apply the question() and answer() prompts to the different task formats. This is because tasks come in very different formats.
> > > - COPA/StoryCloze contain context and require determining which of two provided statements is the better continuation to the context.
> > > - NLI tasks require determining the validity of one input statement, given context.
> > >
> > > For instance, in our work we convert each of the two statements in the COPA and StoryCloze tasks to questions independently of one another, whereas the NLI tasks just require converting one input statement to a question. Automatically inferring the task format to make these decisions is an exciting direction for future work.
> > >
> > > **Mapping Predictions to the Output Space**
> > > We map predictions to the outputs space using an **exact match** between the generated answer and gold label. If the model outputs “Science” on a task where the valid output class is “Science and Technology” (i.e. in AGNews), this would be considered **incorrect; we use no specialized verbalizers**. The only modification is treat the outputs “true” and “yes” as equivalent in the exact match computation. Note that we do not use rank-classification (i.e. outputting the answer with the maximum logits amongst pre-provided answer choices) unless the task is a multiple-choice task; rather, we have the model generate text in an unconstrained fashion. Overall, the mapping is simple and natural. Our evaluation code for all 20 benchmarks is included in the supplementary code submission.
> > >
> > > ***We also add Appendix D, which details the end to end AMA procedure with algorithmic sketches.***
> > >
> > > **StoryCloze**
> > > Thank you! We have updated the categorization of StoryCloze to be in the NLU category in Tables 1 and 2.
> > >
> > > Thank you again for your review!

---

> ### Author Response · Authors · 2022-12-06
> **Dear reviewer SsWm: we'd love to know if you have any more questions after our response**
>
> Dear reviewer SsWm,
>
> Thank you very much for your helpful feedback and suggestions, they helped us to improve the paper. We tried to carefully address all of your comments in our response and the updated paper. Please let us know if you have any further questions, and we are very happy to follow up!
>
> Thank you for your time!

---

> ### Author Response · Authors · 2022-12-12
> **Dear reviewer SsWm: we'd love to know if you have any more questions after our response**
>
> Dear reviewer SsWm,
>
> As today is the final day of the discussion period, we want to check if you have any further questions or found our responses helpful?
>
> Thank you again for your time!

---

### Official Review · Reviewer_CQxp · 2022-10-24

**Confidence:** 3
**Correctness:** 4
**Technical Novelty And Significance:** 2
**Empirical Novelty And Significance:** 2
**Recommendation:** 6

**Clarity, Quality, Novelty And Reproducibility:**

# Comments

It would be helpful to have finetuning baselines. For example Brown et al mention such baselines for BoolQ (91), CB (96.9), etc. in their Table 3.8. Others: DROP (89.1), NQ (44.5), WebQS (45.5). Sometimes the improvement over GPT-3 is great but still far behind finetuning.

# Related Work

Self-consistency is a very crucial baseline / related work. Unlike the text says, it does not require additional training. It is an insightful innovation that plurality vote is effective when you have different reasoning paths, which runs counterintuitive for how samples from a standard prompt behave.

Another related work with diverse prompting is Li et al. On the Advance of Making Language Models Better Reasoners. https://arxiv.org/abs/2206.02336

There is a lot of other works that describe strategies for exemplar selection. One particular relevant example is Shi et al. (https://arxiv.org/pdf/2204.11454), which found aggregating 5 prompts (3 distinct exemplars each) is better than single prompt with 15 exemplars.

There are other works that do sequential prompting like least-to-most and SeqZero.

# Minor Comments

The analysis / diagnostic in Fig 5 is great, and builds confidence that sub-tasks are related to final performance.

It was not always clear to me what is the source of weak supervision... Are you using the labels from the task, and if so, isn't this standard supervision? Are the prompts themselves considered the source of noise?

It was not immediately clear to me what was meant by "highly-correlated outputs".

nit: Are you taking majority vote or plurality vote?

nit: The framing of perfect vs. imperfect prompts is not the strongest. Perhaps it is sufficient to reframe and say you are introducing an efficient and effective ensembling method?

**Strength And Weaknesses:**

# Strengths

- Consistent improvements over the base model when using the new aggregation.

- There is useful analysis: the observations related to dependencies between prompts, the diagnostic, etc. Including many details throughout the methods section motivating the approach.

- The extensive results and appendix will be valuable to others using prompting.

# Weaknesses

- I found the story line a little confusing. The aggregation based on weak supervision is touted as main accomplishment, but this actually only gives a small improvement in Table 1 (but this small push does give edge over GPT-3 sometimes). In many of the tasks the 6B model already beats the 175B without weak supervision based aggregation. Also this table is not really apples-to-apples comparison making it hard to interpret. Should make it more clear what is the goal in this table --- is GPT-3 simply used more like a reference than a baseline? Also, is it fair to compare against GPT-3 like this when so many other prompting techniques have been developed since release?

- To explain more previous weakness, I was not clear when reading if it is enough to do prompt re-formatting or aggregation is also necessary. I think still I am a little not sure, also about whether re-formatting is necessary for aggregation to work well. In general, it could help to tailor overall story as being primarily about prompt re-formating or aggregation.

- It is well known that aggregating can be effective for prompting, and there are existing techniques like (self-consistency combined with chain of thought) that typically yield good improvements. Perhaps worth using those as baselines, or mentioning why they weren't used. Maybe CoT is not effective for smaller models?



**Summary Of The Paper:**

The authors present a technique for aggregating predictions from multiple prompts inspired by weak supervision. They exploit the idea that there is a dependency graph labels, errors, and prompts, and use this to train a mapper from raw prompt output to a final predictions competing mostly against a simple voting baseline.

For the most part, their technique for aggregating prompt predictions is treated as a black box. Also, there is another aspect of their technique: they transform input examples into a special format for their prompt. Presumably, this transformation works better for aggregation.

To me, this is primarily an empirical study, and the main contribution is strong results showing their approach enables a 6B param model to outperform a less advanced version of a 175B param model.

**Summary Of The Review:**

To me, the work is valuable because it provides some useful results and extensive analysis in a popular topic, and probably more so because it is with an open source model. Simultaneously, it almost seems like a combination of a technique from weak supervision and prompting, and does not provide a surprising insight (since others have done something similar before regarding prompt reformatting and aggregation, and the aggregation method is not new), although perhaps I have missed some key point about why this is particularly relevant for the included models. I wish the story was more clean with respect to whether the paper is about analysis alone, presenting a new aggregation method, or prompt reformatting. Or maybe the bigger point is that we do not need big models and can simply get the same performance by fixing small models. Also, I wish there was more analysis about dependencies between prompts that inspired the aggregation method --- this did not give as big an improvement in Table 1 as prompt reformatting AFAICT, but I found this to be an interesting point of the paper.

---

> ### Author Response · Authors · 2022-11-15
> **Response Overview and Contribution Clarifications**
>
> Thank you for your thoughtful review! We appreciate that you find the results strong and extensive, analysis useful, and overall method useful for those using prompting.
>
> Here we address your questions and concerns. We include (1) clarifications around our contributions, (2) clarifications and several new-experiments to show the roles of prompt re-formatting vs. aggregation/ weak supervision in AMA (Appendix B, Appendix H), (3) additions to the related work in the revised manuscript (Section 2), and (4) new experimental baselines with Self-Consistency (Appendix B.6).
>
> **Clarifying our contributions**
> Thank you for your questions around the contributions! We provide a response to your concerns in the main thread / general response. Please let us know if this is helpful!
>
> We clarify that our work finds that both reformatting the prompts and using weak supervision for aggregation improve model performance and is not only focused on the latter aggregation step. The QA prompt re-formatting approach we propose was not previously demonstrated in the literature and is also a contribution we motivate and validate. Accordingly, as you note in Table 1, both the QA prompt-reformatting and the aggregation are complementary. Sometimes, reformatting provides most of the gain (e.g. CB, WSC, DBPedia, AGNews), sometimes aggregation plays a large role (e.g. RTE, BoolQ, MultiRC, RealTimeQA) and sometimes both give comparable improvements (e.g., Amazon, the three ANLI tasks).
>
> **Why both re-formatting and aggregation?**
> Thank you for raising the concern that it is not clear whether both re-formatting and aggregation should be included in the same paper!
> First, in scalably transforming thousands of task inputs across diverse tasks, it is difficult to audit all the transformations and the process can be quite noisy. Aggregation is useful to manage this noise.
>
> Second, in Section 3.1 Simple Baseline, we discuss the results of aggregation without re-formatting to motivate that re-formatting is also beneficial. In light of your comments, we run ***new experiments*** across a larger number of tasks, representing the range of task-types in Table 1. Specifically, we use different few-shot demonstrations to construct prompts, generate outputs using each prompt, and aggregate the results using majority vote and weak-supervision. I.e. these experiments use aggregation, but no re-formatting. We find that aggregation alone leaves significant performance gaps. As you mention, aggregation alone is useful compared to the average performance, however we find that aggregation and re-formatting are both required in AMA to yield an effective and efficient prompting solution.
>
> Finally, we hope the work helps show that more effective prompting strategies can enable the use of far smaller models, especially for users who cannot afford to use the large scale models. Both techniques are required to achieve this as demonstrated in the experimental results.
>
> **Table 1: Additional Results of Aggregation without Reformatting**
> | Benchmark | GPT-J Few-Shot (k=3) Average | GPT-J Few-Shot (k=3) Majority Vote | GPT-J Few-Shot (k=3) Weak Supervision | GPT-J AMA |
> | --------- | ---------------------------- | ---------------------------------- | ------------------------------------- | --------- |
> | CB        | 23.8                         | 17.9                               | 50.0                                  | 83.9      |
> | RTE       | 53.5                         | 53.1                               | 54.2                                  | 75.1      |
> | WSC       | 46.2                         | 38.5                               | 38.5                                  | 77.9      |
> | COPA      | 80.0                         | 81.0                               | 81.0                                  | 84.0      |
> | Amazon    | 61.9                         | 62.4                               | 62.5                                  | 68.2      |
> | AGNews    | 70.3                         | 70.7                               | 75.0                                  | 86.4      |
> | ANLI-R1   | 33.4                         | 33.5                               | 33.5                                  | 37.8      |
> | ANLI-R2   | 33.2                         | 32.9                               | 32.2                                  | 37.9      |
> | ANLI-R3   | 35.4                         | 36.5                               | 34.6                                  | 40.2      |

---

> > ### Author Response · Authors · 2022-11-15
> > **Aggregation in AMA, the role of weak supervision.**
> >
> > ***You also raised the interesting question of when re-formatting, aggregation, or both are necessary.*** We motivate when each is necessary in the Section 3 observations. First, in Section 3.2, we find re-formatting is beneficial when the original prompt is restrictive rather than open-ended. We hypothesize that the open-ended prompts are better aligned with the next-token prediction pre-training objective. We also include new analysis of the pretraining data in Appendix H, to motivate the re-formatting. Second, in Section 3.3, we find that aggregation is beneficial because prompts can have varying accuracies and failure modes.
> >
> > We hope this explanation and the new results we provide demonstrate to you that the re-formatting and aggregation approach are both important and complementary to achieving the 175B level quality.
> >
> > **The role of weak supervision**
> > We agree that the weak supervision method we use and the idea of aggregating prompt outputs are not new, and we discuss key related works in the Introduction and Section 2, including [4, 5] and Self-Consistency. Instead, our novelty in the aggregation step is the application of a particular weak supervision technique to prompting, motivated by our observations that the prompt outputs (whether from the ones we write, or from publicly available prompts from PromptSource) have varying accuracies and dependencies. As weak supervision aims to learn the true model of accuracies and correlations among the prompt-chain outputs, it provides a more reliable aggregation method whose performance always matches or exceeds that of more heuristic baseline aggregations, like majority vote (which Self-Consistency uses) or picking the single best prompt. On the benchmarks, we show weak supervision can achieve up to 8.7 points of lift over majority vote and does no worse than majority vote on 16 of 20 tasks (Section 5.3). Our results in Table 4 further reflect how weak supervision matches or exceeds baselines.
> >
> > We further report on how to apply weak supervision to the setting -  we found that applying standard WS frameworks like Snorkel [6], Data Programming [7], and FlyingSquid [8] did not do well because they either do not handle dependencies or do not capture variation among class-conditional accuracies. We therefore looked at the Snorkel MeTaL framework [9], which is expressive enough to handle both the dependencies and class-conditional accuracies we observe from the prompt outputs. However, it requires the users to specify the dependencies. We found that heuristics like simply looking at which pairs of prompts have highly correlated output was insufficient, as correlation does not imply the mathematical definition of conditional dependence of random variables. We therefore used a structure learning algorithm [10] to recover the exact conditional dependencies to be used as input into the MeTaL framework. Lastly, we found that MeTaL always modeled abstains in the prompt outputs, even in cases where there weren’t any. We had to make non-trivial changes to the original MeTaL loss function to enforce this. We hope our findings here are insightful to the community as prompt aggregation methods are explored more.
> >
> > **Analyzing the dependencies between prompts**
> > You raised the question of analyzing the dependencies between prompts, which we use to motivate our aggregation approach. We clarify that in Section 3.3, we run ablations over three tasks, in which we compute the Jaccard indices, which measure the overlap in the sets of examples on which the model makes errors with different prompts. We show the Jaccard indices are much higher than if the prompts were to make independent errors, suggesting that prompt outputs are dependent on each other.

---

> > > ### Author Response · Authors · 2022-11-15
> > > **Additional Baseline Results and Related Work**
> > >
> > > ### Self-Consistency Results and Related Works
> > > **Additional aggregation baseline (Self-Consistency with Chain of Thought Prompting)**
> > > Thank you for raising the concerns around Self-Consistency! We apologize for mis-representing Self-Consistency as using training in Section 2. In that citation, we mean to refer to the Learned Verifiers [1] and STaR [2] works, which generate multiple completions and fine-tune to achieve improvements with small models (<10B).
> > >
> > > We have updated the references in the revision and include new results below and in Appendix B.6, comparing AMA the tasks on which the Self-Consistency work evaluates using the prompts in the Self-Consistency paper appendix and temperature-based sampling. Thank you for making the paper stronger!
> > >
> > > | Baseline                               | Aggregation over # Outputs | RTE Task | ANLI-R1 Task | BoolQ Task |
> > > | -------------------------------------- | -------------------------- | -------- | ------------ | ---------- |
> > > | Self-consistency with Chain-of-Thought | 5                          | 47.3     | 33.4         | 63.1       |
> > > | Ask Me Anything                        | 5                          | 75.1     | 37.8         | 67.2       |
> > >
> > > We originally did not compare to Self-Consistency, because the results both in this work and in the Chain of Thought [3] work did not demonstrate large gains in accuracy for the <10B model scale. We see similar takeaways in the above table.
> > >
> > > **Related works**
> > > Thank you for pointing out the related works! We have addressed the Self-Consistency point by adding new comparisons and fixing the discussion in the paper.
> > >
> > > We discuss paper from Li et al. as well as add discussion on related works from Cobbe et al. [1] and Zelikman et al. [2] in Section 2, thank you for the pointers!
> > >
> > > We add discussion on works describing strategies for exemplar selection in Section 2, including the work from Shi et al.
> > >
> > > We add the pointers (Least-to-Most and SeqZero) to the related works. In our original manuscript, we discussed PromptChains, AIChains, Selection-Inference, Maieutic Prompting, and other sequential prompting works, and agree with you that there are related works here and thanks so much for the additional pointers!
> > >
> > > ### Baselines in Table 1
> > > **Finetuning baselines**
> > > Thank you for the suggestion on adding finetuning baselines! We agree that this is a useful baseline to contextualize the results and have added them to the main table in our revision.
> > >
> > > **Clarifying the GPT3-175B baseline and takeaways of Table 1**
> > > Thank you for asking about our chosen baseline, GPT-3 175B few-shot performance and the takeaways from Table 1!
> > >
> > > In Table 1, the key message we seek to demonstrate is that LMs can see significant quality gains if we pay attention to how we prompt them. In the table, GPT-J few-shot and GPT-3 few-shot reflect the off-the-shelf gap using the same prompt formats – this is an apples-to-apples comparison. The columns in between show the effect of reformatting (QA) and aggregation (QA + WS) respectively to demonstrate how the gap closes on a task-by-task basis.
> > >
> > > Beyond this, the GPT3-175B OpenAI models are widely used in prompting work and the baselines from the Brown paper are often included for reference. They perform well off-the-shelf and on a broad variety of tasks. We have added a note to address this in Section 5.
> > >
> > > As you note, there have been other prompting proposals, but none closes the gap between small and large LMs, especially so on the broad set of tasks we evaluate. You note that the version of the 175B parameter model we use is less advanced. We clarify that we report the GPT3-175 **few shot** numbers and not the zero-shot baseline, providing a competitive baseline. Overall, this is an apples-to-apples comparison between LMs of drastically different sizes, with no additional fine-tuning, and only varying the prompting strategy. We do not report the Instruct fine-tuned GPT3 models because these are varying other knobs besides the prompting technique, and are closed source artifacts of OpenAI at the time of this work. We also note that we evaluate the BLOOM and OPT 175B parameter models (Figure 5 and Appendix B.2).

---

> > > > ### Author Response · Authors · 2022-11-15
> > > > **Responses to Minor Comments and References**
> > > >
> > > > **Responses to minor comments**
> > > > 1. Source of weak supervision: the prompt outputs themselves are the noisy labels that are the sources of supervision in weak supervision. We do not use any true task labels.
> > > > 2. Highly-correlated outputs: we are referring to the observation that pairs of prompts have very similar outputs on the same set of data (e.g. highly overlapping sets of errors), and thus exhibit dependencies among each other. We will clarify this in our draft.
> > > > 3. Majority vs. plurality: we take plurality vote in the case that the task output is multiclass.
> > > >
> > > > **References**
> > > > [1] Cobbe et al. Training Verifiers to Solve Math Word Problems. arXiv:2110.14168. 2021.
> > > > [2] Zelikman et al. STaR: Self-Taught Reasoner Bootstrapping Reasoning With Reasoning. arXiv:2203.14465. 2022.
> > > > [3] Wei et al. Chain of Thought Prompting Elicits Reasoning in Large Language Models. 2022.
> > > > [4] Jiang et al. How can we know what language models know? TACL 2020.
> > > > [5] Schick and Schütze. It’s not just size that matters: Small language models are also few-shot learners. arXiv:2009.07118v2 2021.
> > > > [6] Ratner et. al. Snorkel: Rapid Training Data Creation with Weak Supervision. VLDB 2017.
> > > > [7] Ratner et. al. Data Programming: Creating Large Training Sets, Quickly. NeurIPS 2016.
> > > > [8] Fu et. al. Fast and Three-rious: Speeding Up Weak Supervision with Triplet Methods. ICML 2020.
> > > > [9] Ratner et. al. Training Complex Models with Multi-Task Weak Supervision. AAAI 2019.
> > > > [10] Varma et. al. Learning Dependency Structures for Weak Supervision Models. ICML 2019.
> > > >
> > > > Thank you again for your review!

---

> ### Author Response · Authors · 2022-12-06
> **Dear reviewer CQxp: we'd love to know if you have any more questions after our response**
>
> Dear reviewer CQxp,
>
> Thank you very much for your helpful feedback and suggestions, they helped us to improve the paper. We tried to carefully address all of your comments in our response and the updated paper. Please let us know if you have any further questions, and we are very happy to follow up!
>
> Thank you for your time!

---

> > ### Comment · Reviewer_CQxp · 2022-12-12
> > **Thanks**
> >
> > Thank you for the detailed response. I find the clarifications helpful, although I will keep my original score.

---

### Official Review · Reviewer_Pytd · 2022-10-24

**Confidence:** 4
**Correctness:** 3
**Technical Novelty And Significance:** 3
**Empirical Novelty And Significance:** 3
**Recommendation:** 8

**Clarity, Quality, Novelty And Reproducibility:**

Clarity: ok
I really found this paper hard to follow. After reading the main paper the reader is left wondering as to what was actually proposed. The intro should really focus more on Figure 1 and highlight the overall algorithm better (generate the questions, generate the answers for these generated questions, aggregate the output).  I feel that Figure 1+ Appendix F should form a good chunk of the main paper, showing what happens, why you believe that questions and answers that were generated by AMA are of good quality.
Additionally, apart from "insights" as to what makes good questions, authors don't provide enough details of what is being generated. How many templates do you end up using? Is it task dependent? Were these templates selected for each dataset manually?
Finally, the weak supervision should be also described in the main paper, as it is an integral part of the approach.

Novelty: after having (finally) understood the approach, I do find it witty. Authors are generating a number of prompts that hopefully highlight different angles of the input, and combine the outputs of the model based on these prompts to come up with a final solution.

Reproducibility: see my comments on clarity. As far as I understand authors open source their code (I have not seen where???) so this should help reproducibility, but overall just by reading the paper, I don't think I would have enough details to reproduce the approach.

**Strength And Weaknesses:**

Pros:
-  impressive results (zero shot performance of smaller models is comparable or better than few shot of larger) on context dependent tasks
- the idea is really neat
cons:
- not flashed out enough to allow me to reproduce/code it up (without looking at authors' code)
- clarity (please see more detailed comments)
- less impressive results on closed book tasks
- not sure how much effort/tuning goes into prompt genereation and WS

**Summary Of The Paper:**

This paper focuses on zero-shot setting and looks into generating prompts that will improve zero shot performance. It has been known that LLMs behave very differently given even slightly different prompts, and a lot of effort can be spent finding the perfect prompt. Authors propose to, given the input, generate several "imperfect" prompts for this input and then aggregate the results. To generate the imperfect prompts, authors come up with a number of templates (question template and answer templates) that teach model to, given an input, generate the questions for the input and then to answer these questions. The final prediction is an aggregation of prediction from the answer step. The aggregation happens via weak supervision (the model for this is also learnt). The experiments show impressive improvements  for smaller models, allowing them to match that of larger models few shot performance.
Additionally, authors provide some insights as to what makes the good prompt. For example, they find that open ended prompts do better than prompts with suggested restricted output


**Summary Of The Review:**

Update: in light of additional comparison with prompt tuning, i am raising my score. I appreciate all the additional experiments authors ran. I still encourage to improve the flow/the story/the presentation, but the method deserves to be seem

Update: i am keeping my score of marginally above. I do think the narrative can be improved, but it is also an interesting peace of work and it is pretty witty to make the model generate the questions and answers to improve the prompting.

Overall It is a very good idea of forcing the model to write the prompts that it is able to respond to. I do think the main paper should be reworked to include more examples/explanations of what is being proposed plus WS part.

Additional questions
1) Are all the models you tested are next token (perfix) trained, not span corruption trained correct?
2) With respect to having less of improvement on closed book tasks - do you think it is because that the answers you show in the "anwers" part of the chain don't draw on closed book knowledge? I assume your "answers" templates are the same for various tasks so they don't include that closed book knowledge?
3) What is the complexity (time and space) of weak supervision?
4) You say in Table 1 that prompts can abstain from the predictions. What does it mean? Don't you get an output for each of your prompt chains?
5) Are questions prompts and answer prompts task specific? How did you create them
How many chains do you create for each input during inference?
Building on this
Isn't it the case that AMA performance will depend greatly on question() step? If the questions are not well done then when you create answer prompts, the model will not produce the result you are looking for
6) This seems to work well on simpler tasks (yes or no, question answering), but i am struggling to understand how the combining of the results will work for more generative tasks like summary tasks where variance of the predictions will be much higher

---

> ### Author Response · Authors · 2022-11-15
> **Response Overview, Analyzing AMA Question Quality, and Clarifications**
>
> Thank you for your review! We appreciate that you find the idea novel and interesting, and the results to be impressive.
>
> We seek to address your questions in our response below and updates to the paper! We include new experiments to answer your questions around the quality of LM-generated questions and several clarifications on the AMA procedure.
>
> ## Code Accessibility
> We provided the anonymized code zip-file in our submission and will release the code on Github for reproducibility. We will update the non-anonymous Github URL in the final paper.
>
> ## AMA Prompt Quality
> ***Why are the questions and answers AMA generated of good quality?***
> Thank you for this question! We conduct a new analysis the pretraining corpus to address this question; please find this in Section 3.2 of the main paper and in Appendix H.
>
> ***Isn’t AMA performance dependent on the question() step?***
> Thank you for the question! Yes, AMA performance is dependent on the quality of the questions and answers as you suggest. We release a diagnostic task (Section 5) to help users diagnose whether their models succeed on question generation. Our error analysis (Section 5 and Appendix ?) shows that using smaller models and generating questions from longer statements decreases the question quality.
>
> To further address your question on the effect of question-quality, we provide a new experiment in which we generate questions using a larger, higher-quality model (GPT-J-6B) and use those with the small model (GPT-Neo-1.3B) for the answer() step of AMA. We observe the following results, which further support as you say that AMA performance improves with higher quality questions (which we assume are generated by the larger model). This suggests that by training smaller models to be better question-generators in future work, we may see even larger performance lifts from AMA.
>
> | Baseline                                                                                  | WSC Task | CB Task | RTE Task |
> | ----------------------------------------------------------------------------------------- | -------- | ------- | -------- |
> | Original performance (GPT-Neo-1.3 generated questions and GPT-Neo-1.3 generated answers) | 61.5     | 62.5    | 64.7     |
> | Ablation performance (GPT-J-6B generated questions and GPT-Neo-1.3 generated answers)       | 67.3     | 64.3    | 66.7     |
>
> ## Clarifying the AMA Steps
> Thank you for raising your concerns about the unclear aspects of AMA. To address them, we add detailed algorithmic sketches and explanations in Appendices D and E for the prompt reformatting and weak-supervision aggregation steps respectively. From these algorithmic sketches, as well as the prompt demonstrations in Appendix K, we hope the reader can reproduce the results without referring to our implementation. Further, within the main paper, we include a new analysis of the Pile pretraining data to address why the QA prompt structure may be so effective. We also add more details about the weak supervision aggregation algorithm in section 3.4.
>
> We answer your specific questions about the procedure below:
>
> ***Were templates selected for each dataset manually?***
> The QA template is consistently used across tasks In the submission, we discuss that prompts consist of prompt-templates, which contain placeholders for prompt-examples.  The template or structure of the prompt-chains is the same for all tasks – we form open-ended questions and answer them.
>
> The same in-context examples (not task-specific) are reused to achieve our results on many tasks (ANLIR1, ANLIR2, ANLIR3, BoolQ, CB, COPA, DROP, RTE, StoryCloze). For other tasks, differences typically arise because the format of the prompt differs:
> - COPA/StoryCloze contain context and require determining which of two provided sentences is the better continuation.
> - NLI tasks require determining the validity of one input statement, given context.
> - Some tasks (e.g. NQ, WebQ) come with open-ended questions, but no context.
>
> Because the structure of the task differs, the types of questions the LM should generate can change, but for each task open-ended (non-restrictive) questions are obtained and then answered.
>
> ***What does it mean for prompts to abstain from predictions? Don’t we get an output for each prompt chain?***
> Suppose the model outputs “Science”, but there is no such class in the output space. We mark such invalid outputs as abstains.
>
> ***Clarifying the AMA Aggregation Step’s complexity***
> Given k prompts each producing an output on n examples, weak supervision has a runtime complexity of $O(kn^2)$ (computing the empirical covariance matrix of the $k$ prompts) and a space complexity of $O(kn)$ (the matrix of all prompt outputs). Note that weak supervision uses SGD to learn the aggregation weight parameter on each prompt but is not training the LM itself. The weak supervision algorithm also has theoretical guarantees, learning the true output with error rate $O(\frac{1}{\sqrt{(n)}})$.

---

> > ### Author Response · Authors · 2022-11-15
> > **Additional Questions**
> >
> > **Performance on Specific Task Types**
> > Closed-book tasks: Thanks for asking about the closed-book tasks, we are very happy to discuss this point further! All tasks in the paper rely only on the latent knowledge the language model memorized during its pretraining stage, and no additional knowledge beyond what is provided in the raw task inputs. Note that some tasks such as those in SuperGLUE already contain context and require answering questions given this context. For closed-book tasks, the prompts we provide in the context are not task-specific or contain any relevant knowledge to the inference-time question. The task contains no provided context and the answer must simply be generated given the model’s memorized knowledge, as in the original GPT-3 paper.
> >
> > We believe that the lower performance gain on this style of task compared to tasks with explicitly provided context is an interesting finding of our work. This suggests that natural language reasoning abilities are acquired by models at smaller scales, and scale enables the model to memorize additional facts.
> >
> > Summarization tasks: We focus on supervised tasks (mentioned in Section 3.1), like the GPT-3 paper, which also does not evaluate on summarization. Strategies for summarization are an exciting direction for future work.
> >
> > **Additional Questions**
> > 1. ***How was the model trained?*** We clarify that all the GPT models we test are next-token prefix trained and not span corruption trained (i.e. the EleutherAI, BLOOM, and OPT models of all sizes). T5, which uses span corruption pretraining objectives, is the base pretrained model for the T0 model. T0 is then fine-tuned on instruction-input-output pairs to improve its ability to follow prompt instructions.
> > 2. ***How many templates do we end up using?*** We used 3-6 templates or chains per task. We report this in the main paper in Section 5.
> >
> > We answer the remaining questions you posed in the above sections. Thanks for raising these great points and helping to make the paper stronger!

---

> > > ### Comment · Reviewer_Pytd · 2022-11-28
> > > **response**
> > >
> > > Thank you. Please mention in the main paper that this strategy will not be applicable (in a straightforward manner) to summarization and more open ended tasks. I think it is important to make it clear

---

> > > > ### Author Response · Authors · 2022-11-28
> > > > **Responses**
> > > >
> > > > Thank you for your comments! Please find our responses below:
> > > >
> > > > 1. ***Open-ended tasks:*** We will certainly emphasize in the final paper that the approach will not work out-of-the-box for open-ended tasks. We specifically propose to add the following statement to Section 5.1 in the paper: “AMA does not directly apply to open-ended generation tasks such as summarization. Developing prompt reformatting and aggregation strategies for such tasks is an exciting direction for future work.”
> > > > 2. ***Large model performance:*** We will insert the experiments discussed in our response (i.e., using the larger GPT-6B model to generate questions and the smaller 1.3B model to answer()) to highlight the importance of the question() step. Additionally during the revision stage we added a new table (Appendix B, Table 3), which contains end to end AMA results with the OPT and BLOOM open-source 175B parameter large models.
> > > > 3. ***Abstaining:*** We clarify that we do not come up with the valid answer choices, but use the label-strings that are provided in the original task formulation. You are correct; if the model does not output any valid string, the prediction is marked as an abstain. This occurs 3% of the time for AGNews and 4% of the time for Amazon Products when using the GPT-J-6B model for an AMA prompt. This occurs more frequently when the AMA questions warrant open-ended answers rather than yes-no questions.
> > > > 4. ***End-to-end algorithm:*** We apologize for not including Algorithm 1 in the main paper. We clarify that the information in Appendix D is in the main paper and we propose to include the Algorithm 1 box in the main paper by moving Figure 3 to the appendix in a camera ready version.
> > > >
> > > > Please let us know if these modifications help address your concerns. Thank you again for your suggestions to improve the paper!

---

> > > > > ### Comment · Reviewer_Pytd · 2022-12-12
> > > > > **Do you know how it compares to prompt tuning?**
> > > > >
> > > > > Any idea on how it compares with prompt tuning https://aclanthology.org/2021.emnlp-main.243.pdf?

---

> > > > > > ### Author Response · Authors · 2022-12-12
> > > > > > **Comparing to prompt tuning**
> > > > > >
> > > > > > Thank you for the question! Prompt tuning is a powerful approach, which gives high quality results using small language models and a small number of trainable parameters. We are happy to include this baseline and discussions in the main-paper! We have already added the full fine-tuning baselines to our main results tables as requested by other reviewers during the revision stage.
> > > > > >
> > > > > > We did not compare to prompt tuning strategies originally because the methods operate in different regimes than in-context learning. Specifically, these methods have access to a hyperparemter tuning stage [4] and more labeled [3] and/or unlabeled [1, 2] training points than AMA. For instance, [3] trains on the full labeled training datasets up to 100,000 examples and [1] uses 20,000 unlabeled training examples with 32 labeled examples. An exciting property of in-context learning approaches is the out-of-the-box flexibility. In-context learning can therefore be useful for a set of users that complements the audience of prompt tuning methods. Again, we are very happy to add discussion on prompt tuning baselines!
> > > > > >
> > > > > > Next we provide detailed comparisons to popular prompt tuning baselines. For reference, the average SuperGLUE performance with the 6.7B parameter GPT-J model is 56.7 for few-shot prompting and 73.4 with AMA. On AGNews, performance is 74.5 with few-shot prompting and 86.4 with AMA.
> > > > > >
> > > > > > **Comparing to PET/iPET [1, 2, 4]:** Using the iPET method gives 76.8 on SuperGLUE with ALBERT [1]. iPET performance is achieved by tuning with 32 labeled and 20,000 unlabeled examples. When the 20,000 unlabeled points are not utilized, iPET performance drops to 68.0 from 81.0 (across CB, RTE, and MultiRC) in [1], while AMA performance does not degrade when no unlabeled points are used (Appendix B.5, Figure 6). Notably, AMA performance using majority vote and *no* weak-supervision is higher than iPET with no unlabeled data on average across CB, RTE, and MultiRC (Table 6).
> > > > > >
> > > > > > Further, in experiments that swap ALBERT to RoBERTa, performance drops from 71.8 to 63.7 on average (3 SuperGLUE tasks - CB, RTE, and MultiRC) [1], which is far below AMA.
> > > > > >
> > > > > > [4] presents negative results pertaining to strategies such as [1], finding that these works "have greatly overestimated the few-shot ability of LMs", given their approaches to hyperparameter tuning. In response, the authors of [1] present [2], with a modified hyperparameter tuning approach. In [2], iPET gives 85% on AGNews with RoBERTa, which is below AMA performance (the overlapping task in our evaluations). In [4], with fairer hyperparmeter tuning approaches, AMA with GPT-J-6B outperforms PET on SuperGLUE on average.
> > > > > >
> > > > > > **Comparing to T5 Prompt tuning [3]:** Using the prompt-tuning strategy in [3] gives 90.5 on average on SuperGLUE with an 11B parameter T5-XXL model and prompt-ensembling gives 91.3 on average. Using the 3B parameter T5-XL model gives ~80.0 on average. Both are much higher than AMA.
> > > > > >
> > > > > > The runtime to reach convergence is 50 minutes to 4 hours for the XL and XXL models depending on the prompt length (Appendix of [3]), while AMA does not require training. Notably, hyperparameter tuning is involved so the training costs are multiplied by the number of hyperparameter tuning runs, per task. [3] also has access to the full labeled training datasets (up to 100k points for ReCoRD) to achieve these results, which AMA does not.
> > > > > >
> > > > > > We hope this is helpful, and we are very happy to discuss these works in the final paper!
> > > > > >
> > > > > > References:
> > > > > > 1. Schick and Schütze, It's Not Just Size That Matters: Small Language Models Are Also Few-Shot Learners. In arXiv:2009.07118v2., 2021.
> > > > > > 2. Schick and Schütze, True Few-Shot Learning with Prompts – A Real-World Perspective, In arXiv:2111.13440v1., 2021.
> > > > > > 3. Lester, Al-Rfou, and Constant, The Power of Scale for Parameter-Efficient Prompt Tuning. In Proceedings of the 2021 Conference on EMNLP. 2021.
> > > > > > 4. Perez et al., True Few-Shot Learning with Language Models, In arXiv:2105.11447v1. 2022.

---

> > ### Comment · Reviewer_Pytd · 2022-11-28
> > **clarifications**
> >
> > Dear authors
> > Thanks for your clarifications
> > - Please include these new experiments (larger gpt model) at least in the appendix - i could not find them. I think it provides an important angle
> > - I do see appendix D and it does help. But i do believe that rather putting the meat of the algorithm to the appendix, the paper would have been better if the algorithm was clearly spelled out in the main section. I think several other reviewers also commented on the clarity and being confused as to what is suggested.  It is probably too late for such a surgical revision though...
> > - Re: abstaining. If i read correctly, I think you do suggest to the model what answers are valid. So abstaining is when you did provide valid answers but the model output something outside of these anyway? How often does it happen empirically (e.g. is it 10%? 30%?)
> > Thank you

---

### Official Review · Reviewer_XWv2 · 2022-10-26

**Confidence:** 4
**Correctness:** 4
**Technical Novelty And Significance:** 2
**Empirical Novelty And Significance:** 2
**Recommendation:** 8

**Clarity, Quality, Novelty And Reproducibility:**

This paper is clearly written and seems to be reproducible given that the authors release the prompts and code. See Strengths and Weaknesses section for comments on novelty.

**Strength And Weaknesses:**

Strengths:
- This paper has impressive empirical results in that it up-levels the 6B parameter model to the accuracy of the 175B parameter model for most of the tasks
- In general, the observation that question-based tasks are much more likely to be answered correctly than other formats of tasks is really interesting. This is exactly the sort of quantitative/qualitative evaluation of benchmark datasets that the field needs more of: we often take benchmark datasets for granted, and even though we know that they're flawed in various ways, we still use them to define SOTA.
- Similarly, thinking about what sorts of text features and distributions work "well" with LLMs  (e.g., formatting a task as a natural language question rather than templating it in less natural ways) is often overlooked, even though it has significant impact on the accuracy.
- The error analysis section in the appendix is good-- much more useful for future work than just presenting a raw accuracy number.

Weaknesses:
- My main concern is that I'm not sure this is enough for a full paper. This is a useful analysis and prompt engineering strategy, but I would expect either a deeper analysis of *why* formulating things as Q/A works so much better (e.g., analysis of the training data), or
- I'm confused about the question() prompt. I thought these contained "task-agnostic examples of how to transform statements to various questions", but it seems like the format is not the same for all tasks? (Appendix H)

**Summary Of The Paper:**

The authors present a new "meta" prompting method to achieve better accuracy in few-shot prompting. This is based on the observation that certain types of questions tend to be more amenable to few-shot prompting (specifically, QA-type prompts). This prompting method (AMA), includes:
- Creating question() prompts, which transfer the original task into an open ended question
- Creating answer() prompts, which transfer the result to the actual answer
They then aggregate a range of answers for a given prompt into a final answer, and find a significant performance lift

**Summary Of The Review:**

At a high level, this paper is starting to push back against the idea that LLMs are agnostic to prompt structure (ie, that prompt structure is a unimportant implementation detail), which is an important step for the field. That being said, it feels a bit more like an analysis of single prompt engineering strategy than a full paper, and for that reason, I am on the fence for acceptance.

---

> ### Author Response · Authors · 2022-11-15
> **Response Overview and Pretraining Corpus Analysis**
>
> Thank you for your thoughtful review! We appreciated that you found the results and evaluations impressive and interesting, error analysis useful, and the idea of aligning task formats with the distributions that work well for LLMs to be an important step for the field.
>
> Here we address your questions about the work. We highlight a new analysis of the pretraining data to understand why the QA prompt format may be effective.
>
> ***Novelty and contributions***
> Thank you for raising these concerns! We discuss why we believe AMA is novel and surprising in the main thread/general response. Please find our responses to your questions there!
>
> ***Pretraining data analysis***
> We agree that an interesting question is why the QA prompt format works so much better than restrictive prompts. In light of your suggestions, we conduct an analysis of The Pile pre-training corpus, which was used to train the EleutherAI models [9, 10]. This provides illuminating evidence for why QA works better. We have added this to the paper (Main paper Section 3.2, Appendix H) and thank you for making the paper stronger!
>
> Specifically, we compute the frequency of regular expression matches corresponding to restrictive prompts (i.e., output “True or False.”, “Yes or No”) versus open-ended questions (e.g. “Is …?”, “Who …?”) in a 2% random subsample of the ~200B token Pile corpus, ignoring capitalization (Table 1). We observe that questions are quite frequent relative to the restrictive prompts that appear in the original GPT-3 paper. Further, we find several instances of yes/no questions followed by “yes” or “no”, which mimics the AMA format (Table 2). We find that QA structured text appears much more frequently in the pretraining corpus, which may help explain why the language models perform better on QA prompt formats.
>
> Table 1: Frequency of Various Prompt Templates in the Pretraining Corpus
> |                              | Regular Expression Patterns                                                                                                                                                                                                                                                                                                                                                                                                                                           | Frequency |
> | ---------------------------- | --------------------------------------------------------------------------------------------------------------------------------------------------------------------------------------------------------------------------------------------------------------------------------------------------------------------------------------------------------------------------------------------------------------------------------------------------------------------- | --------- |
> | Restrictive Patterns         | ".\* true or false\\?", ".\* true or false\\.", ".\* true, false, or neither\\?", ".\* true, false, or neither\\.",  ".\* yes or no\\?", ".\* yes or no\\.", ".\* yes, no, or maybe\\?", ".\* yes, no, or maybe\\.",  ".\* correct or incorrect\\?", ".\* correct or incorrect\\.", ".\* correct, incorrect, or inconclusive\\?", ".\* correct, incorrect, or inconclusive\\.",  "choose between:", "pick one from:", | 222       |
> | Yes/No Question Patterns     | "is .\*\\?", "was .\*\\?", "did .\*\\?", "do .\*\\?", "are .\*\\?", "will .\*\\?",                                                                                                                                                                                                                                                                                                                                                                     | 387379    |
> | Open-Ended Question Patterns | "when .\*\\?", "where .\*\\?", "why .\*\\?", "who .\*\\?", "what .\*\\?", "how many .\*\\?"                                                                                                                                                                                                                                                                                                                                                            | 639573    |

---

> > ### Author Response · Authors · 2022-11-15
> > **Pretraining Corpus Analysis Continued**
> >
> > **Table 2:  Yes-No Question and Answer Patterns**
> > |                                  | Regular Expression Frequencies                                                                                                                                                                                                                                                                                                                        |
> > | -------------------------------- | ----------------------------------------------------------------------------------------------------------------------------------------------------------------------------------------------------------------------------------------------------------------------------------------------------------------------------------------------------- |
> > | Yes/No Question + Answer Pattern | “is .\*\\\\? yes": 536, "was .\*\\\\? yes": 248, "did .\*\\\\? yes": 109, "do .\*\\\\? yes": 210, “are .\*\\\\? yes": 233, "were .\*\\\\? yes": 91, "will .\*\\\\? yes": 121, "is .\*\\\\? no": 2356, "was .\*\\\\? no": 983, "did .\*\\\\? no": 534, "do .\*\\\\? no": 935, "are .\*\\\\? no": 978, "were .\*\\\\? no": 422, "will .\*\\\\? no": 423 |
> >
> > When applying the few-shot restrictive prompts, we observe large imbalances in the F1-scores for different classes (see Table 3). Therefore, we next ask if answering the restrictive prompts is challenging due to biases acquired during pretraining. Over the same Pile sample as before, the mean word count is $25.3 \pm 7309$ occurrences. We compute the frequency of individual words in the “restrictive” and “open-ended question” patterns from the above Table 1, and these can be found in Table 4. This leads to two hypotheses about why QA prompts perform well.
> > 1. First we see that there are imbalances between the occurrence of “yes” vs. “no”, and “true” vs. “neither” for instance. This may bias the model towards certain answer choices. Indeed [2] also hypothesizes, but does not provide any analysis over the pretraining corpus, that pretraining may instill particular biases in the model.
> > 2. The frequency of the words in the “question words” categories is typically an order of magnitude larger than those in the “restrictive words” category. We hypothesize that the representations for the “question words” will be the most context-specific, which is useful for the prompting tasks we consider. Findings in [8] support this hypothesis, finding that frequently occurring stop-words have the most context-specific representations.
> >
> > **Table 3: F1-Score by Class using Restrictive vs. AMA Prompts**
> > | Task | Output Space         | F1 Score 0-shot                     | F1 Score Few-Shot with two in-context examples per class | F1 Score AMA QA with a single prompt chain / no aggregation |
> > | --------- | -------------------- | ----------------------------------- | --------------------------------------------------- | ----------------------------------------------------- |
> > | CB        | True, False, Neither | True: 36.8, False: 0.0, Neither: 21.7 | True: 55.6, False: 0.0, Neither: 12.5                 | True: 95.7, False: 92.3 Neither: 28.6                  |
> > | RTE       | True, False          | True: 40.4, False: 58.3              | True: 70.6, False: 31.3                              | True: 58.8, False: 64.9                                |
> > | WSC       | Yes, No              | Yes: 53.5, No: 0.0                   | Yes: 53.3,  No: 13.7                                 | Yes: 61.3, No: 78.2                               |
> >
> > **Table 4: Frequency of Individual Words by Prompt Structure**
> > |  Prompt Structure              | Word Frequency                                                                                |
> > | ------------------------- | --------------------------------------------------------------------------------------------- |
> > | Restrictive words         | true: 69658, false: 41961, neither: 20891, yes: 12391, no: 452042, maybe: 36569 |
> > | Yes/no question words     | is: 3580578, was: 1926273, did: 200659, do: 394140, are: 1441487, will: 619490      |
> > | Open-ended question words | when: 583237, where: 303074, why: 97324, who: 417798, what: 548896, how: 298140     |

---

> > > ### Author Response · Authors · 2022-11-15
> > > **Additional Questions and References**
> > >
> > > **Question Answer prompt structure is general across tasks**
> > > Thank you for raising this concern. We would like to clarify the generality of our prompting approach and we have updated the paper to reflect these details.
> > >
> > > In the submission, we discuss that prompts consist of prompt-templates, which contain placeholders for prompt-examples. The template or structure of the prompt-chains is task-agnostic — for all tasks we form questions and answers. The same in-context examples (not task-specific) are reused to achieve our results on many tasks (ANLIR1, ANLIR2, ANLIR3, BoolQ, CB, COPA, DROP, RTE, StoryCloze). For these we note that 3-6 prompt chains were used per task for aggregation, and we randomly included one per task for Appendix K.
> > >
> > > For other tasks, differences must arise because of the fundamental format of the prompt:
> > > - COPA/StoryCloze contain context and require determining which of two provided sentences is the better continuation.
> > > - NLI tasks require determining the validity of one input statement, given context.
> > > - Some tasks (e.g. NQ, WebQ) come with open-ended questions, but no context.
> > > Because the structure of the task differs, the way in which question() prompts are applied to the task input may need to change, but for each task open-ended (non-restrictive) questions are obtained and then answered. For instance, in our work we convert each of the two statements in the COPA and StoryCloze tasks to questions independently of one another, whereas the NLI tasks just require converting one input statement to a question.
> > >
> > > We have updated the paper to read that the prompting-structure is agnostic. Rather than starting from scratch in prompt design, we hope this template helps users get started. We hope this response is clarifying and our revisions to the paper are satisfying!
> > >
> > > **References**
> > > [1] Gao et al. The Pile: An 800GB Dataset of Diverse Text for Language Modeling. arXiv:2101.00027 2020.
> > > [2] Zhao et al. Calibrate Before Use: Improving Few-Shot Performance of Language Models. PMLR 2021.
> > > [3] Mishra et al. Reframing Instructional Prompts to GPTk’s Language. ACL Findings 2022.
> > > [4] Wu et al. AI Chains: Transparent and Controllable Human-AI Interaction by Chaining Large Language Model Prompts. arXiv:2110.01691 2022.
> > > [5] Liu et al. What Makes Good In-Context Examples for GPT-3? arXiv:2101.06804 2021.
> > > [6] Wei et al. Chain-of-Thought Prompting Elicits Reasoning in Large Language Models. 2022.
> > > [7] Wang et al. Self-Consistency Improves Chain of Thought Reasoning in Language Models. arXiv:2203.11171 2022.
> > > [8] Ethayarajh. How Contextual are Contextualized Word Representations? Comparing the Geometry of BERT, ELMo, and GPT-2 Embeddings. EMNLP 2019.
> > > [9] Black et al. GPT-Neo: Large Scale Autoregressive Language Modeling with Mesh-Tensorflow. 2021.
> > > [10]  Wang and Komatsuzaki. Gpt-j-6b: A 6 billion parameter autoregressive language model, 2021.

---

> > > > ### Comment · Reviewer_XWv2 · 2022-11-28
> > > > **Updated review post author comments**
> > > >
> > > > Updated my my recommendation to 8-- I think the analysis of The Pile makes the paper stronger in explaining where the method's success might be coming from. Thanks for including it.

---

### Author Response · Authors · 2022-11-15
**General Response and Revisions**

We thank our reviewers for their thoughtful reviews and insightful comments. Reviewers consistently commented on our proposed method’s impressive results, noting its significant improvements across models [R-XWv2, R-Pytd, R-CQxp, R-SsWm], a large variety of tasks [R-XWv2, R-CQxp,  R-SsWm], and by a large margin over existing baselines [R-XWv2, R-Pytd,  R-SsWm].

Reviewers also noted the novelty of AMA [R-Pytd,  R-SsWm], scalability of AMA [R-SsWm] and the importance of our findings for the community [R-XWv2, R-CQxp]. Reviewers found the analysis of when and where our proposed method succeeds to be insightful [R-XWv2, R-CQxp].

In our revision, we made several reviewer-suggested changes towards building understanding around our method and explaining why our method works well across tasks and models. We hope these new analyses and experiments demonstrate why AMA is a novel and exciting contribution. These changes include:

1. Clarifying and performing additional experiments to further demonstrate why AMA works [All Reviewers].
- We present a new analysis of the Pile pretraining corpus to study why the QA format is so effective in Section 3.2 and Appendix H [All Reviewers].
- We perform new experiments on 10 benchmarks, where we perform the AMA aggregation step, without re-formatting the prompts to QA, to show the value of both the reformatting and aggregation steps [R-CQxp,  R-SsWm]. This is in Appendix B.4.
- We perform several additional ablations of the weak-supervision method, which are included in Appendix B.5 [R-CQxp,  R-SsWm].

2. Clarifying our method’s components in Appendix D and E [R-Pytd,  R-SsWm].

3. Comparing to and discussing an additional baseline, Self-consistency with Chain of Thought prompting, in Appendix B.6, and including new related works in Section 2 [R-CQxp]

The changes are detailed in the main paper and appendix (the appendix is now attached to the 9 page main paper / main submission file). We clarify that all code was included in our supplementary submission and we will publicly release code for reproducing the results on all 20 benchmarks.

Please find our comments for individual reviewers in their respective threads below, and responses to all reviewers below.

---

> ### Author Response · Authors · 2022-11-15
> **Scope and Aims of Our Study**
>
> Here we clarify the scope and aims of our study, which (a) finds that formatting prompts in QA is effective, (b) proposes an approach for recursively using the LM itself to perform the reformatting. Finally, because reformatting is automatic (i.e. LM-generated) and applied at scale, the process can be noisy, we propose to aggregate over the noisy predictions of multiple prompts. We (c) are the first to apply and investigate how to effectively use weak supervision for aggregation.
>
> We would like to discuss why we believe AMA is novel and surprising. There are several insights underpinning the method.
>
> ***Prompt Design:*** R-CQxp mentions that several prompting approaches have been proposed since GPT-3 and R-XWv2 mentions that our work focuses on a single prompt engineering strategy. We think these points are exactly why the work is so surprising and interesting — amongst the significant amount of prior prompting work we are the first to demonstrate that a 30x parameter smaller model with no additional training can match the performance of GPT3-175B consistently. The overall proposal to reformat prompts to question-answering templates is conceptually simple and this is a strength of the method. It is quite surprising that a ***single prompt template*** (i.e. QA) works well across 15 benchmarks and 14 models (of various sizes).
>
> Categorizing the prior prompt design works:
> 1. ***Manual:*** Some propose search spaces for prompt writing [1, 2] and requires the user to manually experiment with prompt design for every new task. Work in this category has not bridged the gap between small (<10B) and large (>100B) parameter models and is challenging to scale.
> 2. ***Limited improvement:*** Some work proposes off-the-shelf strategies for improving the effectiveness of few-shot prompts such as few-shot example selection [e.g., 3] and model calibration [e.g., 4], but these have not bridged the gap between small (<10B) and large (>100B) parameter models. We also evaluate [4] in our work.
> 3. ***Do not generalize:*** A few works propose that a single prompt template is broadly effective (e.g., Chain of Thought prompting), like we do in AMA, but these are not reported to be effective for small LMs (<10B) [5, 6]. We add new evaluations of [6] in our work (Appendix B.6), thanks to the great suggestion from R-CQxp.
>
> ***We further include new analysis of the Pile corpus,*** which is used to pretrain the EleutherAI LMs, to study why the single QA prompt structure may be so effective, following the great suggestion from R-XWv2!
>
> ***Aggregation:*** R-CQxp mentions that several prior works use prompt aggregation. We agree and discuss prior work in our manuscript. Though again, this is exactly why our work is quite surprising and important. We are the first to show that the strategy for prompt aggregation in the prior work – Majority Vote – is unreliable. We show that prompts can have dependencies and varying accuracies, motivating the use of Weak Supervision for the first time in prompt aggregation. We build off of a particular weak supervision framework that can model dependencies and varying class-conditional accuracies, and we modify it with an initial dependency learning step to yield a reliable aggregation method that can identify and model complex relationships among the prompt outputs and true labels. We show weak supervision can achieve up to 8.7 points of lift over the majority vote strategy of prior work (Section 5.3), improving the reliability of prompt aggregation. Further, our weak supervision approach requires no labeled data! As R-CQxp notes, [6] also uses Majority Vote, so we hope our study encourages aggregation strategies that account for the dependencies and varying accuracies of prompts.
>
> ***Putting everything together:*** R-CQxp raised a concern that it is not clear why the method includes two ideas – prompt reformatting and aggregation – rather than focusing on one. The need for scalability is the key insight tying the two ideas together. R-Pytd and  R-SsWm appreciate that AMA is a scalable approach, as AMA recursively uses the LM to convert task inputs to the effective QA prompt template. Because the LM is producing the reformatting, and LMs are probabilistic, the transformations can be quite noisy! Especially in applying AMA at scale to thousands of examples, it becomes challenging to audit each transformation. Our scalable solution for QA reformatting is noisy, so aggregation is important to manage the noise. To summarize, reformatting at scale and aggregation go hand-in-hand to yield a scalable and effective prompting strategy!
>
> We hope this discussion is helpful to see the underlying insights and principles introduced in this work, that allow the simple method to achieve the results that all four reviewers note as impressive. Thank you for your questions and thoughtful reviews!

---

> > ### Author Response · Authors · 2022-11-15
> > **Corresponding Citations**
> >
> > [1] Mishra et al. Reframing Instructional Prompts to GPTk’s Language. ACL Findings 2022.
> >
> > [2] Wu et al. AI Chains: Transparent and Controllable Human-AI Interaction by Chaining Large Language Model Prompts. arXiv:2110.01691 2022.
> >
> > [3] Liu et al. What Makes Good In-Context Examples for GPT-3? arXiv:2101.06804 2021.
> >
> > [4] Zhao et al. Calibrate Before Use: Improving Few-Shot Performance of Language Models. PMLR 2021.
> >
> > [5] Wei et al. Chain-of-Thought Prompting Elicits Reasoning in Large Language Models. 2022.
> >
> > [6] Wang et al. Self-Consistency Improves Chain of Thought Reasoning in Language Models. arXiv:2203.11171 2022.

---

### Decision · Program_Chairs · 2023-01-20

**Decision:**

Accept: notable-top-25%

**Justification For Why Not Higher Score:**

I think recommending this paper for a spotlight is already a bit of a stretch, although not without justification. I would recommend raising the score further without multiple reviewers giving a stronger rating.

**Justification For Why Not Lower Score:**

The reviewers unanimously support acceptance. The paper is interesting. There is no good reason to reject it.

**Metareview: Summary, Strengths And Weaknesses:**

The reviewers unanimously support the publication of this paper, which proposes a way of effectively ensembling different versions of the same LLM only differentiated by the prompt, with the reasoning that albeit imperfect, the prompts condition the model in complementary enough ways that aggregating their predictions yields a surprisingly good model. The idea is cool, and while there are aspects of the presentation and narrative which could be improved and clarified (see reviewer comments), the paper is certainly worthy of publication. I would go so far as recommending it for a spotlight, not solely on the basis of the scores, but rather on the basis that the work will spark interest and discussion within the community and is quite original.

**Note From Pc:**

if the above contains the word "oral" or "spotlight" please see: "oral" presentation means -> notable-top-5% and "spotlight" means -> notable-top-25%. As stated in our emails, we are disassociating presentation type from AC recommendations